# Fractional Diffusion Bridge Models

**Gabriel Nobis** [* ✉]
Fraunhofer HHI

**Maximilian Springenberg** [* ✉]
Fraunhofer HHI

**Arina Belova**
Fraunhofer HHI

**Rembert Daems**
Ghent University–imec
FlandersMake–MIRO

**Christoph Knochenhauer**
Technical University of Munich

**Manfred Opper**
Technical University of Berlin
University of Potsdam
University of Birmingham

**Tolga Birdal** [†]
Imperial College London

**Wojciech Samek** [†]
Fraunhofer HHI
Technical University of Berlin

## Abstract

We present *Fractional Diffusion Bridge Models* (FDBM), a novel generative diffusion bridge framework driven by an approximation of the rich and non-Markovian fractional Brownian motion (fBM). Real stochastic processes exhibit a degree of memory effects (correlations in time), long-range dependencies, roughness and anomalous diffusion phenomena that are not captured in standard diffusion or bridge modeling due to the use of Brownian motion (BM). As a remedy, leveraging a recent Markovian approximation of fBM (MA-fBM), we construct FDBM that enable tractable inference while preserving the non-Markovian nature of fBM. We prove the existence of a coupling-preserving generative diffusion bridge and leverage it for future state prediction from paired training data. We then extend our formulation to the Schrödinger bridge problem and derive a principled loss function to learn the unpaired data translation. We evaluate FDBM on both tasks: predicting future protein conformations from aligned data, and unpaired image translation. In both settings, FDBM achieves superior performance compared to the Brownian baselines, yielding lower root mean squared deviation (RMSD) of $C_\alpha$ atomic positions in protein structure prediction and lower Fréchet Inception Distance (FID) in unpaired image translation.

## 1 Introduction

Stochastic differential equations (SDEs) offer a natural framework for modeling the inherent randomness and continuous-time dynamics of real-world systems [1, 2]. This is precisely why they serve as the backbone of state-of-the-art generative diffusion models [3–5]. Traditionally, these models assume noise driven by standard Brownian motion (BM) [6–8], which is Markovian with independent increments [9]. However, this choice is motivated by mathematical tractability and simplicity rather than faithfulness and fidelity to real-world data. Empirical data, particularly in complex systems such as proteins, often exhibit long-range temporal dependencies, heavy-tailed behaviors, and intricate dynamics that are poorly captured by memoryless processes [10]. A generative process, lacking temporal dependencies, may lead to insufficient approximations of such intricate data, due to the absence of modeled memory effects. These limitations have motivated recent efforts to explore generative models with non-standard noise sources [11–18]. Our work extends

---

[*] Equal contribution; [†] Shared senior authorship

[✉] corresponding authors: {gabriel.nobis, maximilian.springenberg}@hhi.fraunhofer.de

39th Conference on Neural Information Processing Systems (NeurIPS 2025).

this line of research to generative diffusion bridge models [19–21], where the goal is to transform a structured, non-Gaussian source distribution into a complex target distribution. We specifically investigate stochastic bridges driven by fractional Brownian motion (fBM) [22, 23], a generalization of BM with dependent increments, characterized by the Hurst index $H$, which governs both roughness (i.e., pathwise regularity) and long-range dependence. However, directly using fBM as the driving noise in a stochastic bridge introduces an intractable drift [24]. To address this, we adopt a Markov approximation of fBM (MA-fBM) [25, 26] that enables efficient simulation. By using MA-fBM as the driving process, we introduce a more expressive and flexible framework for building bridges: when $H = 0.5$, fBM recovers classical BM, whereas other values of $H$ flexibly allow us to model a broader range of temporal behaviors, as demonstrated in our experiments. Our framework, *Fractional Diffusion Bridge Models (FDBM)*, enables generative bridge modeling with fractional noise for both paired and unpaired training data, applicable across a broad range of machine learning tasks. In this work, we focus on predicting conformational changes in proteins to explore effects in paired-data problems, as well as unpaired image translation. In the context of protein generation, diffusion processes driven by MA-fBM have proven effective in their superdiffusive regime, showing improvements in both sample fidelity and diversity [17], potentially due to a better capture of long-range correlations in protein structures. Building on this observation, we propose MA-fBM-driven diffusion bridges as a principled extension for modeling conformational changes in proteins. To the best of our knowledge, our framework is the first to incorporate fractional noise into generative bridge modeling within machine learning. Our contributions are:

- We propose a method for learning generative diffusion bridges that interpolate between two unknown distributions via a non-Markovian trajectory with controllable correlation of increments and long-range dependencies, enabling more flexible modeling of real-world variability and biological dynamics.
- We prove that, for these generalized stochastic dynamics, there exists a process solving a stochastic differential equation that preserves the coupling given in the training data.
- We formulate the Schrödinger bridge problem with a reference process approximating fractional Brownian motion and propose a method to learn stochastic transport trajectories, whose roughness and long-range dependencies are controlled by the Hurst index.
- We apply our framework broadly to common use cases of stochastic bridges in machine learning, including inferring conformational changes in proteins and performing unpaired image translation, achieving lower root mean squared deviation (RMSD) of $C_\alpha$ atomic positions in protein prediction, and improved Fréchet Inception Distance (FID) scores for image translation.

We accompany our work with several publicly available implementations to facilitate the adoption of our framework in both paired and unpaired settings, as well as a stand-alone reimplementation of the method proposed by Bortoli et al. [27].[1] [2] [3]

## 2 Background

Stochastic bridges interpolate between two given data points by conditioning a prior reference process to start and end at prescribed values. A common choice for this reference process in machine learning is a scaled BM $X = \sqrt{\varepsilon}B$ with $\varepsilon > 0$. Conditioning on the endpoints $(x_0, x_1) \in \mathbb{R}^d \times \mathbb{R}^d$ yields the scaled Brownian bridge (BB) $X_{|0,1}$, which starts at $x_0$ and ends at $x_1$, while evolving for $t \in (0,1)$ according to the stochastic dynamics [28]

$$\mathrm{d}X_{|0,1}(t) = \varepsilon \frac{x_1 - X_{|0,1}(t)}{1-t}\mathrm{d}t + \sqrt{\varepsilon}\mathrm{d}B_t, \quad X_{|0,1}(0) = x_0. \tag{1}$$

This scaled BB, or a generalization thereof, serves as the starting point for many machine learning applications [20, 21, 27, 29–34], where the goal is to learn a stochastic process $X^\star$ that interpolates not only between the fixed endpoints $(x_0, x_1)$, but in law between two unknown distributions $\Pi_0$ and $\Pi_1$ on $\mathbb{R}^d$. Since the drift of such a stochastic process is generally intractable, the drift term in eq. (1) serves as a target for a neural network, which is optimized by minimizing a conditional expectation.

**Coupling-preserving data translation**. Data translation aims to map between two unknown distributions. In the setting where training data is provided in pairs—such as the unbound and bound

---

[1] https://github.com/GabrielNobis/FDBM_paired
[2] https://github.com/mspringe/FDBM_unpaired
[3] https://github.com/mspringe/Schroedinger-Bridge-Flow

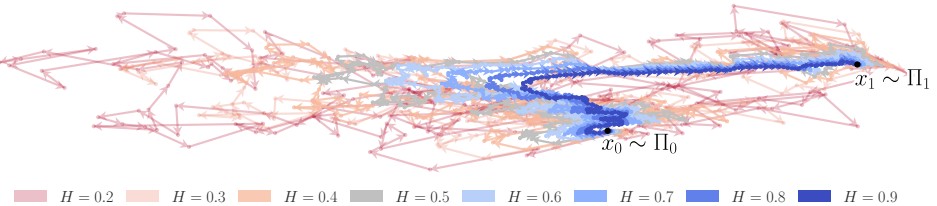

Figure 1: Trajectories from the approximate $2d$-fractional Brownian bridge for different Hurst indices $H$.

states of a protein [31, 35], a distorted and a clean image [36, 37], or two snapshots of cell differentiation recorded on different days [31]—the additional objective is to *preserve* the coupling given in the training data. We build our framework for paired data translation on Augmented Bridge Matching (ABM) [32], where a stochastic process $X^\star$ is learned that transports an unknown distribution $\Pi_0$ on $\mathbb{R}^d$ to another unknown distribution $\Pi_1$ on $\mathbb{R}^d$, while preserving the coupling $(X_0^\star, X_1^\star) \sim \Pi_{0,1}$ on $\mathbb{R}^d \times \mathbb{R}^d$. Additionally, $X^\star$ solves an SDE such that we can sample from the coupling $(x_0, x_1) \sim \Pi_{0,1}$ by first sampling $X_0^\star = x_0 \sim \Pi_0$ according to the first marginal of $\Pi_{0,1}$, and then simulating the SDE forward in time to arrive at a sample $X_1^\star = x_1 \sim \Pi_1$. Bortoli et al. [32, Proposition 3] show that for the scaled Brownian reference process $X = \sqrt{\varepsilon}B$ there exists such a coupling preserving process $X^\star$ with associated path measure $\mathbb{P}^\star$ that solves

$$\mathrm{d}X_t^\star = \varepsilon \mathbb{E}_{\mathbb{P}_{1|0,t}^\star}\left[\frac{X_1^\star - X_t^\star}{1-t}\Big|X_t^\star, X_0^\star\right]\mathrm{d}t + \sqrt{\varepsilon}\mathrm{d}B_t, \quad X_0^\star \sim \Pi_0. \tag{2}$$

The drift of $X^\star$ is intractable and approximated by a time-dependent neural network $v_t^\theta$, resulting in a process $X^\theta$ with associated path measure $\mathbb{P}^\theta$. Minimizing now the KL divergence $D_{\mathrm{KL}}(\mathbb{P}^\star|\mathbb{P}^\theta)$ with respect to the weight vector $\theta$ yields the loss function

$$\mathcal{L}_{ABM}(\theta) := \int_0^1 \mathbb{E}_{\mathbb{P}^\star}\left[\left\|v_t^\theta(X_0^\star, X_t^\star) - \frac{X_1^\star - X_t^\star}{1-t}\right\|^2\right]\mathrm{d}t. \tag{3}$$

Given paired training data sampled from the unknown coupling $\Pi_{0,1}$, we can approximate the above loss function since, by construction, $\mathbb{P}^\star = \Pi_{0,1}\mathbb{Q}_{|0,1}$, where $\mathbb{Q}_{|0,1}$ denotes the path measure of the scaled BB $X_{|0,1}$ solving eq. (1). Consequently, to compute the loss during training, we first sample $(x_0, x_1) \sim \Pi_{0,1}$ and then sample $x_t \sim \mathbb{Q}_{t|0,1}(\cdot \mid x_0, x_1)$.

**Unpaired data translation via the Schrödinger bridge**. On the other hand, in unpaired data translation via the Schrödinger bridge, the objective is to *find* the coupling that corresponds to the optimal transport [38] between two unknown distributions. Here, we aim to learn the stochastic process $X^{SB}$ corresponding to the solution of the dynamic Schrödinger bridge problem [39–42]

$$\mathbb{P}^{SB} = \underset{\mathbb{T}\in\mathcal{P}(\mathcal{C}^d)}{\arg\min}\left\{D_{\mathrm{KL}}(\mathbb{T}|\mathbb{Q}) \; ; \; \mathbb{T}_0 = \Pi_0, \; \mathbb{T}_1 = \Pi_1\right\}, \tag{4}$$

where the minimization is taken over all path measures $\mathbb{T}$ defined on the set of continuous functions $\mathcal{C}^d$ from the unit interval $[0, 1]$ to $\mathbb{R}^d$. We build our framework for unpaired data translation on Schrödinger Bridge Flow (SBFlow) [27], whose unique stationary point corresponds to the Schrödinger bridge. See Section E for a detailed summary.

In the following, we incorporate fractional noise into generative diffusion bridge models in order to control the roughness and long-range dependencies of the interpolating stochastic trajectories, replacing the BM used as the driving noise in traditional diffusion bridge models. Our work builds directly on Daems et al. [26] for the approximation of fBM, on Somnath et al. [31] and Bortoli et al. [32] for the paired-data setting, and on Peluchetti [33], Shi et al. [34], and Bortoli et al. [27] for the unpaired-data setting. See Section D for a detailed discussion of related work.

## 3 A stochastic bridge driven by fractional noise

We first define and characterize the fractional noise that serves as the driving process replacing BM. For mathematical details, we refer the reader to Section B, along with the notational conventions in Section A.

### 3.1 Fractional noise

We begin with the definition of Riemann-Liouville (Type II) fBM, a non-Markovian, centered Gaussian process with non-stationary and correlated increments.

**Definition 1** (Type II Fractional Brownian motion [22]). *Let $B = (B_t)_{t \geq 0}$ be a (multidimensional) standard Brownian motion (BM) and $\Gamma$ the Gamma function. The centered Gaussian process*

$$B_t^H := \frac{1}{\Gamma(H + \frac{1}{2})} \int_0^t (t - s)^{H - \frac{1}{2}} \mathrm{d}B_s, \quad t \geq 0, \tag{5}$$

*is called Type II fractional Brownian motion (fBM) with Hurst index $H \in (0, 1)$.*

Compared to BM with independent increments (diffusion), the paths of fBM become smoother for $H > 0.5$ due to positively correlated increments (super-diffusion) and rougher for $H < 0.5$ due to negatively correlated increments (sub-diffusion), while $H = 0.5$ recovers BM. A stochastic bridge can be derived for Gaussian processes, including fBM; however, the drift of the fractional Brownian bridge (fBB) is intractable [24] and therefore unsuitable both for sampling from its marginals and as a loss-function target analogous to eq. (3). Rather than introducing an additional approximation error by attempting to approximate the drift of the fBB, we follow Harms and Stefanovits [25], Daems et al. [26] and first approximate fBM by a linear superposition of Ornstein–Uhlenbeck (OU) processes. These augmenting OU processes are all driven by the same standard BM, thereby approximating the time-correlated behavior of fBM.

**Definition 2** (Markov approximation of fBM [25, 26]). *Choose $K \in \mathbb{N}$ Ornstein–Uhlenbeck (OU) processes*

$$Y_t^k := \int_0^t e^{-\gamma_k(t-s)} \mathrm{d}B_s, \quad k = 1, \ldots, K, \quad K \in \mathbb{N}, \quad t \geq 0, \tag{6}$$

*with speeds of mean reversion $\gamma_1, ..., \gamma_K$ and dynamics $\mathrm{d}Y_t^k = -\gamma_k Y_t^k \mathrm{d}t + \mathrm{d}B_t$. Given a Hurst index $H \in (0, 1)$ and a geometrically spaced grid $\gamma_k = r^{k-n}$ with $r > 1$ and $n = \frac{K+1}{2}$ we call the process*

$$\hat{B}_t^H := \sum_{k=1}^K \omega_k Y_t^k, \quad H \in (0, 1), \quad t \geq 0, \tag{7}$$

*(multidimensional) Markov-approximate fractional Brownian motion (MA-fBM) with approximation coefficients $\omega_1, ..., \omega_K \in \mathbb{R}$.*

While the choice of approximation coefficients in Harms [43] enables strong convergence to fBM with high polynomial order in $K$ for $H < 0.5$, we opt for the computationally more efficient method proposed by Daems et al. [26]. This method selects the $L^2(\mathbb{P})$ optimal approximation coefficients for a given $K$, achieving empirically good results in approximating fBM, even with a small number of OU processes. See Daems, Rembert [44, Figures 3.13–3.15] for the approximation error of Type II fBM. We fix $K = 5$ throughout all experiments presented in the main text.

**Proposition 3** (Optimal Approximation Coefficients [26]). *The optimal approximation coefficients $\omega = (\omega_1, ..., \omega_K) \in \mathbb{R}^K$ for a given Hurst index $H \in (0, 1)$, a terminal time $T > 0$ and a fixed geometrically spaced grid to minimize the $L^2(\mathbb{P})$-error*

$$\mathcal{E}(\omega) := \int_0^T \mathbb{E}\left[\left(B_t^H - \hat{B}_t^H\right)^2\right] \mathrm{d}t \tag{8}$$

*are given in closed form by the linear system $A\omega = b$, where $A \in \mathbb{R}^{K,K}$ and $b \in \mathbb{R}^K$ are known.*

We now use MA-fBM, equipped with the optimal approximation coefficients, as a reference process to approximate a fBB, thereby enabling efficient simulation and closed-form drift computation in the stochastic bridge derived in the next section.

### 3.2 A Markov approximate fractional Brownian bridge

Towards the goal of defining a stochastic bridge driven by fractional noise we fix the reference process to $X = \sqrt{\varepsilon}\hat{B}^H$ with $\varepsilon > 0$, and write $Y = (Y^1, \ldots, Y^K)$ for the vector of the OU processes and $Z = (X, Y)$ for the augmented reference process. The reference process $X$ is non-Markovian (see Theorem 8) and becomes Markovian only after augmenting it with the OU processes, resulting in the Markovian process $Z$. To define a stochastic bridge connecting two given data points $x_0 \sim \Pi_0$ and $x_1 \sim \Pi_1$ via X, we only need to steer the first dimension of $Z$ towards $x_1$, while the terminal values of $Y$ are not required to attain a specific value. The dynamics of the resulting stochastic bridge $Z_{|x_0,x_1}$ can be derived directly from Daems, Rembert [44, Chapter 4], where a posterior SDE steered towards $x_1$ is constructed. In Section B, we present an alternative derivation using Doob's $h$-transform [2]. Both approaches yield the dynamics stated in the following proposition.

**Proposition 4** (Markov approximation of a fractional Brownian bridge [44, 45]). *The partially pinned process* $Z_{|x_0,x_1} := Z|(X_0 = x_0, X_1 = x_1)$ *solves for* $d = 1$ *the SDE*

$$dZ_{|x_0,x_1}(t) = F Z_{|x_0,x_1}(t)dt + GG^T u(t, Z_{|x_0,x_1}(t))dt + GdB(t), \quad Z_{|x_0,x_1}(0) = (x_0, 0_K), \quad (9)$$

$$u(t, z) = [1, \omega_1 \zeta_1(t, 1), ..., \omega_K \zeta_K(t, 1)]^T \frac{x_1 - \mu_{1|t}(z)}{\sigma_{1|t}^2}, \quad (10)$$

*where* $F \in \mathbb{R}^{K+1, K+1}$ *and* $G \in \mathbb{R}^{K+1}$ *are known,* $\zeta_k(t, t+s) := \sqrt{\varepsilon}(e^{\gamma_k s} - 1)$ *and* $\mu_{1|t}(z)$ *and* $\sigma_{1|t}^2$ *denote the mean and the variance of the conditional terminal* $X_1|(Z_t = z)$, *respectively. We call the process* $Z_{|x_0,x_1}$ *a scaled Markov-approximate fractional Brownian bridge (MA-fBB).*

See Figure 1 for a visualization of two-dimensional MA-fBB trajectories for different Hurst indices. We now incorporate fractional noise into generative diffusion bridge models by using the defined MA-fBB for both paired and unpaired data translation.

## 4 Fractional diffusion bridge models

**Paired data translation**. Paired training data arises in tasks such as predicting conformational changes in proteins, where the unbound and bound states of the same protein form a pair [31, 35, 46];

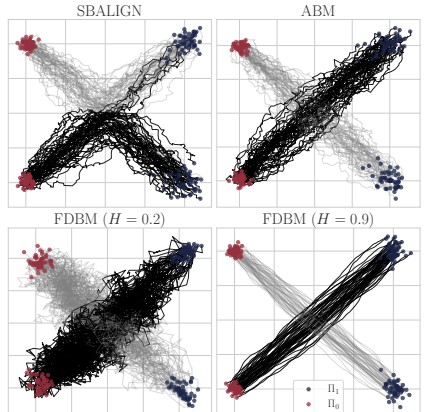

Figure 2: Illustration of FDBM coupling-preserving property shown in Theorem 5: ABM and FDBM preserve the intended coupling, unlike SBALIGN, while FDBM offers a broader range of trajectories.

forecasting the future state of a cell, where two snapshots of cell differentiation are recorded on different days [31]; or reconstructing a clean image from its distorted counterpart [36, 37]. We assume access to paired training data $(x_0^i, x_1^i)_{1 \le i \le N}$ independently sampled from the unknown coupling $(x_0^i, x_1^i) \sim \Pi_{0,1}$ on $\mathbb{R}^d \times \mathbb{R}^d$, with unknown marginals $\Pi_0$ and $\Pi_1$ on $\mathbb{R}^d$. The goal is to transport $\Pi_0$ to $\Pi_1$ via stochastic trajectories driven by MA-fBM, while preserving the coupling $\Pi_{0,1}$. To this end, we construct in the following proposition a stochastic process $X^\star$ that preserves the coupling in the sense that $(X_0^\star, X_1^\star) \sim \Pi_{0,1}$, and that solves an SDE, generalizing the result of Bortoli et al. [32] to a driving MA-fBM. See Section B.2 for the proof.

**Proposition 5.** *Fix the non-Markovian reference process* $X = \sqrt{\varepsilon}\hat{B}^H$ *with associated path measure* $\mathbb{Q}$, *and denote by* $Z = (X, Y)$ *the augmented reference process with associated path measure* $\mathbb{S}$. *We write* $\mathbb{S}_{1|t}^1$ *for the conditional distribution of* $X_1|Z_t$. *Recall that* $\mathbb{Q}_{|0,1}$ *denotes the path measure of the references process* $X$ *conditioned on* $(x_0, x_1) \in \mathbb{R}^d \times \mathbb{R}^d$, *and define* $\mathbb{P} = \Pi_{0,1}\mathbb{Q}_{|0,1}$ *by integrating* $(x_0, x_1)$ *with respect to* $\Pi_{0,1}$. *Assuming that* $\mathbb{P}$ *is absolutely continuous with respect to* $\mathbb{Q}$ *we can lift the path measure* $\mathbb{P}$ *to a coupling preserving path measure* $\mathbb{P}^\star$ *on the augmented space. Under the additional Assumption 2, the SDE*

$$dZ_t^\star = F Z_t^\star dt + GG^T \mathbb{E}_{\mathbb{P}_{1|0,t}^\star}[\nabla_z \log \mathbb{S}_{1|t}^1(X_1^\star|Z_t^\star)|Z_0^\star, Z_t^\star]dt + GdB_t, \quad (11)$$

*with initial vector* $Z_0^\star = (X_0, 0 \ldots 0)$ *admits a pathwise unique strong solution* $Z^\star = (X^\star, Y^\star)$ *with distribution* $\mathbb{P}^\star$. *In particular,* $X^\star$ *preserves the coupling* $\Pi_{0,1}$, *that is,* $(X_0^\star, X_1^\star) \sim \Pi_{0,1}$.

Given a data point $X_0^\star = x_0 \sim \Pi_0$, and assuming we could simulate the coupling preserving process $Z^\star$, we could sample from the coupling $\Pi_{0,1}$ by simulating the SDE in eq. (11) forward in time on $[0, 1]$ to arrive at a sample $X_1^\star = x_1$. As $X^\star$ preserves the coupling, it follows that $(x_0, x_1)$ is drawn from $\Pi_{0,1}$. However, the expectation in the drift of $Z^\star$ is intractable and hence we approximate this expectation by a time-dependent neural network $u^\theta$. We now define *Fractional Diffusion Bridge Models (FDBM)* for paired data translation as the stochastic process $Z^\theta$ associated with the path measure $\mathbb{P}^\theta$ solving

$$dZ_t^\theta = F Z_t^\theta dt + GG^T u^\theta(t, X_0, Z_t^\theta)dt + GdB_t, \quad Z_0^\theta = (X_0, 0, \ldots, 0), \quad (12)$$

$$u_i^\theta(t, x_0, z) = [1, \omega_1 \zeta_1(t, 1), \ldots, \omega_K \zeta_K(t, 1)]^T \tilde{u}_i^\theta(t, x_0, \mu_{1|t}(z)), \quad u^\theta = (u_1^\theta, \ldots, u_d^\theta), \quad (13)$$

where $\tilde{u}^\theta = (\tilde{u}_1^\theta, \ldots, \tilde{u}_d^\theta)$ is a time-dependent neural network that takes the starting value $x_0$ and the mean $\mu_{1|t}(z)$ of the conditional terminal $X_1|(Z_t = z)$ as an input. Note that the output dimensionality of the neural network $\tilde{u}^\theta$, trained in the following, correspond to the data dimension $d$. It is only scaled via eq. (94) to obtain $u_t^\theta$, which has the output dimensionality of the augmented space. Hence, for FDBM, we can employ exactly the same model architectures as in ABM and simply transform the network input and output according to eq. (94). As a result, replacing BM with MA-fBM incurs minimal additional computational cost compared to ABM, as shown in Section H. To train FDBM for paired data translation we derive in Section B.2 the KL-divergence $D_{\mathrm{KL}}(\mathbb{P}^\star|\mathbb{P}^\theta)$, which yields the loss function

$$\mathcal{L}_{\mathrm{FDBM}}^{\mathrm{paired}}(\theta) := \int_0^1 \mathbb{E}_{\mathbb{P}^\star}\left[\left\|\left\|\frac{X_1^\star - \mu_{1|t}(Z_t^\star)}{\sigma_{1|t}^2} - \tilde{u}^\theta(t, X_0, \mu_{1|t}(Z_t^\star))\right\|\right\|^2\right] \mathrm{d}t. \tag{14}$$

To compute the above loss during training, we first sample $(x_0, x_1) \sim \Pi_{0,1}$ and $t \sim \mathcal{U}[0,1]$, and then sample $z_t \sim \mathbb{S}_{t|X_0,X_1}(\,\cdot\,|x_0, x_1)$. This is justified since $\mathbb{P}^\star = \Pi_{0,1}\mathbb{S}_{|X_0,X_1}$ by Corollary 12.

To provide a first proof of concept of FDBM in the paired data setting, and in particular to illustrate the practical implications of Theorem 5, we replicate the toy experiment from Bortoli et al. [32, Figure 1]. Initial samples from a Gaussian centered at $(-2, -2)$ are paired with a Gaussian centered at $(2, 2)$, and samples from a Gaussian centered at $(-2, 2)$ are paired with one centered at $(2, -2)$. In Figure 2, we observe that this coupling is not preserved by SBALIGN, in contrast to ABM. Consistent with Theorem 5, FDBM preserves the intended coupling while offering a broader range of trajectories. In the rough regime ($H = 0.2$), trajectories explore a larger portion of the space, whereas in the smooth regime ($H = 0.9$), nearly straight-line paths emerge.

**Unpaired data translation via optimal transport**. For unpaired data translation, the goal is again to transport $\Pi_0$ to $\Pi_1$, but the training data consist of unpaired samples from $\Pi_0$ and $\Pi_1$ without a given coupling. The dynamic formulation of Entropic Optimal Transport (EOT) seeks the transport plan between $\Pi_0$ and $\Pi_1$ as the solution to the Schrödinger Bridge (SB) problem [42], which induces the corresponding optimal coupling. In the SB problem, the reference process defines the underlying stochastic dynamics that regularize the transport, determining how probability mass evolves between $\Pi_0$ and $\Pi_1$. We replace in the following the BM commonly used as a reference process in the formulation of SB problems in machine learning [19–21, 27, 29, 30, 33, 34] Let $X = \sqrt{\varepsilon}\hat{B}^H$ be our scaled MA-fBM reference process associated with the non-Markovian path measure $\mathbb{Q}$. We seek a solution to the dynamic Schrödinger Bridge problem

$$\mathbb{T}^{SB} = \underset{\mathbb{T} \in \mathcal{P}(\mathcal{C}^d)}{\arg\min} \left\{ D_{\mathrm{KL}}(\mathbb{T}|\mathbb{Q}) \; ; \; \mathbb{T}_0 = \Pi_0, \; \mathbb{T}_1 = \Pi_1 \right\}. \tag{15}$$

We assume that $\mathbb{T}^{SB}$ denotes a solution to eq. (15), inducing the coupling $\Pi_{0,1}^{SB} := \mathbb{T}_{0,1}^{SB}$. Assuming that $\mathbb{P} := \Pi_{0,1}^{SB}\mathbb{S}_{X_0,X_1}$ is absolutely continuous with respect to $\mathbb{Q}$ and under Assumption 2, we can, via Proposition 5, construct the $\Pi_{0,1}^{SB}$-coupling preserving path measure $\mathbb{P}^\star$ associated to the process $Z^\star = (X^\star, Y^\star)$ following the dynamics eq. (11). On the other hand, letting $\mathbb{S}$ be the path measure associated with the augmented reference process $Z$, we define using the marginals of $\mathbb{P}^\star$ the SB problem on the augmented space via

$$\mathbb{V}^{SB} = \underset{\mathbb{V} \in \mathcal{P}(\mathcal{C}^{d\cdot(K+1)})}{\arg\min} \left\{ D_{\mathrm{KL}}(\mathbb{V}|\mathbb{S}) \; ; \; \mathbb{V}_0 = \mathbb{P}_0^\star, \; \mathbb{V}_1 = \mathbb{P}_1^\star \right\}. \tag{16}$$

Since $Z$ is a Markov process, the path measure solving the lifted SB problem in eq. (16) is associated with a Markovian process [42], whereas $Z^\star$ in eq. (11) is non-Markovian due to its dependency on $X_0$ in the drift function. Motivated by this observation, we generalize in the following the definition of a reciprocal class [34, 47] and the notation of a Markovian projection [21, 34, 48] to our setting of a scaled MA-fBM reference process. We define the augmented reciprocal class $\mathcal{R}_a(\mathbb{S})$ of $\mathbb{S}$ as the set of path measures $\mathbb{V}$ on the augmented space whose marginals can be sampled by first drawing $(x_0, x_1) \sim \mathbb{V}_{X_0,X_1}$ and then sampling $z_t \sim \mathbb{S}_{t|X_0,X_1}(\,\cdot\,|\,x_0, x_1)$.

**Definition 6.** *We say that $\mathbb{V} \in \mathcal{P}(\mathcal{C}^{d\cdot(K+1)})$ is in the augmented reciprocal class $\mathcal{R}_a(\mathbb{S})$ of $\mathbb{S}$ if*

$$\mathbb{V} = \int_{\mathbb{R}^d \times \mathbb{R}^d} \mathbb{S}_{|X_0,X_1}(\,\cdot\,|x_0, x_1)\mathrm{d}\mathbb{V}_{X_0,X_1}(x_0, x_1) =: \mathbb{V}_{X_1,X_0}\mathbb{S}_{|X_0,X_1}. \tag{17}$$

*For any $\mathbb{V} \in \mathcal{P}(\mathcal{C}^{d\cdot(K+1)})$ we define the augmented reciprocal projection by*

$$\mathrm{proj}_{\mathcal{R}_a(\mathbb{S})}(\mathbb{V}) := \mathbb{V}_{X_0,X_1}\mathbb{S}_{|X_0,X_1}. \tag{18}$$

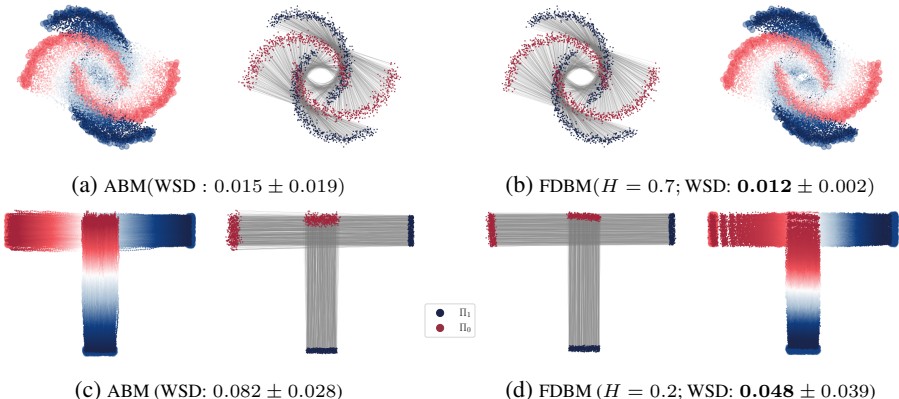

(a) ABM(WSD : $0.015 \pm 0.019$)  (b) FDBM($H = 0.7$; WSD: $\mathbf{0.012} \pm 0.002$)

(c) ABM (WSD: $0.082 \pm 0.028$)  (d) FDBM ($H = 0.2$; WSD: $\mathbf{0.048} \pm 0.039$)

Figure 3: Qualitative comparison on *Moons* and *T-Shape*. Plots and datasets design follow Somnath et al. [31].

Since we know that the solution to the lifted SB problem in eq. (16) is a Markovian measure, we project any element of the augmented reciprocal class to a Markovian path measure by the following definition.

**Definition 7.** *For* $\mathbb{V} \in \mathcal{P}(\mathcal{C}^{d \cdot (K+1)})$ *with* $\mathbb{V} \in \mathcal{R}_a(\mathbb{S})$ *we define the augmented Markovian projection* $\mathrm{proj}_{\mathcal{M}_a}(\mathbb{V})$ *by the path measure associated to* $M = (M^1, M^2, \dots M^{K+1})$ *solving for* $M_0^1 \sim \mathbb{V}_{M_0^1}$

$$\mathrm{d}M_t = FM_t\mathrm{d}t + GG^T \mathbb{E}_{\mathbb{V}_{1|t}}\left[\nabla_{m_t} \log \mathbb{S}_{1|t}^1(M_1^1|M_t)|M_t\right]\mathrm{d}t + G\mathrm{d}B_t, \quad M_0 = (M_0^1, 0_K). \quad (19)$$

Finally, we define FDBM for unpaired data translation as a stochastic process $Z^\theta$ associated with the path measure $\mathbb{P}^\theta$ solving

$$\mathrm{d}Z_t^\theta = FZ_t^\theta\mathrm{d}t + GG^T v^\theta(t, Z_t^\theta)\mathrm{d}t + G\mathrm{d}B_t, \quad Z_0^\theta = (X_0, 0, \dots, 0), \quad (20)$$

$$v_i^\theta(t, z) = [1, \omega_1\zeta_1(t, 1), \dots, \omega_K\zeta_K(t, 1)]^T \tilde{v}^\theta(t, \mu_{1|t}(z))_i, \quad v^\theta = (v_1^\theta, \dots, v_d^\theta), \quad (21)$$

where, in contrast to the paired setting in eq. (12), we do not provide the starting value $X_0$ as an input to the neural network $v_t^\theta$. We conjecture that the results of Peluchetti [33] and Shi et al. [34] generalize to our setting, such that the path measure solving the lifted SB problem in eq. (16) is the only Markovian path measure in the augmented reciprocal class $R_a(\mathbb{S})$ *and* that a solution to the lifted SB problem give in its first marginal a solution to the SB problem in eq. (15). Following Bortoli et al. [27] we define for our scaled MA-fBM reference process a flow of path measures $(\tilde{\mathbb{P}}^s, \hat{\mathbb{P}}^s)_{s \geq 0}$ recursively by

$$\hat{\mathbb{P}}^0 = (\Pi_0 \otimes \Pi_1)\mathbb{S}_{|X_0, X_1}, \quad \partial_s\hat{\mathbb{P}}^s = \mathrm{proj}_{\mathcal{R}_a(\mathbb{S})}(\mathrm{proj}_{\mathcal{M}_a(\mathbb{S})}(\hat{\mathbb{P}}^s)) - \hat{\mathbb{P}}^s, \quad \tilde{\mathbb{P}}^s = \mathrm{proj}_{\mathcal{M}_a(\mathbb{S})}(\hat{\mathbb{P}}^s), \quad (22)$$

and propose the generalized loss function

$$\mathcal{L}_{\mathrm{FDBM}}^{\mathrm{unpaired}}(\theta, \tilde{\mathbb{P}}) = \int_0^1 \int_{\mathbb{R}^{d \cdot (K+1)}} \int_{(\mathbb{R}^d)^2} \left\| \tilde{v}^\theta(t, \mu_{1|t}(z_t)) - \frac{x_1 - \mu_{1|t}(z)}{\sigma_{1|t}^2} \right\|^2 \mathrm{d}\tilde{\mathbb{P}}_{X_0, X_0}(x_0, x_1)\mathrm{d}\mathbb{S}_{t|X_0, X_1}(z_t|x_0, x_1)\mathrm{d}t. \quad (23)$$

We define $\alpha$-Iterative Markovian Fitting ($\alpha$-IMF) with respect to a scaled MA-fBM reference process using the loss function in eq. (23), following Bortoli et al. [27, Algorithm 1] with a two-stage training procedure consisting of pretraining and finetuning. As discussed in Section B.5, simulating the time reversal of eq. (20) is generally intractable, since the terminal value of the noise process depends on information from the initial distribution $\Pi_0$. We therefore adopt the forward-forward training strategy described in Bortoli et al. [27, Appendix I], and mitigate error accumulation through the loss scaling proposed in Section B.5.

We emphasize that we do not claim convergence of the resulting algorithm to the solution of the Schrödinger bridge problem in eq. (15). Empirically, we observe that the finetuning stage with an MA-fBM reference process performs reliably only in regimes close to $H = 0.5$. We hypothesize that this limitation arises from discrepancies between the Schrödinger bridge transforming $\Pi_0 \to \Pi_1$ and the Schrödinger bridge transformation $\Pi_1 \to \Pi_0$. See Section B.3 for more details on challenges and limitations of FDBM in the unpaired data setting.

| D3PM Test Set [31] | RMSD(Å) ↓ | | | % RMSD(Å) $< \tau$ ↑ | | | $\Delta$ RMSD(Å) ↑ | | |
|---|---|---|---|---|---|---|---|---|---|
| | Median | Mean | Std | $\tau = 2$ | $\tau = 5$ | $\tau = 10$ | Median | Mean | Std |
| EGNN* [31] | 19.99 | 21.37 | 8.21 | 1% | 1% | 3% | - | - | - |
| SBALIGN*$_{(10,10)}$ [31] | 3.80 | 4.98 | 3.95 | 0% | 69% | 93% | - | - | - |
| SBALIGN*$_{(100,100)}$ [31] | 3.81 | 5.02 | 3.96 | 0% | 70% | 93% | - | - | - |
| SBALIGN* [35] | 3.67 | 4.82 | 3.93 | 0% | 71% | 93% | 1.30 | 1.92 | 2.59 |
| Sesame* [35] | 2.87 | 3.65 | 2.95 | 38% | 82% | 96% | 2.15 | 3.11 | 4.26 |
| ABM [32] (retrained) | 2.40 | 3.49 | 3.54 | 43% | 84% | 96% | 2.43 | 3.35 | 4.29 |
| FDBM($H = 0.3$) (ours) | 2.33 | 3.42 | 3.42 | 43% | 85% | 97% | **2.52** | **3.49** | 4.39 |
| FDBM($H = 0.2$) (ours) | **2.12** | **3.34** | 3.59 | **48%** | **86%** | 96% | 2.44 | 3.39 | 4.28 |
| FDBM($H = 0.1$) (ours) | 2.20 | 3.44 | 3.57 | 46% | 83% | **97%** | 2.47 | 3.45 | 4.29 |

Table 1: D3PM Conformational changes, results marked with an asterisk (⋆) are obtained from the specified reference. Metrics for FDBM and ABM are averaged over 5 training trials.

# 5 Experiments

We evaluate the performance of FDBM on both **paired** and **unpaired** data translation tasks; see Section I for a detailed description of the evaluation metrics. In the paired setting, we first show in a proof-of-concept on synthetic data that the alignment of training data is preserved and then predict conformational changes in proteins. In the unpaired setting, we consider image-to-image translation across visually distinct domains. Detailed architectural specifications, compute resources, training protocols, and dataset descriptions are provided in Sections F and G, additional experiments are reported in Section K, and an additional use case on cell differentiation is presented in Section J.

## 5.1 Experiments on paired data translation

**Synthetic data**. We evaluate FDBM on the *Moons* and *T-Shape* datasets introduced by Somnath et al. [31], and depicted in Figure 7, where the goal is to transport the initial distribution (blue) to the target distribution (red) while preserving training data alignment. Quantitative performance is assessed using the Wasserstein-1 distance (WSD) between the generated and true target distributions, averaged across the two data dimensions and over ten training trials, where for each training trial 10,000 trajectories are sampled for evaluation. We first do in Table 8 an ablation on the best performing diffusion coefficient $\sqrt{\varepsilon}$ for our baseline ABM, where we find the best performance for $\sqrt{\varepsilon} = 0.8$ on Moons and $\sqrt{\varepsilon} = 0.2$ on T-shape. In Table 9 we observe that FDBM improves the quantiative performance on both datasets. For the Moons dataset, optimal performance is achieved for $\sqrt{\varepsilon} = 0.8$ in the smoother regime with $H \in \{0.6, 0.7\}$, suggesting benefits from more regular trajectories. Conversely, for the T-shape dataset, rougher dynamics with $H = 0.2$ and $\sqrt{\varepsilon} = 0.1$ yield the lowest WSD. In Figure 3, we observe that both ABM and FDBM preserve the training data alignment, with FDBM showing qualitatively better performance on T-Shape.

**Conformational changes in proteins.** Following the training and evaluation setup of Somnath et al. [31], we use their curated subset of the D3PM dataset [49] to evaluate the ability of FDBM to predict 3D ligand-bound (holo) structures from given 3D ligand-free (apo) unbound protein conformations. Performance is quantified using the root-mean-square deviation (RMSD) over carbon atom coordinates. To assess whether a predicted structure is closer to the target holo conformation than to the initial apo conformation, we further compute the $\Delta$RMSD, where positive values indicate better performance [46]. We first optimize our baseline ABM with respect to the diffusion coefficient $\sqrt{\varepsilon}$ and find that a low value of $\sqrt{\varepsilon} = 0.2$ yields the best performance for ABM (see Table 10). We then use the same training configuration and a diffusion coefficient of $\sqrt{\varepsilon} = 0.2$, to train ABM and FDBM five times and report the averaged scores over sampled trajectories from these trials. We first observe in Table 1 that ABM outperforms SBALIGN [31] across all variants and metrics, and Sesame [35] in all but one metric, highlighting the strength of our baseline. For our FDBM, we find in Table 1 that all configurations in the rough regime ($H = 0.3, 0.2, 0.1$) of MA-fBM achieve equal or better performance across all but one metric compared to the best-performing baseline, ABM. The best overall performance for the $\Delta$RMSD metric is achieved for $H = 0.3$, indicating that FDBM generated structures are closer to the target holo conformations-relative to their apo starting points-than those produced by ABM or Sesame. For $H = 0.2$ and $H = 0.3$, FDBM matches or exceeds ABM and Sesame across all evaluated metrics. In particular, an RMSD below 2Å is commonly used as a threshold for correct bound structure prediction [50] and structural discernibility [31, 35]. Accordingly, the proportion of predictions falling below this threshold is a direct indicator of the model's ability to generate physically realistic conformations. FDBM increases the proportion of correct and discernible predictions ($RMSD < 2$Å) from 43% with ABM to 48%, while also improving the median RMSD from 2.40Å to 2.12Å . This indicates that, in the rough regime of MA-fBM, FDBM produces on average a slightly higher fraction of near-native structures compared to ABM.

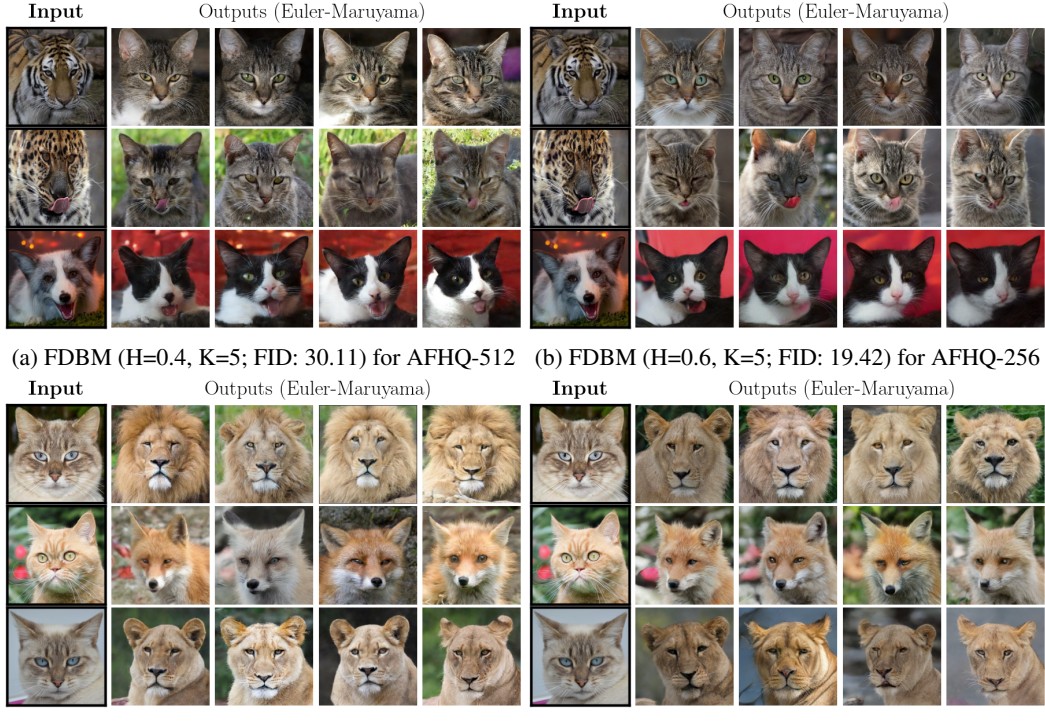

| Input | Outputs (Euler–Maruyama) | Input | Outputs (Euler–Maruyama) |

(a) FDBM (H=0.4, K=5; FID: 30.11) for AFHQ-512    (b) FDBM (H=0.6, K=5; FID: 19.42) for AFHQ-256

(c) FDBM (H=0.4, K=5; FID: 14.27) for AFHQ-512    (d) FDBM (H=0.6, K=5; FID: 11.62) for AFHQ-256

Figure 4: Exemplary FDBM samples (**ours**) for wild $\to$ cat (a, b) and cat $\to$ wild (c, d) using DiT-L/2 on AFHQ-512 and AFHQ-256. Left: inputs; right: Euler–Maruyama samples (distinct seeds).

## 5.2   Unpaired data translations

Unpaired data translation is evaluated for the cat and wild subsets of the AFHQ dataset [51]. Experiments range from low-resolution pixel space ($32 \times 32$) to high-resolution latent space settings ($256 \times 256$, $512 \times 512$) [52]. Following the regime in Bortoli et al. [27], we report Fréchet Inception Distance (FID) [53] and Learned Perceptual Image Patch Similarity (LPIPS) [54] scores. Given the sensitivity of metrics—especially at low resolutions where pixel-level perturbations dominate—each configuration is evaluated at ten distinct seeds, with mean and standard deviation (or error bands) reported. To ensure comparability, pixel data is normalized by the standard deviation of AFHQ-32, while latent representations are scaled using the standard deviation of the latent space. This harmonization enables consistent settings for $\varepsilon$ in both domains, leading to consistent performance trends (Figures 5a and 5d). We use a Diffusion Transformer (DiT) [55] backbone, where DiT-B/2 is used for ablations and DiT-L/2 for final evaluations. Pretraining is conducted for 100K steps, followed by 4K finetuning steps, samplings follow the Euler–Maruyama method [1]. We compare to SBFlow and adopt an SBFlow-optimized entropic regularization parameter for FDBM experiments. Further, we evaluate Hurst indices $H \in \{0.1, 0.2, \ldots, 0.9\}$ and the number of OU processes $K \in \{1, \ldots, 6\}$ to analyze sensitivity in sparse (AFHQ-32) and dense (AFHQ-256 and AFHQ-512) features.

**Results for unpaired data translation**. The ablation study reveals stable generation performance for $H \geq 0.4$ and $K \leq 5$, with instabilities and accuracy degradation observed for $K > 5$ and $H < 0.3$, see Figures 5b, 5c, 5e and 5f. Our method remains stable for high dimensional data, such as AFHQ-512 even for $0.4 \leq H < 0.5$ (see Figure 4). Across various configurations, our method consistently outperforms the SBFlow pretraining- and online finetuning baseline (see Table 2, as well as Figure 5). Notably, with $K = 5$ we do not recover BM, as we fix $\gamma_1, \ldots, \gamma_K$, even when $H = 0.5$. MA-fBM with $H = 0.5$ and $K = 5$ is non-Markovian, though its distribution is empirically close to BM. This subtle differences may be the reason why FDBM performs better than SBFLow on AFHQ when $H = 0.5$ and $K = 5$. Bortoli et al. [27] propose a finetuning method for processes driven by BM, which can yield significant improvements over their proposed pretraining for natural images. The online finetuning assumes the bidirectional processes to transition on the same bridge with matching pairings and respective terminal distributions. In our framework, we can not assume a shared Schrödinger bridge for the transformation $\Pi_0 \to \Pi_1$ and $\Pi_1 \to \Pi_0$. In general, two distinct bridges are learned. Improvements during fine-tuning were observed only for MA-fBM

Table 2: Results for AFHQ-32 and AFHQ-256 (10 runs average). Standard deviations are reported beside each score. Bold indicates the best result and those within one standard deviation.

(a) AFHQ-32 results with hyperparameters $\varepsilon = 1$ and $H = 0.5, K = 5$.

| Method | Architecture | Pretraining | | | | Online Finetuning | | | |
|---|---|---|---|---|---|---|---|---|---|
| | | cats $\rightarrow$ wild | | cats $\leftarrow$ wild | | cats $\rightarrow$ wild | | cats $\leftarrow$ wild | |
| | | FID $\downarrow$ | LPIPS $\downarrow$ | FID $\downarrow$ | LPIPS $\downarrow$ | FID $\downarrow$ | LPIPS $\downarrow$ | FID $\downarrow$ | LPIPS $\downarrow$ |
| SBFlow | DiT-B/2 | 59.04 ±1.14 | 0.104 ±0.001 | 74.36 ±1.02 | **0.151** ±0.001 | 43.85 ±0.48 | 0.083 ±0.001 | 64.77 ±0.78 | 0.107 ±0.000 |
| SBFlow | DiT-L/2 | 50.68 ±0.72 | 0.106 ±0.001 | 71.77 ±0.77 | 0.152 ±0.001 | 33.92 ±0.59 | 0.091 ±0.000 | 54.10 ±0.72 | 0.098 ±0.001 |
| FDBM (**ours**) | DiT-B/2 | 40.21 ±1.18 | **0.097** ±0.001 | 45.74 ±0.69 | 0.154 ±0.002 | 25.66 ±0.81 | **0.073** ±0.001 | 28.33 ±0.35 | **0.078** ±0.001 |
| FDBM (**ours**) | DiT-L/2 | **35.99** ±0.72 | 0.101 ±0.001 | 48.84 ±0.75 | 0.165 ±0.002 | **20.26** ±0.59 | 0.079 ±0.001 | **26.79** ±0.50 | 0.085 ±0.001 |

(b) AFHQ-256 results with hyperparameters $\varepsilon = 1$ and $H = 0.6, K = 5$.

| Method | Architecture | Pretraining | | | | Online Finetuning | | | |
|---|---|---|---|---|---|---|---|---|---|
| | | cats $\rightarrow$ wild | | cats $\leftarrow$ wild | | cats $\rightarrow$ wild | | cats $\leftarrow$ wild | |
| | | FID $\downarrow$ | LPIPS $\downarrow$ | FID $\downarrow$ | LPIPS $\downarrow$ | FID $\downarrow$ | LPIPS $\downarrow$ | FID $\downarrow$ | LPIPS $\downarrow$ |
| SBFlow | DiT-B/2 | 15.67 ±0.65 | 0.578 ±0.002 | 30.75 ±0.88 | 0.594 ±0.001 | 17.50 ±0.87 | **0.528** ±0.001 | **25.86** ±0.32 | **0.537** ±0.001 |
| SBFlow | DiT-L/2 | 16.62 ±0.83 | 0.604 ±0.001 | 33.96 ±0.87 | 0.600 ±0.001 | **16.98** ±0.53 | 0.560 ±0.001 | 27.82 ±0.41 | 0.547 ±0.001 |
| FDBM (**ours**) | DiT-B/2 | 16.77 ±0.71 | **0.530** ±0.002 | 19.14 ±0.38 | **0.551** ±0.001 | – | – | – | – |
| FDBM (**ours**) | DiT-L/2 | **11.62** ±0.73 | 0.548 ±0.002 | 19.42 ±0.41 | 0.561 ±0.002 | – | – | – | – |

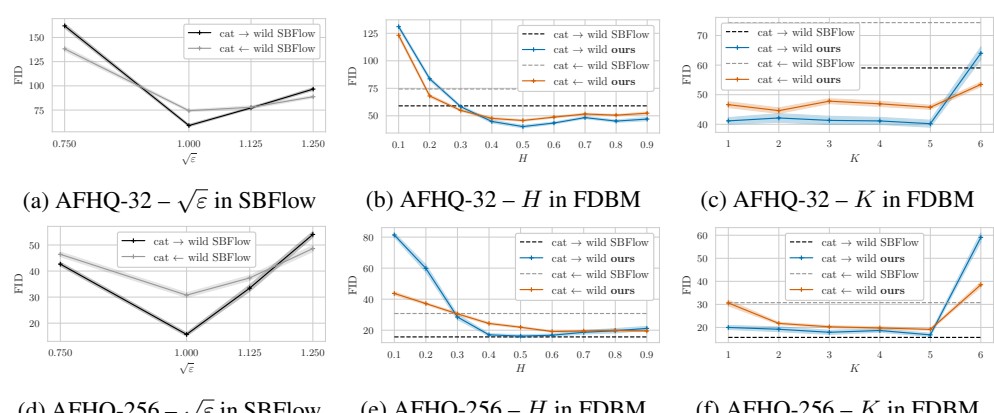

(a) AFHQ-32 – $\sqrt{\varepsilon}$ in SBFlow  (b) AFHQ-32 – $H$ in FDBM  (c) AFHQ-32 – $K$ in FDBM

(d) AFHQ-256 – $\sqrt{\varepsilon}$ in SBFlow  (e) AFHQ-256 – $H$ in FDBM  (f) AFHQ-256 – $K$ in FDBM

Figure 5: DiT-B/2 ablation for AFHQ-32 (a-c) and AFHQ-256 (d-f); (c): $H = 0.5$, and (f): $H = 0.6$. We show error bands with averages over 10 runs. SBFlow baselines are marked in (b, c, e), and (f).

with $H = 0.5$, likely because the forward ($\Pi_0 \rightarrow \Pi_1$) and backward bridge ($\Pi_1 \rightarrow \Pi_0$) are close—though not identical—to the Brownian case with a bidirectional bridge. However, this effect does not generalize to other $H$, and developing a principled finetuning strategy for FDBMs distinct bridges remains an important direction for future work. Table 2 shows that our method can significantly improve the fidelity of generated samples, while maintaining data alignment. Figure 4 highlights that we can obtain cohesive data alignment without online finetuning for $H = 0.4$ and $H = 0.6$ at scale. See Section K and in particular Figures 5 and 6 for samplings at scale.

## 6 Conclusion

We introduced Fractional Diffusion Bridge Models (FDBM), a new generative framework that extends diffusion bridges beyond the Markovian assumptions by incorporating a Markovian approximate fractional Brownian motion to retain computational tractability while preserving long-range dependencies or roughness that are absent in Brownian generative models. Our fractional generative diffusion bridge is coupling-preserving in the paired case and generalizes the Schrödinger bridge formulation for unpaired settings. In the paired regime, FDBM improved the near-native structures of predicted protein conformations potentially by capturing non-local dependencies; in the unpaired regime, it achieved superior quality in image translation scaling robustly across high-dimensional domains and image resolutions.

FDBM opens a broader avenue for generative modeling, bridging fractional stochastic dynamics and machine learning, and poses a foundation for learning from the correlated, memory-rich phenomena in real-world. Future work includes theoretical guarantees for fractional Schrödinger bridges, finetuning of asymmetric bridges, and extensions to manifold-valued fractional processes.

## Acknowledgments and Disclosure of Funding

This research received funding from the Flemish Government under the 'Onderzoeksprogramma Artificiële Intelligentie (AI) Vlaanderen' programme. Furthermore it was supported by Flanders Make under the SBO project CADAIVISION. This work also received funding from imec-PROSPECT project ADAPT ('Affinity and Developability through AI for Protein Therapeutics'). This work was also supported by the German Research Foundation (DFG) through research unit DeSBi [KI-FOR 5363] (project ID: 459422098). T. Birdal acknowledges support from the Engineering and Physical Sciences Research Council [grant EP/X011364/1]. T. Birdal was supported by a UKRI Future Leaders Fellowship [grant number MR/Y018818/1] as well as a Royal Society Research Grant RG/R1/241402.

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

# Appendix of Fractional Diffusion Bridge Models

# A    Notational conventions

| | |
|---|---|
| $\mathbb{R}^m$ | $m \in \mathbb{N}$ dimensional Euclidean space |
| $\mathcal{B}(\mathbb{R}^m)$ | Borel-$\sigma$-algebra on $\mathbb{R}^m$ |
| $S = (S_t)_{t \in [0,T]}$ | Stochastic process taking values in $\mathbb{R}^m$ |
| $\mathcal{C}^m$ | Set of continuous functions (paths) $\mathcal{C}^m = C([0,1], \mathbb{R}^m)$ from the unit time interval $[0,1]$ to $\mathbb{R}^m$ |
| $\mathcal{B}(\mathcal{C}^m)$ | Borel-$\sigma$-algebra on $\mathcal{C}^m$ |
| $\mathcal{P}(\mathcal{C}^m)$ | Set of probability measures on $(\mathcal{C}^m, \mathcal{B}(\mathcal{C}^m))$ |
| $\mathbb{P} \in \mathcal{P}(\mathcal{C}^m)$ | Path measure |
| $\mathbb{P}_{S_{t_1}, \dots S_{t_n}}$ | Distribution of $(S_{t_1}, \dots S_{t_n})$ under the path measure $\mathbb{P}$ |
| $\mathbb{P}_{t_1, \dots t_n}$ | Path measure $\mathbb{P}$ is associated with a process $S$ and $\mathbb{P}_{t_1, \dots t_N}$ denotes the distribution of $(S_{t_1}, \dots S_{t_n})$ |
| $\mathbb{P}_{t_1, \dots t_n \mid r_1, \dots r_l}$ | Conditional distribution of $(S_{t_1}, \dots S_{t_n})$ given $(S_{r_1}, \dots S_{r_l})$ |
| $\mathbb{P}_{\mid r_1, \dots r_l}$ | Conditional distribution of $S$ given $(S_{r_1}, \dots S_{r_l})$ |
| $d \in \mathbb{N}$ | Data dimension |
| $\Pi_0, \Pi_1$ | Source and target distribution on $\mathbb{R}^d$ |
| $\Pi_{0,1}$ | Joint (coupling) distribution on $\mathbb{R}^d \times \mathbb{R}^d$ |
| $B$ | (Multidimensional) standard Brownian motion |
| $B^H$ | (Multidimensional) Riemann-Liouville (Type II) fractional Brownian motion (fBM) |
| $Y^\gamma$ | (Multidimensional) OrnsteinUhlenbeck (OU) process with speed of mean reversion $\gamma \in \mathbb{R}$ |
| $K, k \in \mathbb{N}$ | Number of augmenting processes $K$ and $1 \leq k \leq K$ |
| $\gamma_1, \dots, \gamma_K$ | Geometrically spaced grid |
| $\omega_1, \dots, \omega_K$ | Approximation coefficients |
| $\hat{B}^H$ | (Multidimensional) Markov-approximate fractional Brownian motion (MA-fBM) |
| $X$ | Scaled MA-fBM reference process $X = \sqrt{\varepsilon}\hat{B}^H$ with $\varepsilon > 0$ |
| $\mathbb{Q}$ | Path measure of the reference process |
| $\Pi_{0,1}\mathbb{Q}_{\mid 0,1}$ | Mixture of bridge measures $\int_{\mathbb{R}^d \times \mathbb{R}^d} \mathbb{Q}_{\mid 0,1}(\cdot \mid x_0, x_1) \mathrm{d}\Pi_{0,1}(x_0, x_1)$ |
| $Y^k$ | Augmenting process $Y^k = Y^{\gamma_k}$ |
| $Y$ | Stacked augmenting OU processes $Y = (Y^1, \dots, Y^K)$ taking values in $\mathbb{R}^{dK}$ |
| $Z$ | Augmented process $Z = (X, Y)$ on $\mathbb{R}^{d \cdot (K+1)}$ |
| $\mathbb{S}$ | Path measure of the augmented process $Z$ |
| $F$ | Drift matrix $F \in \mathbb{R}^{d(K+1), d(K+1)}$ of the augmented forward process |
| $G$ | Diffusion vector $G \in \mathbb{R}^{d(K+1)}$ of the augmented forward process |
| $Z_{\mid X_0, X_1}$ | Partially pinned process $Z \mid (X_0, X_1)$ |
| $\mathbb{S}_{\mid X_0, X_1}$ | Path measure associated with the partially pinned process $Z_{\mid X_0, X_1}$ |
| $\mathbb{T}^{SB}$ | Solution to the dynamic Schrödinger bridge problem |
| $\mathbb{V}$ | Path measure on the augmented path space $\mathcal{P}(\mathcal{C}^{d(K+1)})$ |

## B   Mathematical framework of fractional diffusion bridge models

In this section, we present the mathematical details of Fractional Diffusion Bridge Models (FDBM). The main contribution of this section is the proof of Theorem 5 in Section B.2, which generalizes the construction of a coupling-preserving stochastic process by Bortoli et al. [32] to our fractional noise setting.

**Notation**. For any $m \in \mathbb{N}$, we equip the Euclidean space $\mathbb{R}^m$ with its Borel-$\sigma$-algebra $\mathcal{B}(\mathbb{R}^m)$. Below, $m$ will typically be either equal to $1$, $d$, $dK$, or $d(K+1)$. Next, we write $\mathcal{C}^m = C([0,1], \mathbb{R}^m)$ for the set of continuous functions (or continuous "paths") from the unit time interval $[0,1]$ to $\mathbb{R}^m$ and equip this set with its Borel-$\sigma$-algebra $\mathcal{B}(\mathcal{C}^m)$ where open sets are understood with respect to the topology of uniform convergence. The set of probability measures on $(\mathcal{C}^m, \mathcal{B}(\mathcal{C}^m))$ is denoted by $\mathcal{P}(\mathcal{C}^m)$, and we refer to the elements of this set as path measures. If $X$ is a stochastic process and $\mathbb{P} \in \mathcal{P}(\mathcal{C}^m)$ denotes the distribution of $X$, we subsequently say that the path measure $\mathbb{P}$ is associated with the process $X$. Observe that any $\mathbb{P} \in \mathcal{P}(\mathcal{C}^m)$ is associated with some stochastic process $X$, as we may take the space $(\mathcal{C}^m, \mathcal{B}(\mathcal{C}^m), \mathbb{P})$ as our probability space and let $X$ be the canonical process given by

$$X_t(\omega) := \omega_t, \qquad t \in [0,1], \omega \in \mathcal{C}^m. \tag{24}$$

Given a path measure $\mathbb{P} \in \mathcal{P}(\mathcal{C}^m)$ associated with a process $X$ and time points $t_1, \ldots, t_n \in [0,1]$ for some $n \in \mathbb{N}$, we write $\mathbb{P}_{t_1,\ldots,t_n}$ for the joint distribution of $(X_{t_1}, \ldots, X_{t_n})$, that is

$$\mathbb{P}_{t_1,\ldots,t_n} := \mathbb{P}(\{\omega \in \mathcal{C}^m : (\omega(t_1), \ldots, \omega(t_n)) \in \cdot \}). \tag{25}$$

In particular, $\mathbb{P}_t$ denotes the marginal distribution of $X(t)$ for any $t \in [0,1]$. Moreover, given $s_1, \ldots, s_\ell \in [0,1]$ and $x_{s_1}, \ldots, x_{s_\ell} \in \mathbb{R}^m$, we write $\mathbb{P}_{t_1,\ldots,t_n|s_1,\ldots,s_\ell}$ and $\mathbb{P}_{t_1,\ldots,t_n|s_1,\ldots,s_\ell}(\cdot | x_{s_1}, \ldots, x_{s_\ell})$ for the (regular) conditional distribution of $(X_{t_1}, \ldots, X_{t_n})$ given $(X_{s_1}, \ldots, X_{s_\ell})$ and $\{X_{s_1} = x_{s_1}, \ldots, X_{s_\ell} = x_{s_\ell}\}$, respectively. In the same spirit, we write $\mathbb{P}_{|s_1,\ldots,s_\ell}$ and $\mathbb{P}_{|s_1,\ldots,s_\ell}(\cdot | x_{s_1,\ldots,x_{s_\ell}})$ for the (regular) conditional distribution of the process $X$ given $(X_{s_1}, \ldots, X_{s_\ell})$ and $\{X_{s_1} = x_{s_1}, \ldots, X_{s_\ell} = x_{s_\ell}\}$, respectively.

### B.1   A Markov approximate fractional Brownian bridge.

We fix a $d$-dimensional Brownian motion $B$ and define the Riemann–Liouville (Type II) fractional Brownian motion (fBM) [22] with Hurst index $H \in (0,1)$ via

$$B_t^H := \frac{1}{\Gamma(H+\frac{1}{2})} \int_0^t (t-s)^{H-\frac{1}{2}} \mathrm{d}B_s, \quad t \geq 0. \tag{26}$$

For a given Hurst index $H \in (0,1)$, we consider a Markovian approximation of fBM [25, 26]. For $K \in \mathbb{N}$ and geometrically-spaced speed of mean reversion parameters $\gamma_1, \ldots, \gamma_K > 0$, we consider Ornstein–Uhlenbeck (OU) processes of the form

$$Y_t^k := \int_0^t e^{-\gamma_k(t-s)} \mathrm{d}B_s, \qquad t \in [0,1], \ k = 1, \ldots, K. \tag{27}$$

With this, for a given scaling parameter $\varepsilon > 0$ and suitably chosen approximation weights $\omega_1, \ldots, \omega_K \in \mathbb{R}$, the process $X := \sqrt{\varepsilon}\hat{B}^H$ defined in terms of the weighted superposition

$$\hat{B}^H := \sum_{k=1}^K \omega_k Y^k \tag{28}$$

of the OU processes is a scaled Markovian approximation of fBM (MA-fBM). While the choice of approximation coefficients in Harms [43] enables strong convergence to fBM with high polynomial order in $K$ for $H < 0.5$, we opt for the computationally more efficient method proposed by Daems et al. [26]. This method selects the $L^2(\mathbb{P})$ optimal approximation coefficients for a given Hurst index $H \in (0,1)$ and a given $K \in \mathbb{N}$ by minimizing

$$(\omega_1, \ldots, \omega_K) = \underset{\omega_1,\ldots,\omega_K}{\arg\min} \left\{ \int_0^T \mathbb{E}\left[ \left( B_t^H - \hat{B}_t^H \right)^2 \right] \mathrm{d}t \right\}. \tag{29}$$

Following [26, Proposition 5], the so defined optimal approximation coefficients $\omega = (\omega_1, \ldots, \omega_K)$ solve the system $A\omega = b$, where $A$ and $b$ are given in closed form [26, eq. (19), eq. (21)] and hence we choose $\omega := A^{-1}b$. Note that these optimal approximation coefficients depend on the Hurst index $H \in (0,1)$ and the number of OU processes $K \in \mathbb{N}$, since the matrix $A$ and the vector $b$ are functions of these parameters. We subsequently refer to $X = \sqrt{\varepsilon}\hat{B}^H$ as the reference process, $Y := (Y^1, \ldots, Y^K)$ as the vector of OU processes, and $Z := (X, Y)$ as the augmented reference process, respectively. Note that the dynamics of the augmented reference process are given by [16, 26]

$$\mathrm{d}Z_t = FZ_t\mathrm{d}t + G\mathrm{d}B_t \tag{30}$$

for a matrix $F \in \mathbb{R}^{d(K+1),d(K+1)}$ and a vector $G \in \mathbb{R}^{d(K+1)}$ [16, 26]. The path measure associated with the reference process $X$ is denoted by $\mathbb{Q} \in \mathcal{P}(\mathcal{C}^d)$, whereas the path measure associated with the augmented reference process is denoted by $\mathbb{S} \in \mathcal{P}(\mathcal{C}^{d(K+1)})$ and we write $\mathbb{S}^1_{1|t}$ for the conditional distribution of $X_1|Z_t$. Note that the reference process $X$, as well as its corresponding path measure $\mathbb{Q}$ is non-Markovian and becomes Markovian only after augmenting it with the OU processes, resulting in the Markovian augmented reference process $Z$.

**Proposition 8.** *The reference process $X$ is for $K > 1$ and all $H \in (0,1)$ the non-Markovian process*

$$X_{t+s} = X_t + \sum_{k=1}^{K} \omega_k \zeta_k(t, t+s)Y_t^k + \sqrt{\varepsilon}\sum_{k=1}^{K}\omega_k \int_t^{t+s} e^{-\gamma_k(t+s-u)}\mathrm{d}B_u, \tag{31}$$

*where for each $k = 1, \ldots, K$, and $t, s \in [0,1]$ with $t+s \leq 1$*

$$\zeta_k(t, t+s) := -\sqrt{\varepsilon}\gamma_k \int_t^{t+s} e^{-\gamma_k(u-t)}\mathrm{d}u = \sqrt{\varepsilon}(e^{\gamma_k s} - 1). \tag{32}$$

*In particular, we see that $\mathbb{S}^1_{t+s|t}(\cdot\,|z)$ is Gaussian and hence for $d = 1$, with $s = 1 - t$,*

$$\nabla_z \log \mathbb{S}^1_{1|t}(x_1|z) = [1, \omega_1\zeta_1(t, 1), \ldots, \omega_K\zeta_K(t, 1)]^T \frac{x_1 - \mu_{1|t}(z)}{\sigma^2_{1|t}}, \tag{33}$$

*where*

$$\mu_{1|t}(z) := x + \sum_k \omega_k y_k \zeta_k(t, 1), \quad z = (x, y_1, .., y_K) \tag{34}$$

*denotes the conditional mean and*

$$\sigma^2_{1|t} := \varepsilon \sum_{k,\ell=1}^{K} \frac{\omega_k \omega_\ell}{\gamma_k + \gamma_\ell}\left(1 - e^{-(1-t)(\gamma_k+\gamma_\ell)}\right) \tag{35}$$

*the conditional variance of the reference process $X_1|(Z_t = z)$.*

*Proof.* For the scaled MA-fBM we have for $s, t \in [0,1]$ with $t+s \leq 1$ by the Stochastic Fubini Theorem [25]

$$X_{t+s} = -\sum_{k=1}^{K}\omega_k\gamma_k\int_0^t Y_r^k dr + \sum_{k=1}^{K}\omega_k B_t - \sum_{k=1}^{K}\omega_k\gamma_k\int_t^{t+s} Y_r^k dr + \sum_{k=1}^{K}\omega_k(B_{t+s} - B_t) \tag{36}$$

$$= \hat{B}_t^H - \sum_{k=1}^{K}\omega_k\gamma_k\int_t^{t+s}[e^{-\gamma_k(r-t)}Y_t^k + \int_t^r e^{-\gamma_k(r-u)}dB_u]dr + \sum_{k=1}^{K}\omega_k(B_{t+s} - B_t) \tag{37}$$

$$= \hat{B}_t^H + \sum_{k=1}^{K}\omega_k\left[Y_t^k(e^{-\gamma_k s} - 1) + (B_{t+s} - B_t) - \gamma_k\int_t^{t+s}\int_t^r e^{-\gamma_k(r-u)}dB_u dr\right] \tag{38}$$

$$= \hat{B}_t^H + \sum_{k=1}^{K}\omega_k\left[Y_t^k(e^{-\gamma_k s} - 1) + (B_{t+s} - B_t) - \gamma_k\int_t^{t+s}\int_u^{t+s} e^{-\gamma_k(r-u)}dr dB_u\right] \tag{39}$$

$$= \hat{B}_t^H + \sum_{k=1}^{K}\omega_k Y_t^k(e^{-\gamma_k s} - 1) + \sum_{k=1}^{K}\omega_k\int_t^{t+s}[1 - \gamma_k\int_u^{t+s} e^{-\gamma_k(r-u)}dr]dB_u \tag{40}$$

$$= X_t + \sum_{k=1}^{K}\omega_k\zeta_k(t, t+s)Y_t^k + \sqrt{\varepsilon}\sum_{k=1}^{K}\omega_k\int_t^{t+s} e^{-\gamma_k(t+s-u)}\mathrm{d}B_u. \tag{41}$$

Additionally, we calculate via eq. (41) with $s = 1 - t$ and $z = (x, y_1, .., y_K)$ the conditional mean

$$\mu_{1|t}(z) := \mathbb{E}\left[X_1 | Z_t = z_t\right] = x + \sum_k \omega_k y_k \zeta_k(t, 1), \quad z = (x, y_1, .., y_K) \tag{42}$$

and the conditional variance

$$\sigma_{1|t}^2 := \text{Cov}\left(X_1, X_1 | Z_t = z\right) = \varepsilon \sum_{k,\ell=1}^K \frac{\omega_k \omega_\ell}{\gamma_k + \gamma_\ell}\left(1 - e^{-(1-t)(\gamma_k + \gamma_\ell)}\right), \tag{43}$$

where we use Itô's isometry. To see that $X$ is non-Markovian, we note that the future $X_{t+s}$ depends not only on $X_t$ but also on $Y_t^1, ..., Y_t^K$ which depend on the path of $B$ up to time $t$. For a more precise argument, we have by definition

$$\hat{B}_t^H = \int_0^t \sum_{k=1}^K \omega_k e^{-\gamma_k(t-u)} \mathrm{d}B_u \tag{44}$$

and note that a process $\hat{X} = (\hat{X}_t)_{t \in [0,1]}$ with

$$\hat{X}_t = \int_{-\infty}^t \kappa(t, u) dB_u \tag{45}$$

is a Markov process, if and only if we can find functions $f$ and $g$ such that [56, Theorem II.1]

$$\kappa(t, u) = f(t)g(u). \tag{46}$$

Since we have $\gamma_1 \neq \gamma_2 \neq \cdots \neq \gamma_K$ for the defined MA-fBM, functions $f$ and $g$ satisfying

$$\sqrt{\varepsilon} \sum_{k=}^K \omega_k e^{-\gamma_k(t-u)} = f(t)g(u) \tag{47}$$

exist for $K > 1$ if and only if $\omega_j \neq 0$ for at most one $1 \leq j \leq K$. Hence, MA-fBM—and therefore our reference process $X$—is not a Markov-process for $K > 1$ and any choice of $H \in (0, 1)$. $\qquad\square$

To define a stochastic bridge with respect to $X$ connecting two given points $x_0 \in \mathbb{R}^d$ and $x_1 \in \mathbb{R}^d$, observe that we only have to steer the first dimension of the augmented reference process $Z = (X, X)$ towards $x_1$, while the terminal values $Y$ are not required to attain a specific value.

**Proposition 9** (Markov approximation of a fractional Brownian bridge [45]). *Let $X = \sqrt{\varepsilon}\hat{B}^H$ be a scaled MA-fBM, $\varepsilon > 0$ and $Z = (X, Y)$ the augmented reference process. The partially pinned process $Z_{|x_0,x_1} := Z|(X_0 = x_0, X_1 = x_1)$ associated to the path measure $\mathbb{S}_{|x_0,x_1}$ follows the dynamics*

$$\mathrm{d}Z_{|x_0,x_1}(t) = F Z_{|x_0,x_1}(t)\mathrm{d}t + GG^T u(t, Z_{|x_0,x_1}(t))\mathrm{d}t + G\mathrm{d}B_t, \tag{48}$$

$$u_i(t, z) = [1, \omega_1 \zeta_1(t, 1), ..., \omega_K \zeta_K(t, 1)]^T \left[\frac{x_1 - \mu_{1|t}(z)}{\sigma_{1|t}^2}\right], \quad u = (u_1, \ldots, u_d) \tag{49}$$

*Proof.* Daems et al. [45] use a Gaussian expression for the reference process to construct the posterior SDE that is steered towards $x_1$. We derive for a fixed data pair $(x_0, x_1)$ the dynamics of the partially pinned process $Z_{|x_0,x_1} = Z|(X_1 = x_0, X_1 = x_1)$ using Doob's $h$-transform [2], resulting in the same dynamics as in Daems et al. [45]. Towards that goal, we define the transform

$$h : [0, 1] \times \mathbb{R}^{d(K+1)} \to [0, 1], \quad (t, z) \mapsto \mathbb{S}_{1|t}^1(x_1|z), \tag{50}$$

where $\mathbb{S}_{1|t}^1$ satisfies

$$\mathbb{P}(X_1 \in A | Z_t = z) = \int_A \mathbb{S}_{1|t}^1(x|z)dx, \quad A \subset \mathbb{R}^d. \tag{51}$$

Denote by $\mathbb{S}_t(z) = \mathbb{S}_t(x, y)$ the density of $Z_t$ such that

$$\mathbb{S}_1^1(x) = \int_{\mathbb{R}^{dK}} \mathbb{S}_1(x, y)\mathrm{d}y \tag{52}$$

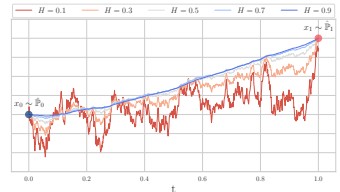 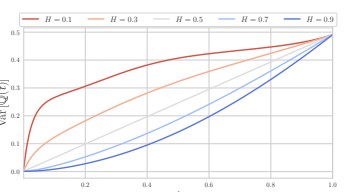 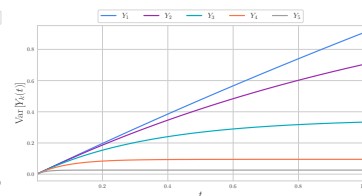

(a) Trajectories from the approximate $1d$-fractional Brownian bridge for different Hurst indices.

(b) Variance of MA-fBM with normalized terminal variance.

(c) Variance of the augmenting OU processes $Y_1, ..., Y_5$ approximating fBM as a weighted sum.

Figure 6: Evolution of variance in MA-fBM.

and write $\mathbb{S}_{t+s|s}(\tilde{z}|z)$ for the transition density of $Z$ from time $t$ to $t+s$. To show that $h$ defined in eq. (50) satisfies the space-time regularity property we mimic the proof of [2, Theorem 7.11]. We write with Bayes rule

$$\mathbb{S}_{t+s|t,x_1}(\tilde{z}|z,x_1) = \frac{\mathbb{S}^1_{1|t+s,t}(x_1|\tilde{z},z)\mathbb{S}_{t+s|t}(\tilde{z}|z)}{\mathbb{S}^1_{1|t}(x_1|z)} = \frac{\mathbb{S}^1_{1|t+s}(x_1|\tilde{z})\mathbb{S}_{t+s|t}(\tilde{z}|z)}{\mathbb{S}^1_{1|t}(x_1|z)} \tag{53}$$

where we use for the second equation that $Z$ is a Markov process. Hence, equivalently

$$\mathbb{S}_{t+s|t}(\tilde{z}|z)\mathbb{S}^1_{1|t+s}(x_1|\tilde{z}) = \mathbb{S}_{t+s|t,x_1}(\tilde{z}|z,x_1)\mathbb{S}^1_{1|t}(x_1|z) \tag{54}$$

such that

$$\int_{\mathbb{R}^{d(K+1)}} \mathbb{S}_{t+s|s}(\tilde{z}|z)h(t+s,z)\mathrm{d}\tilde{z} = \int_{\mathbb{R}^{d(K+1)}} \mathbb{S}_{t+s|s}(\tilde{z}|z)\mathbb{S}^1_{1|t}(x_1|z)\mathrm{d}\tilde{z} \tag{55}$$

$$= \int_{\mathbb{R}^{d(K+1)}} \mathbb{S}_{t+s|t,x_1}(\tilde{z}|z,x_1)\mathbb{S}^1_{1|t}(x_1|z)\mathrm{d}\tilde{z} \tag{56}$$

$$= \mathbb{S}^1_{1|t}(x_1|z)\underbrace{\int_{\mathbb{R}^{d(K+1)}} \mathbb{S}_{t+s|t,x_1}(\tilde{z}|z,x_1)\mathrm{d}\tilde{z}}_{=1} \tag{57}$$

$$= h(t,z) \tag{58}$$

Hence, by Särkkä and Solin [2, eq. (7.73) - eq. (7.78)], we conclude that the partially pinned process $Z_{|x_0,x_1}$ satisfies

$$\mathrm{d}Z_{|x_0,x_1}(t) = FZ_{|x_0,x_1}(t)\mathrm{d}t + GG^T\nabla_z \log \mathbb{S}^1_{1|t}(x_1|Z_{|x_0,x_1}(t))\mathrm{d}t + G\mathrm{d}B_t. \tag{59}$$

Moreover, from eq. (33), we obtain $\nabla_z \log \mathbb{S}^1_{1|t} = \left( \left[\nabla_z \log \mathbb{S}^1_{1|t}\right]_1, \ldots, \left[\nabla_z \log \mathbb{S}^1_{1|t}\right]_d \right)$ with

$$\left[\nabla_z \log \mathbb{S}^1_{1|t}\right]_i (x_1|z) = [1, \omega_1\zeta_1(t,1), \ldots, \omega_K\zeta_K(t,1)]^T \left[\frac{x_1 - \mu_{1|t}(z)}{\sigma^2_{1|t}}\right]_i =: u(t,z)_i. \tag{60}$$

$\square$

See Figure 6 for a visualization of $1d$-trajectories and Figure 1 for $2d$-trajecotires of the above defined Markov approximate fractional Brownian bridge (MA-fBB).

## B.2 Theoretical framework for paired training data

Fix a probability measure $\Pi_{0,1}$ on $\mathbb{R}^d \times \mathbb{R}^d$, which we refer to as the coupling measure. The marginals of this measure are denoted by $\Pi_0$ and $\Pi_1$, respectively, which means that

$$\Pi_0(A) := \int_{A \times \mathbb{R}^d} \mathrm{d}\Pi_{0,1}(x_0, x_1) \quad \text{and} \quad \Pi_1(A) := \int_{\mathbb{R}^d \times A} \mathrm{d}\Pi_{0,1}(x_0, x_1), \qquad A \in \mathcal{B}(\mathbb{R}^d).$$

Our goal is to construct a stochastic process $X^\star$ that preserves the coupling in the sense that $(X^\star_0, X^\star_1) \sim \Pi_{0,1}$, and that $X^\star$ solves a stochastic differential equation (SDE). If that is achieved,

we can sample from the coupling $\Pi_{0,1}$ by first sampling $X_0^\star = x_0 \sim \Pi_0$ according to the first marginal of $\Pi_{0,1}$, and then simulating the SDE forward in time on $[0,1]$ to arrive at a sample $X_1^\star = x_1$. As $X^\star$ preserves the coupling, it follows that $(x_0, x_1)$ is drawn from $\Pi_{0,1}$. Recall that $\mathbb{Q}_{|0,1}(\cdot \,|x_0, x_1) \in \mathcal{P}(\mathcal{C}^d)$ denotes the path measure of the reference process $X$ conditioned on $(X_0, X_1) = (x_0, x_1) \in \mathbb{R}^d \times \mathbb{R}^d$. We define a new path measure $\mathbb{P} \in \mathcal{P}(\mathcal{C}^d)$ by integrating $(x_0, x_1)$ with respect to $\Pi_{0,1}$, that is

$$\mathbb{P} := \int_{\mathbb{R}^d \times \mathbb{R}^d} \mathbb{Q}_{|0,1}(\cdot \,|x_0, x_1) \mathrm{d}\Pi_{0,1}(x_0, x_1). \tag{61}$$

To wit, the process $X^\star$ associated with $\mathbb{P}$ is the reference process conditioned on $(X_0^*, X_1^*) \sim \Pi_{0,1}$. Indeed, this is seen immediately as for any Borel sets $A_0, A_1 \subset \mathbb{R}^d$ we have

$$\mathbb{P}(\{\omega \in \mathcal{C}^d : \omega(0) \in A_0, \omega(1) \in A_1\})$$

$$= \int_{\mathbb{R}^d \times \mathbb{R}^d} \mathbb{Q}_{|0,1}(\{\omega \in \mathcal{C}^d : \omega(0) \in A_0, \omega(1) \in A_1\}|x_0, x_1) \mathrm{d}\Pi_{0,1}(x_0, x_1) \tag{62}$$

$$= \int_{\mathbb{R}^d \times \mathbb{R}^d} \mathbb{1}_{A_0}(x_0) \mathbb{1}_{A_1}(x_1) \mathrm{d}\Pi_{0,1}(x_0, x_1) \tag{63}$$

$$= \Pi_{0,1}(A_0 \times A_1). \tag{64}$$

A key assumption for establishing the existence of an SDE whose solution $X^\star$ has distribution $\mathbb{P}$ is that $\mathbb{P}$ is absolutely continuous with respect to $\mathbb{Q}$.

**Assumption 1.** *The path measure $\mathbb{P} \in \mathcal{P}(\mathcal{C}^d)$ is absolutely continuous with respect to the path measure $\mathbb{Q} \in \mathcal{P}(\mathcal{C}^d)$ of the reference process $X$. In particular, there exists a density*

$$\frac{\mathrm{d}\mathbb{P}}{\mathrm{d}\mathbb{Q}} : \mathcal{C}^d \to [0, \infty). \tag{65}$$

The density $\mathrm{d}\mathbb{P}/\mathrm{d}\mathbb{Q}$ allows us to lift the measure $\mathbb{P}$ to a path measure $\mathbb{P}^\star$ on the augmented path space $\mathcal{C}^{d(K+1)}$ via the Radon–Nikodým density

$$\frac{\mathrm{d}\mathbb{P}^\star}{\mathrm{d}\mathbb{S}}(\omega) := \frac{\mathrm{d}\mathbb{P}}{\mathrm{d}\mathbb{Q}}, \qquad \omega = (\omega_X, \omega_Y) \in \mathcal{C}^{d(K+1)}. \tag{66}$$

As a first step, we show that $\mathbb{P}^\star$ still preserves the coupling.

**Lemma 10.** *For any Borel sets $A_0, A_1 \in \mathbb{R}^d$, it holds that*

$$\mathbb{P}^\star(\{\omega \in \mathcal{C}^{d(K+1)} : \omega(0) \in A_0 \times \mathbb{R}^{dK}, \omega(1) \in A_1 \times \mathbb{R}^{dK}\}) = \Pi_{0,1}(A_0 \times A_1). \tag{67}$$

*In other words, $\mathbb{P}^\star$ preserves the coupling $\Pi_{0,1}$.*

*Proof.* Any $\omega \in \mathcal{C}^{d(K+1)}$ decomposes uniquely into a pair $\omega = (\omega_X, \omega_Y)$ with $\omega_X \in \mathcal{C}^d$ and $\omega_Y \in \mathcal{C}^{dK}$. Next, we subsequently write $\mathbb{Q}^y(\cdot \,|\omega_X)$ for the (regular) conditional distribution of the OU process $Y$ conditional on the path of the reference process $X$ being $\omega_X \in \mathcal{C}^d$. Using the disintegration theorem, it therefore follows that

$$\mathbb{P}^\star(\{\omega \in \mathcal{C}^{d(K+1)} : \omega(0) \in A_0 \times \mathbb{R}^{dK}, \omega(1) \in A_1 \times \mathbb{R}^{dK}\}) \tag{68}$$

$$= \mathbb{P}^\star(\{\omega \in \mathcal{C}^{d(K+1)} : \omega_X(0) \in A_0, \omega_X(1) \in A_1\}) \tag{69}$$

$$= \int_{\mathcal{C}^{d(K+1)}} \frac{\mathrm{d}\mathbb{P}}{\mathrm{d}\mathbb{Q}}(\omega_X) \mathbb{1}_{A_0}(\omega_X(0)) \mathbb{1}_{A_1}(\omega_X(1)) \mathrm{d}\mathbb{S}(\omega) \tag{70}$$

$$= \int_{\mathcal{C}^d} \frac{\mathrm{d}\mathbb{P}}{\mathrm{d}\mathbb{Q}}(\omega_X) \mathbb{1}_{A_0}(\omega_X(0)) \mathbb{1}_{A_1}(\omega_X(1)) \int_{\mathcal{C}^{dK}} \mathrm{d}\mathbb{Q}^y(\omega_Y|\omega_X) \mathrm{d}\mathbb{Q}(\omega_X) \tag{71}$$

$$= \int_{\mathcal{C}^d} \frac{\mathrm{d}\mathbb{P}}{\mathrm{d}\mathbb{Q}}(\omega_X) \mathbb{1}_{A_0}(\omega_X(0)) \mathbb{1}_{A_1}(\omega_X(1)) \mathrm{d}\mathbb{Q}(\omega_X) \tag{72}$$

$$= \mathbb{P}(\{\omega \in \mathcal{C}^d : \omega(0) \in A_0, \omega(1) \in A_1\}) \tag{73}$$

$$= \Pi_{0,1}(A_0 \times A_1), \tag{74}$$

showing that $\mathbb{P}^\star$ preserves the coupling $\Pi_{0,1}$. $\qquad\square$

For any $z_0 \in \mathbb{R}^{d(K+1)}$, we subsequently denote by

$$\frac{d\mathbb{P}^\star_{1|0}}{d\mathbb{S}_{1|0}}(\,\cdot\,|z_0) : \mathbb{R}^{d(K+1)} \to [0,\infty) \tag{75}$$

the density of $\mathbb{P}^\star_{1|0}(\,\cdot\,|z_0)$ with respect to $\mathbb{S}_{1|0}(\,\cdot\,|z_0)$. In the same spirit, given $x_0 \in \mathbb{R}^d$, we write

$$\frac{d\mathbb{P}_{1|0}}{d\mathbb{Q}_{1|0}}(\,\cdot\,|x_0) : \mathbb{R}^d \to [0,\infty) \tag{76}$$

for the density of $\mathbb{P}_{1|0}(\,\cdot\,|x_0)$ with respect to $\mathbb{Q}_{1|0}(\,\cdot\,|x_0)$. By eq. (66), it follows that

$$\frac{d\mathbb{P}^\star_{1|0}}{d\mathbb{S}_{1|0}}(z_1|z_0) = \mathbb{1}_{\{0\}}(y_0)\frac{d\mathbb{P}_{1|0}}{d\mathbb{Q}_{1|0}}(x_1|x_0) \tag{77}$$

for all $z_0 = (x_0, y_0), z_1 = (x_1, y_1) \in \mathbb{R}^{d(K+1)}$. Now introduce two functions

$$h_1 : \mathbb{R}^{d(K+1)} \times \mathbb{R}^d \to [0,\infty), \qquad (z_0, x_1) \mapsto h_1(z_0, x_1) := \mathbb{1}_{\{0\}}(y_0)\frac{d\mathbb{P}_{1|0}}{d\mathbb{Q}_{1|0}}(x_1|x_0) \tag{78}$$

and, with this, $h : \mathbb{R}^{d(K+1)} \times [0,1] \times \mathbb{R}^{d(K+1)} \to [0,\infty)$ given by

$$h(z_0, t, z) := \int_{\mathbb{R}^{d(K+1)}} h_1(z_0, x_1)\mathbb{S}_{1|t}(dz_1|z), \quad (z_0, t, z) \in \mathbb{R}^{d(K+1)} \times [0,1] \times \mathbb{R}^{d(K+1)}. \tag{79}$$

Observe that

$$h(z_0, t, z) = \mathbb{E}_{\mathbb{S}_{1|t}}[h_1(z_0, X_1)|Z_0 = z_0, Z_t = z] = \mathbb{E}_{\mathbb{S}^1_{1|t}}[h_1(z_0, X_1)|Z_0 = z_0, Z_t = z], \tag{80}$$

where $\mathbb{S}^1_{1|t}$ denotes the conditional distribution of $X_1$ given $Z_t$. In particular, $h(z_0, 1, z) = h_1(z_0, x)$ whenever $z = (x, y)$. In what follows, we enforce the following assumptions on $h$.

**Assumption 2.** *The function $h$ defined in eq. (79) is jointly measurable. Moreover, for all fixed $z_0 \in \mathbb{R}^{d(K+1)}$, the mapping $(t, z) \mapsto h(z_0, t, z)$ satisfies*

$$\inf\{h(z_0, t, z) : (t, z) \in [0,1] \times \mathbb{R}^{d(K+1)}\} > 0 \tag{81}$$

*and is a member of $C^2_b([0,1] \times \mathbb{R}^{d(K+1)}, [0,\infty))$, the space of bounded and twice continuously differentiable functions with bounded first- and second-order derivatives.*

Under these assumptions, it is possible to show that the coupling preserving augmented measure $\mathbb{P}^\star$ is the distribution of a solution of a stochastic differential equation.

**Proposition 11.** *The SDE*

$$dZ^\star_t = FZ^\star_t dt + GG^T\mathbb{E}_{\mathbb{P}^\star_{1|0,t}}[\nabla_z \log \mathbb{S}^1_{1|t}(X^\star_1|Z^\star_t)|Z^\star_0, Z^\star_t]dt + GdB_t, \quad Z^\star_0 = (X_0, 0\ldots 0), \tag{82}$$

*admits a pathwise unique strong solution $Z^\star = (X^\star, Y^\star)$ with distribution $\mathbb{P}^\star$. In particular, $X^\star$ preserves the coupling $\Pi_{0,1}$, that is, $(X^\star_0, X^\star_1) \sim \Pi_{0,1}$.*

*Proof.* For $z_0 \in \mathbb{R}^{d(K+1)}$ and $t \in [0,1]$, consider the linear differential operator $\mathscr{L}^{z_0}_t$ mapping functions $\varphi \in C^2_b(\mathbb{R}^{d(K+1)})$ to

$$\mathscr{L}^{z_0}_t\varphi(z) = \langle Fz + (GG^T)\nabla_z \log h(z_0, t, z), \nabla\varphi(z)\rangle + \frac{1}{2}tr(GG^T\nabla^2\varphi(z)), \quad z \in \mathbb{R}^{d(K+1)}. \tag{83}$$

Due to the assumptions imposed on $h$, it follows from Lemma 3.1 in Palmowski and Rolski [57] that the local martingale problem associated with the operator $\mathscr{L}^{z_0}_t$ and initial distribution $\delta_{z_0}$ is solved by $\mathbb{P}^\star_{|0}(\,\cdot\,|z_0)$. Thus, by Theorem 18.7 in Kallenberg [58], it follows that the stochastic differential equation

$$dZ^\star(t) = FZ^\star(t)dt + GG^T\nabla_z \log h(z_0, t, Z^\star_t)dt + GdB_t, \qquad Z^\star_0 = z_0 \tag{84}$$

admits a weak solution in $Z^\star_0 = z_0$ with associated path measure $\mathbb{P}^\star_{|0}(\,\cdot\,|z_0)$. Next, since $h(z_0, \cdot) \in C^2_b([0,1] \times \mathbb{R}^{d(K+1)}, [0,\infty))$ implies that $(t, z) \mapsto \nabla_z \log h(z_0, t, z)$ is Lipschitz continuous and

therefore the solution of the SDE is even strong and pathwise unique. Finally, it follows that the pathwise unique strong solution $Z^\star$ of

$$\mathrm{d}Z^\star(t) = FZ^\star(t)\mathrm{d}t + GG^T\nabla_z \log h(Z_0^\star, t, Z_t^\star)\mathrm{d}t + G\mathrm{d}B_t, \qquad Z_0^\star = (X_0, 0, \ldots, 0) \quad (85)$$

has distribution $\mathbb{P}^\star$ as $\mathbb{P}_0^\star = \tilde{\Pi}_0$. We conclude since

$$\nabla_z \log h(Z_0^\star, t, Z^\star) = \mathbb{E}_{\mathbb{P}_{1|0,t}^\star}[\nabla_z \log \mathbb{S}_{1|t}^1(X_1^\star|Z_t^\star)|Z_0^\star, Z_t^\star] \quad (86)$$

using eq. (80) and following the arguments in Bortoli et al. [32, Proof of Proposition 3]. $\qquad\square$

In Theorem 11 we constructed the coupling-preserving path measure $\mathbb{P}^\star$ associated with the stochastic process we wish to learn. The following corollary establishes that we can obtain samples $z_t^\star \sim \mathbb{P}_t^\star$ by first sampling $(x_0, x_1) \sim \Pi_{0,1}$ and subsequently sampling $z_t \sim \mathbb{S}_{t|X_0, X_1}(\cdot \mid x_0, x_1)$.

**Corollary 12.** *For the coupling-preserving process $Z^\star$ constructed in Theorem 11, the associated path measure satisfies $\mathbb{P}^\star = \Pi_{0,1}\mathbb{S}_{|X_0, X_1}$.*

*Proof.* Since $\mathbb{P}^\star$ preserves the coupling $\Pi_{0,1}$, we have

$$\mathbb{P}^\star = \int_{\mathbb{R}^d \times \mathbb{R}^d} \mathbb{P}_{|X_0, X_1}^\star(\cdot|x_0, x_1)\mathrm{d}\mathbb{P}_{X_0, X_1}^\star(x_0, x_1) \quad (87)$$

$$= \int_{\mathbb{R}^d \times \mathbb{R}^d} \mathbb{P}_{|X_0, X_1}^\star(\cdot|x_0, x_1)\mathrm{d}\Pi_{0,1}(x_0, x_1) \quad (88)$$

$$= \Pi_{0,1}\mathbb{P}_{|X_0, X_1}^\star. \quad (89)$$

For $X_1^\star = x_1$ we find

$$\mathbb{E}_{\mathbb{P}_{1|0,t}^\star}[\nabla_z \log \mathbb{S}_{1|t}^1(X_1^\star|Z_t^\star)|Z_0^\star, Z_t^\star]_{X_1^\star = x_1} = \mathbb{E}_{\mathbb{P}_{1|0,t}^\star}[\nabla_z \log \mathbb{S}_{1|t}^1(x_1|Z_t^\star)|\sigma(Z_0^\star, Z_t^\star)] \quad (90)$$

$$= \nabla_z \log \mathbb{S}_{1|t}^1(x_1|Z_t^\star), \quad (91)$$

since $\nabla_z \log \mathbb{S}_{1|t}^1(x_1|Z_t^\star)$ is measurable with respect to $\sigma(Z_0^\star, Z_t^\star)$. Therefore $Z_{X_0^\star, X_1^\star}^\star$ solves the SDE in eq. (48) of the partially pinned process and we conclude $Z_{X_0, X_1}^\star \overset{d}{=} Z_{X_0, X_1}$ such that

$$\mathbb{P}^\star = \Pi_{0,1}\mathbb{S}_{X_0, X_1}. \quad (92)$$

$\qquad\square$

Given a data point $X_0^\star = x_0 \sim \Pi_0$, and assuming we could simulate the coupling preserving process $Z^\star$, we could sample from the coupling $\Pi_{0,1}$ by simulating the SDE in eq. (11) forward in time on $[0, 1]$ to arrive at a sample $X_1^\star = x_1$. As $X^\star$ preserves the coupling, it follows that $(x_0, x_1)$ is drawn from $\Pi_{0,1}$. However, the expectation in the drift of $Z^\star$ is intractable and hence we approximate this expectation by a time-dependent neural network $u_t^\theta$. We now define *Fractional Diffusion Bridge Models (FDBM)* for paired data translation as the stochastic process $Z^\theta$ associated with the path measure $\mathbb{P}^\theta$ solving

$$\mathrm{d}Z_t^\theta = FZ_t^\theta\mathrm{d}t + GG^T u^\theta(t, X_0, Z_t^\theta)\mathrm{d}t + G\mathrm{d}B_t, \quad Z_0^\theta = (X_0, 0, \ldots, 0), \quad (93)$$

$$u_i^\theta(t, x_0, z) = [1, \omega_1\zeta_1(t, 1), \ldots, \omega_K\zeta_K(t, 1)]^T \tilde{u}_i^\theta(t, x_0, \mu_{1|t}(z)), \quad u^\theta = (u_1^\theta, \ldots, u_d^\theta), \quad (94)$$

where $\tilde{u}^\theta := (\tilde{u}_1^\theta, \ldots, \tilde{u}_d^\theta)$ is a time-dependent neural network that takes the starting value $x_0$ and the mean $\mu_{1|t}(z)$ of the conditional terminal $X_1|(Z_t = z)$ as an input. Denote

$$\tilde{v}(t) = [1, \omega_1\zeta_1(t, 1), \ldots, \omega_K\zeta_K(t, 1)]^T \in \mathbb{R}^{K+1} \quad (95)$$

and define

$$v(t) = \begin{pmatrix} \tilde{v}(t) & 0 & \ldots & 0 \\ 0 & \tilde{v}(t) & \ldots & 0 \\ \vdots & \ddots & \ddots & 0 \\ 0 & \ldots & \ldots & \tilde{v}(t) \end{pmatrix} \in \mathbb{R}^{d(K+1), d}. \quad (96)$$

Since $t \mapsto \|G^T \tilde{v}(t)\|_2^2$ is continuous, it attains its maximum on the compact interval $[0, 1]$. Hence, we find $\|G^T v(t)\|_2^2 \leq c$ for some constant $c > 0$. Parameterizing the learnable process $Z^\theta$ associated with the path measure $\mathbb{P}^\theta$ according to eq. (93) we aim to minimize the KL-divergence $D_{\mathrm{KL}}(\mathbb{P}^\star | \mathbb{P}^\theta)$. We calculate using Girsanovs theorem (See Blessing et al. [59, eq. (30)] for our setting), together with the stochastic Fubini theorem and Jensens inequality

$$D_{\mathrm{KL}}(\mathbb{P}^\star | \mathbb{P}^\theta) = \mathbb{E}_{\mathbb{P}_{0,t}^\star} \left[ \frac{1}{2} \int_0^1 \left\| G^T \left\{ \mathbb{E}_{\mathbb{P}_{1|0,t}^\star} \left[ \nabla_z \log \mathbb{S}_{1|t}^1 (X_1^\star | Z_t^\star) | Z_0^\star, Z_t^\star \right] - u^\theta(t, X_0, Z_t^\star) \right\} \right\|_2^2 dt \right] \quad (97)$$

$$= \frac{1}{2} \int_0^1 \mathbb{E}_{\mathbb{P}_{0,t}^\star} \left[ \| G^T \left\{ \mathbb{E}_{\mathbb{P}_{1|0,t}^\star} [\nabla_z \log \mathbb{S}_{1|t}^1 (X_1^\star | Z_t^\star) - u^\theta(t, X_0, Z_t^\star) | Z_0^\star, Z_t^\star] \right\} \|_2^2 \right] dt \quad (98)$$

$$\leq \frac{1}{2} \int_0^1 \mathbb{E}_{\mathbb{P}_{0,t}^\star} \left[ \left\| G^T v(t) \right\|_2^2 \left\| \mathbb{E}_{\mathbb{P}_{1|t,0}^\star} \left[ \frac{X_1^* - \mu_{1|t}(Z_t^\star)}{\sigma_{1|t}^2} - \tilde{u}^\theta(t, X_0, Z_t^\star)) \Big| Z_0^\star, Z_t^\star \right] \right\|_2^2 \right] dt \quad (99)$$

$$\leq \frac{c}{2} \int_0^1 \mathbb{E}_{\mathbb{P}_{0,t}^\star} \left[ \left\| \mathbb{E}_{\mathbb{P}_{1|t,0}^\star} \left[ \frac{X_1^* - \mu_{1|t}(Z_t^\star)}{\sigma_{1|t}^2} - \tilde{u}^\theta(t, X_0, Z_t^\star)) \Big| Z_0^\star, Z_t^\star \right] \right\|_2^2 \right] dt \quad (100)$$

$$\leq \frac{c}{2} \int_0^1 \mathbb{E}_{\mathbb{P}_{0,t}^\star} \left[ \mathbb{E}_{\mathbb{P}_{1|t,0}^\star} \left[ \left\| \frac{X_1^* - \mu_{1|t}(Z_t^\star)}{\sigma_{1|t}^2} - \tilde{u}^\theta(t, X_0, Z_t^\star)) \right\|_2^2 \Big| Z_0^\star, Z_t^\star \right] \right] dt \quad (101)$$

$$= \frac{c}{2} \int_0^1 \mathbb{E}_{\mathbb{P}^\star} \left[ \left\| \frac{X_1^* - \mu_{1|t}(Z_t^\star)}{\sigma_{1|t}^2} - \tilde{u}^\theta(t, X_0, Z_t^\star)) \right\|_2^2 \right] dt. \quad (102)$$

Hence, we aim to minimize Equation (102) in order to learn the stochastic process $Z^\star$. During training, the loss is computed by first sampling $(x_0, x_1) \sim \Pi_{0,1}$ and subsequently sampling $z_t^\star \sim \mathbb{S}_{t|X_0,X_1}(\cdot \mid x_0, x_1)$. This procedure is justified since $\mathbb{P}^\star = \Pi_{0,1} \mathbb{S}_{|X_0,X_1}$ by Corollary 12.

### B.3 Theoretical framework for unpaired data

Given two unknown distributions $\Pi_0$ and $\Pi_1$ and the reference process $X = \sqrt{\epsilon} \hat{B}_H$ we seek to find a solution to the dynamic Schrödinger Bridge problem [39, 40, 42]

$$\mathbb{T}^{SB} = \underset{\mathbb{T} \in \mathcal{P}(\mathcal{C}^d)}{\arg\min} \left\{ D_{\mathrm{KL}}(\mathbb{T} | \mathbb{Q}) \ ; \ \mathbb{T}_0 = \Pi_0, \ \mathbb{T}_0 = \Pi_1 \right\}. \quad (103)$$

By Föllmer [41], Léonard [42, Proposition 2.3] there is at most one solution $\mathbb{T}^{SB}$ to the dynamic Schrödinger bridge problem in eq. (103) and if the solution $\mathbb{T}^{SB}$ exists, then $\mathbb{T}_{01}^{SB}$ is the solution to the static Schrödinger bridge problem. Assume there exists a solution $\mathbb{T}^{SB}$ for the above dynamik Schrödinger bridge w.r.t. $\mathbb{Q}$ such that $\Pi_{0,1}^{SB} := \mathbb{T}_{0,1}^{SB}$ is the solution to the corresponding static Schrödinger bridge problem. By the above Theorem 11 we can construct a process $Z^\star = (X^\star, Y^\star)$ with path measure $\mathbb{P}^\star$ and dynamics

$$dZ_t^* = F Z_t^* dt + G G^T \mathbb{E}_{\mathbb{P}_{1|0,t}^\star} [\nabla_z \log \mathbb{S}_{1|t}^1 (X_1^*|Z_t^*)|Z_0^*, Z_t^*] dt + G dB(t) \quad (104)$$

that preserves the coupling $\Pi_{0,1}^{SB}$. In contrast to the setting of paired training data, we have no access to samples of $\Pi_{0,1}^{SB}$. On the other hand, letting $\mathbb{S}$ be the path measure associated with the augmented reference process $Z$, we define using the marginals of $\mathbb{P}^\star$ the SB problem on the augmented space via

$$\mathbb{V}^{SB} = \underset{\mathbb{V} \in \mathcal{P}(\mathcal{C}^{d \cdot (K+1)})}{\arg\min} \left\{ D_{\mathrm{KL}}(\mathbb{V} | \mathbb{S}) \ ; \ \mathbb{V}_0 = \mathbb{P}_0^\star, \ \mathbb{V}_1 = \mathbb{P}_1^\star \right\}. \quad (105)$$

Since $Z$ is a Markov process, the path measure solving the lifted SB problem in eq. (105) is associated with a Markovian process [42], whereas $Z^\star$ in eq. (11) is non-Markovian due to its dependency on $X_0$ in the drift function. Motivated by this observation, we generalize in the following the definition of a reciprocal class [34, 47] and the notation of a Markovian projection [21, 34, 48] to our setting of a scaled MA-fBM reference process. We define the augmented reciprocal class $\mathcal{R}_a(\mathbb{S})$ below as the set of path measures $\mathbb{V}$ on the augmented space whose marginals can be sampled by first drawing $(x_0, x_1) \sim \mathbb{V}_{X_0,X_1}$ and then sampling $z_t \sim \mathbb{S}_{t|X_0,X_1}(\cdot \mid x_0, x_1)$.

**Definition 13.** *We say that $\mathbb{V} \in \mathcal{P}(\mathcal{C}^{d \cdot (K+1)})$ is in the augmented reciprocal class $\mathcal{R}_a(\mathbb{S})$ of $\mathbb{S}$ if*

$$\mathbb{V} = \int_{\mathbb{R}^d \times \mathbb{R}^d} \mathbb{S}_{|X_0,X_1}(\cdot | x_0, x_1) d\mathbb{V}_{X_0,X_1}(x_0, x_1) =: \mathbb{V}_{X_1,X_0} \mathbb{S}_{|X_0,X_1}. \quad (106)$$

*For any* $\mathbb{V} \in \mathcal{P}(\mathcal{C}^{d \cdot (K+1)})$ *we define the augmented reciprocal projection by*

$$\text{proj}_{\mathcal{R}_a(\mathbb{S})}(\mathbb{V}) := \mathbb{V}_{X_0, X_1} \mathbb{S}_{|X_0, X_1}. \tag{107}$$

Since we know that the solution to the lifted SB problem in eq. (105) is a Markovian measure, we project any element of the augmented reciprocal class to a Markovian path measure by the following definition.

**Definition 14.** *For* $\mathbb{V} \in \mathcal{P}(\mathcal{C}^{d \cdot (K+1)})$ *with* $\mathbb{V} \in \mathcal{R}_a(\mathbb{S})$ *we define the augmented Markovian projection* $\text{proj}_{\mathcal{M}_a}(\mathbb{V})$ *by the path measure associated to* $M = (M^1, M^2, \ldots M^{K+1})$ *solving for* $M_0^1 \sim \mathbb{V}_{M_0^1}$

$$\mathrm{d}M_t = FM_t \mathrm{d}t + GG^T \mathbb{E}_{\mathbb{V}_{1|t}} \left[ \nabla_{m_t} \log \mathbb{S}_{1|t}^1(M_1^1|M_t)|M_t \right] \mathrm{d}t + G\mathrm{d}B_t, \quad M_0 = (M_0^1, 0_K). \tag{108}$$

Bortoli et al. [27] introduce a flow of path measures $(\mathbb{P}^s, \mathbb{P}^s)_{s \geq 0}$ and show that, for a reference process driven by BM, a time discretization of this flow with step size $\alpha \in (0, 1]$ yields a family of procedures called $\alpha$-IMF, all of which converge to the Schrödinger bridge. For a reference process driven by MA-fBm, we propose to define a flow of path measures $(\tilde{\mathbb{P}}^s, \hat{\mathbb{P}}^s)_{s \geq 0}$ recursively by

$$\hat{\mathbb{P}}^0 = (\Pi_0 \otimes \Pi_1)\mathbb{S}_{|X_0, X_1}, \quad \partial_s \hat{\mathbb{P}}^s = \text{proj}_{\mathcal{R}_a(\mathbb{S})}(\text{proj}_{\mathcal{M}_a(\mathbb{S})}(\hat{\mathbb{P}}^s)) - \hat{\mathbb{P}}^s, \quad \tilde{\mathbb{P}}^s = \text{proj}_{\mathcal{M}_a(\mathbb{S})}(\hat{\mathbb{P}}^s), \tag{109}$$

Both procedures $\alpha$-IMF and IMF are based on the loss function [27, 34]

$$\mathcal{L}(v, \mathbb{P}) = \int_0^t \mathcal{L}_t(v_t, \mathbb{P}) \mathrm{d}t = \int_0^1 \int_{(\mathbb{R}^d)^3} \left\| v_t^\theta(x_t) - \frac{x_1 - x_t}{1 - t} \right\|^2 \mathrm{d}\mathbb{P}(x_0, x_1) \mathrm{d}\mathbb{Q}_{t|0,1}(x_t|x_0, x_1) \mathrm{d}t. \tag{110}$$

We propose to replace the above loss function with

$$\mathcal{L}_{\text{FDBM}}^{\text{unpaired}}(\theta, \tilde{\mathbb{P}}) = \int_0^1 \int_{(\mathbb{R}^{d \cdot (K+1)})} \int_{(\mathbb{R}^d)^2} \left\| \tilde{v}_t^\theta(\mu_{1|t}(z_t)) - \frac{x_1 - \mu_{1|t}(z)}{\sigma_{1|t}^2} \right\|^2 \mathrm{d}\tilde{\mathbb{P}}^x(x_0, x_1) \mathrm{d}\mathbb{S}_{t|X_0, X_1}(z_t|x_0, x_1) \mathrm{d}t. \tag{111}$$

to define $\alpha$-IMF with respect to a scaled MA-fBM reference process.

**Challenges & Limitations**. The dynamic Schrödinger bridge problem can be formulated with a scaled fBM as the reference process, since Léonard [42] includes non-Markovian processes with continuous paths. To sample paths from the resulting solution, one must draw from a fractional Brownian bridge (fBB). Janak [24] constructs such a bridge by leveraging the fact that fBM is a Gaussian process and additionally derives an integral equation characterizing the fBB [24, Theorem 5]. However, the drift of the derived bridge involves an integral that is not available in closed form [24, eq. (17)], necessitating an approximation of this drift term when sampling from an (approximate) solution to the dynamic Schrödinger bridge problem. Hence, we first approximate fBM using a Markovian approximation [25, 26] to enable simulation—up to discretization error—of the exact bridge, which corresponds to a partially pinned process. We leave the analysis of how well the solution to the thus-defined dynamic Schrödinger bridge problem approximates the solution of the corresponding problem with a scaled fBM as the reference process for future work. We emphasize that in the unpaired training data setting, we only propose a method for using FDBM and do not prove convergence of the algorithm to the corresponding solution of the dynamic Schrödinger bridge problem. To the best of our knowledge, the setting of Léonard et al. [47, Theorem 2.14] is not applicable here, as our pinned path measure refers to a partially pinned process, rather than a fully pinned process. As a result, proving the convergence of our method would require an adaptation of Léonard et al. [47, Theorem 2.14], which is beyond the scope of this work. Additionally we point out that we are only able to simulate the learned bridges forward in time, since the terminal distribution of the augmenting processes of the learned stochastic bridge depends on the initial data distribution, see Section B.5 for details.

## B.4 Sampling from partially pinned process

In this section, we derive the marginal distribution of the partially pinned process for any $t \in (0, 1)$, enabling simulation-free sampling. For $s < t < 1$ we know that $(X_t, Y_t, X_1)|(Z_s = z)$ is Gaussian [2] with

$$(X_t, Y_t^1, \ldots, Y_t^K, X_1 | Z_s = z)^T \sim \mathcal{N}\left( \begin{pmatrix} \eta_{t|s}(z) \\ \mu_{1|s}(z) \end{pmatrix}, \begin{pmatrix} \Sigma_{t|s} & \Sigma_{12}(t|s) \\ \Sigma_{21}(t|s) & \sigma_{1|s}^2 \end{pmatrix} \right), \tag{112}$$

with

$$\eta_{t|s}(z) = (\eta_{t|s}^1(z), \eta_{t|s}^2(z), ..., \eta_{t|s}^{K+1}(z))^T \tag{113}$$

and

$$\Sigma_{12}(t|s) = (cov(X_t, X_1), cov(Y_1^1, X_1), ..., cov(Y_t^K, X_1))^T = \Sigma_{21}^T(t|s). \tag{114}$$

Hence, the process partially pinned at $(x_s, x_1)$ follows the distribution

$$Z_t|(X_s = x_s, X_1 = x_1) \sim \mathcal{N}(\bar{\eta}_{t|x_s, x_1}, \bar{\Sigma}_{t|s,1}), \tag{115}$$

with

$$\bar{\eta}_{t|x_0, x_s}(z) = \eta_{t|s}(z) + \frac{1}{\sigma_{1|s}^2}\Sigma_{12}(t|s)(x_1 - \mu_{1|s}(z)) \tag{116}$$

$$\stackrel{s=0}{=} (x_0, 0, ..., 0)^T + \frac{1}{\sigma_{1|0}^2}\Sigma_{12}(t|s)(x_1 - x_0) \tag{117}$$

and

$$\bar{\Sigma}_{t|s,1} = \Sigma_{t|s} - \frac{1}{\sigma_{1|t}^2}\Sigma_{12}(t|s)\Sigma_{21}(t|s) = \Sigma_{t|s} - \frac{1}{\sigma_{1|t}^2}\Sigma_{12}(t|s)\Sigma_{12}^T(t|s). \tag{118}$$

We further calculate for a constant diffusion coefficient $g(t) \equiv g \in \mathbb{R}$

$$\zeta_k(s,t) = \int_s^t -\gamma_k g(u)e^{-\gamma_k(u-s)}\mathrm{d}u = -g\gamma_k \int_s^t e^{-\gamma_k(u-s)}du = g(e^{-\gamma_k(t-s)} - 1) \tag{119}$$

and

$$\mu_{1|t}(z) = x + \sum_k \omega_k y_k \zeta_k(t,1) = x + g \sum_{k=1}^K \omega_k(e^{-\gamma_k(1-t)} - 1)y_k. \tag{120}$$

Left to calculate are the entries of $\Sigma_{t|s}$ and $\Sigma_{12}(t|s)$. With $s < t \le 1$ we calculate

$$\mathrm{Cov}(X_t, X_1|Z_s = z) = \sum_{i,j=1}^K \omega_i\omega_j \int_s^t (\zeta_i(u,t) + g)(\zeta_j(u,1) + g)\,\mathrm{d}u \tag{121}$$

$$= g^2 \sum_{i,j=1}^K \omega_i\omega_j \int_s^t \left((e^{-\gamma_k(t-u)} - 1) + 1\right)\left((e^{-\gamma_k(1-u)} - 1) + 1\right)\mathrm{d}u \tag{122}$$

$$= g^2 \sum_{i,j=1}^K \omega_i\omega_j \int_s^t \left(e^{-\gamma_i(t-u)}\right)\left(e^{-\gamma_j(1-u)}\right)\mathrm{d}u, \tag{123}$$

$$\mathrm{Cov}(Y_t^i, Y_1^j|Z_s = z) = \int_s^t e^{-\gamma_i(t-u)}e^{-\gamma_j(1-u)}du = \frac{e^{-\gamma_j - t\gamma_i}\left(e^{t\gamma_j + t\gamma_i} - e^{s\gamma_j + s\gamma_i}\right)}{\gamma_j + \gamma_i}, \tag{124}$$

and for $s = 0$

$$\mathrm{Cov}(Y_t^l, X_1) = \sum_{k=1}^K \omega_k \int_0^t e^{-\gamma_l(t-u)}(\zeta_k(u,1) + g(u))du \tag{125}$$

$$= g \sum_{k=1}^K \omega_k \int_0^t e^{-\gamma_l(t-u)}e^{-\gamma_k(1-u)}du \tag{126}$$

$$= g \sum_{k=1}^K \omega_k \frac{\left(e^{t(\gamma_l + \gamma_k)} - 1\right)e^{-t\gamma_l - \gamma_k}}{\gamma_l + \gamma_k}. \tag{127}$$

## B.5 Loss regularization via the reverse pinned process

In the derivation of the previous section (Section B.4), we see from eq. (117) that the terminal values of the noise process $Y$ directly depend on $x_0$, i.e., on information from the initial distribution $\Pi_0$. Hence, initializing the time reversal of the partially pinned process is only feasible when the desired endpoint is already known, which makes simulating the time reversal of FDBM impractical in general. However, we derive below the time reversal of the partially pinned process, which allows us to use the drift of the reversed process to regularize the loss function during training in the unpaired setting, where we condition on both an initial and a terminal state. Whenever $X = (X(t))_{t\in[0,1]}$ is a stochastic process and $g$ is a function on $[0,1]$, we write $\bar{X}(t) = X(1-t)$ for the reverse-time model and $\bar{g}(t) = g(1-t)$ for the reverse-time function. In Bortoli et al. [27] the reverse pinned process connecting $x_1$ and $x_0$ is again a Brownian bridge. For our reference process, the reverse model of the partially pinned process follows [60]

$$d\bar{Z}_{|x_0,x_1}(t) = \left[F\bar{Z}_{|x_0,x_1}(t) + GG^T u^{\rightarrow}(1-t,\bar{Z}_{|x_0,x_1}(t)) - GG^T\nabla_z\log\bar{p}_t(\bar{Z}_{|x_0,x_1}(t)|x_0,x_1)\right]dt + Gd\bar{B}(t) \quad (128)$$

$$= \left\{F\bar{Z}_{|x_0,x_1}(t) + GG^T\left[u^{\rightarrow}(1-t,\bar{Z}_{|x_0,x_1}(t)) - \nabla_z\log\bar{p}_t(\bar{Z}_{|x_0,x_1}(t)|x_0,x_1)\right]\right\}dt + Gd\bar{B}(t) \quad (129)$$

$$= \left\{F\bar{Z}_{|x_0,x_1}(t) + GG^T\left[\nabla_z\log q_{1|t}(x_1|\bar{Z}_{|x_0,x_1}(t)) - \nabla_z\log\bar{p}_t(\bar{Z}_{|x_0,x_1}(t)|x_0,x_1)\right]\right\}dt + Gd\bar{B}(t) \quad (130)$$

where $q_{1|t}(\cdot|z) := \mathbb{S}^1_{1|t}(\cdot|z)$ is the density of $X_1|(Z_t = z)$, $p_t := \mathbb{S}_t$ is the marginal density of the augmented reference process $Z$, $p_t(\cdot|x_0,x_1) := \mathbb{S}_{t|X_0,X_1}(\cdot|x_0,x_1)$ is the marginal density of the partially pinned process defined in eq. (9) and $u^{\rightarrow} := u$ according to eq. (10). We find with Bayes' theorem

$$p_t(z|x_0,x_1) = \frac{\rho_t(z,x_0,x_1)}{\pi_{0,1}(x_0,x_1)}, \quad (131)$$

where $\pi_{0,1}$ is the joint density associated to $\Pi_{0,1}$ and $\rho_t$ is the joint density of $(Z_t, X_0, X_1)$. Since $Z$ is Markov with $Z_0 = (X_0, 0_{dK})$, we have

$$q_{1|t,x_0}(x_1|z,x_0) = q_{1|t,s}(x_1|z_t,z_0) = q_{1|t}(x_1|z_t) \quad (132)$$

and

$$q_{1|t}(x_1|z_t) = q_{1|t}(x_1|z_t,z_0) = \frac{\rho_t(z_t,x_0,x_1)}{p(z_t,x_0)} = \frac{\rho_t(z_t,x_0,x_1)}{p_t(z_t|x_0)\pi_0(x_0)}, \quad (133)$$

where $\pi_0$ corresponds to $\Pi_0$. Hence, by the above equations

$$\log q_{1|t}(x_1|z_t) - \log p_t(z_t|x_0,x_1) \quad (134)$$

$$= \log\left(\frac{\rho_t(z_t,x_0,x_1)}{p_t(z_t|x_0)\pi_0(x_0)}\cdot\frac{\lambda(x_0,x_1)}{\rho_t(z_t,x_0,x_1)}\right) \quad (135)$$

$$= \log\left(\frac{\lambda(x_0,x_1)}{p_t(z_t|x_0)\pi_0(x_0)}\right) \quad (136)$$

$$= \log\lambda(x_0,x_1) - \log p_t(z_t|x_0) - \log\pi_0(x_0) \quad (137)$$

and we find for the gradient

$$\nabla_z\left[\log q_{1|t}(x_1|z_t) - \log p_t(z_t|x_0,x_1)\right] = -\nabla_z\log p_t(z_t|x_0), \quad (138)$$

such that

$$d\bar{Z}_{|x_0,x_1}(t) = \left[F\bar{Z}_{|x_0,x_1}(t) - GG^T\nabla_z\log\bar{p}_t(z_{1-t}|x_0)\right]dt + Gd\bar{B}(t). \quad (139)$$

Hence, the reverse dynamics of the partially pinned process coincide with the reverse dynamics of the reference process conditioned on $x_0$. In addition, we have

$$\log p_t(z_t|x_0) = \nabla_y[\log p_t^x(x_t|y_t^1,...,y_t^k,x_0) + \log p_t^y(y_t^1,...,y_t^k|x_0)] \quad (140)$$

$$= \nabla_y[\log p_t^x(x_t|y_t^1,...,y_t^k,x_0) + \log p_t^y(y_t^1,...,y_t^k)], \quad (141)$$

where we use the independence of $(Y^1,...,Y^K)$ and $X_0$. To calculate further, we note that $X_t|(Y_t^1 = y^1,...,Y_t^K = y^K, X_0 = x_0) \sim \mathcal{N}(\mu_t(y,x_0), \sigma_{t|Y}^2)$ is normal distributed with

$$\mu_t(y,x_0) = \mathbb{E}\left[X_t|(Y_t^1 = y^1,...,Y_t^K = y^K, X_0 = x_0)\right] \quad (142)$$

$$\overset{(41)}{=} x_0 + \sum_{k=1}^K \omega_k\zeta_k(0,t)y_k(t) \quad (143)$$

$$= \mu_{t|0}(z) - x + x_0, \quad (144)$$

where $z = (x, y_1, ..., y_K)$ and

$$\sigma_{t|Y}^2 = \mathbb{V}\left[X_t | (Y_t^1 = y^1, ..., Y_t^K = y^K, X_0 = x_0)\right] \tag{145}$$

$$= \sqrt{\epsilon} \sum_{i,j=1}^{K} \frac{\omega_i \omega_j}{\gamma_i + \gamma_j} (1 - e^{-t(\gamma_i + \gamma_j)}) \tag{146}$$

$$\stackrel{(35)}{=} \sigma_{t|0}^2. \tag{147}$$

Therefore

$$\partial_x \log p_t(z_t | x_0) \tag{148}$$

$$= \partial_x [\log p_t^x(x_t | y_t^1, ..., y_t^k, x_0) + \log p_t^y(y_t^1, ..., y_t^k)] \tag{149}$$

$$= \partial_x \log p_t(x_t | y_t^1, ..., y_t^k, x_0) \tag{150}$$

$$= -\frac{x_t - [x_0 + \sum_{k=1}^{K} \omega_k \zeta_k(0, t) y_t^k]}{\sigma_{t|0}^2} \tag{151}$$

$$= \frac{x_0 - x_t + \sum_{k=1}^{K} \omega_k \zeta_k(0, t) y_k(t)}{\sigma_{t|0}^2} \tag{152}$$

and

$$\nabla_y \log p_t(z_t | x_0) = \nabla_y [\log p_t^x(x_t | y_t^1, ..., y_t^K, x_0) + \log p_t^y(y_t^1, ..., y_t^K)] \tag{153}$$

$$= -\frac{x_t - \mu_{t|(Y, x_0)}}{\sigma_{t|0}^2} [-\nabla_y \mu_{t|(Y, x_0)}] + \nabla_y \log p_t^y(y_t^1, ..., y_t^K) \tag{154}$$

$$= [\omega_1 \zeta_1(0, t), ..., \omega_K \zeta_K(0, t)]^T \frac{x_t - \mu_{t|(Y, x_0)}}{\sigma_{t|0}^2} - \left( \Lambda_t^{-1} \begin{pmatrix} 0 \\ \begin{pmatrix} y_t^1 \\ \vdots \\ y_t^K \end{pmatrix} \end{pmatrix} \right), \tag{155}$$

$$\tag{156}$$

such that, in total

$$\mathrm{d}\bar{Z}_{|x_0, x_1}(t) = \left\{ F\bar{Z}_{|x_0, x_1}(t) - GG^T u^{\leftarrow}(1 - t, \bar{Z}_{|x_0, x_1}(t)) \right\} \mathrm{d}t + G\mathrm{d}\bar{B}(t), \tag{157}$$

with

$$u^{\leftarrow}(t, z) = [1, -\omega_1 \zeta_1(0, t), ..., -\omega_K \zeta_K(0, t)]^T \frac{x_0 - x + \sum_{k=1}^{K} \omega_k \zeta_k(0, t) y_k}{\sigma_{t|0}^2} - \left( \Lambda^{-1}(t) \begin{pmatrix} 0 \\ \begin{pmatrix} y_t^1 \\ \vdots \\ y_t^K \end{pmatrix} \end{pmatrix} \right). \tag{158}$$

We use the above calculations to derive a backward loss. Let $Z_{|0,1}(t) \sim \mathcal{N}(\bar{\Sigma}_t, \bar{\mu}_t)$ with

$$\bar{\mu}_t = (x_0, 0, ..., 0)^T + \frac{1}{\sigma_{1|0}^2} \Sigma_{12}(t|0)(x_1 - x_0) \tag{159}$$

and

$$\bar{\Sigma}_t = \Sigma_{t|0} - \frac{1}{\sigma_{T|t}^2} \Sigma_{12}(t|0) \Sigma_{12}^T(t|0) \tag{160}$$

according to the derivations in Section B.4. Since by the calculations of this section

$$\nabla_z \log q_{1|t}(x_1 | z) - \nabla_z \log \bar{p}_t(z | x_0, x_1) - u^{\leftarrow}(t, z) = 0_{d(K+1)}, \tag{161}$$

we aim to enforce

$$0_{d(K+1)} = [1, \omega_1 \zeta_1(t, 1), ..., \omega_K \zeta_K(t, 1)]^T v_t^\theta(\mu_{1|t}(z)) - \nabla_z \log p_t(Z_{|0,1}(t) | x_0, x_1) - u^{\leftarrow}(t, z) \tag{162}$$

for the neural network $v_t^\theta$ learned to approximation the forward dynamics transforming $\pi_0$ to $\pi_1$. Moreover, since

$$\nabla_z \log p_t(Z_{|0,1}(t)|x_0, x_1) = -\bar{\Sigma}_t^{-1}(Z_{|0,1}(t) - \bar{\mu}_t), \tag{163}$$

we aim for

$$\mathbf{0}_{K+1} \stackrel{d}{=} \bar{\Sigma}_t \left\{ [1, \omega_1 \zeta_1(t,1), ..., \omega_K \zeta_K(t,1)]^T \, \overrightarrow{v_t}(\mu_{1|t}(Z_{|0,1}(t))) - \overleftarrow{u}(t, Z_{|0,1}(t)) \right\} - (Z_{|0,1}(t) - \bar{\mu}_t) \tag{164}$$

and define

$$\mathcal{L}_t^{\leftarrow}(\theta, z_t, x_0, x_1) := \left\| \bar{\Sigma}_t \left\{ [1, \omega_1 \zeta_1(t,1), ..., \omega_K \zeta_K(t,1)]^T \, v_t^\theta(\mu_{1|t}(Z_{|0,1}(t))) - \overleftarrow{u}(t, Z_{|0,1}(t)) \right\} - (Z_{|0,1}(t) - \bar{\mu}_t) \right\|^2, \tag{165}$$

with

$$\overleftarrow{u}(t,z) = \left\{ [1, -\omega_1 \zeta_1(0,t), \ldots, -\omega_K \zeta_K(0,t)]^T \, \frac{x_0 - x + \sum_{k=1}^K \omega_k \zeta_k(0,t) y_k}{\sigma_{t|0}^2} - \left( \begin{matrix} 0 \\ \Lambda^{-1}(t) \begin{pmatrix} y_1 \\ \vdots \\ y_K \end{pmatrix} \end{matrix} \right) \right\} \tag{166}$$

to minimize for some $\lambda \in [0,1]$

$$\mathcal{L}_t(\theta, v^\theta, \tilde{\mathbb{P}}) \tag{167}$$

$$= \int_{\mathbb{R}^{d(K+1)}} \int_{(\mathbb{R}^d)^2} \left\| \overrightarrow{v_t}(\mu_{1|t}(z_t)) - \frac{x_1 - x_t - \sum_k \omega_k \zeta_k(t,1) y_k(t)}{\sigma_{1|t}^2} \right\|^2 \mathrm{d}\tilde{\mathbb{P}}_{X_0, X_1}(x_0, x_1) \mathrm{d}\mathbb{S}_{t|X_0, X_1}(z_t|x_0, x_1)$$

$$+ \lambda \int_{\mathbb{R}^{d(K+1)}} \int_{(\mathbb{R}^d)^2} \mathcal{L}_t^{\leftarrow}(\theta, z_t, x_0, x_1) \mathrm{d}\tilde{\mathbb{P}}_{X_0, X_1}(x_0, x_1) \mathrm{d}\mathbb{S}_{t|X_0, X_1}(z_t|x_0, x_1), \tag{168}$$

incorporating, for $\lambda > 0$, the drift of the time-reversal of the partially pinned process.

## C  Broader impact

Fractional Diffusion Bridge Models (FDBM) introduce non-Markovian stochastic dynamics into generative modeling, enabling the learning of long-range dependencies and memory effects observed in real systems. This is largely a theoretical contribution. This framework can benefit scientific domains where temporal correlations are fundamental, including molecular design, protein dynamics, materials discovery, and biological simulation, by improving the physical fidelity of generative models.

By bridging stochastic physics and machine learning, FDBM contributes to more interpretable and physically grounded generative tools, potentially reducing experimental costs and accelerating discovery. Nevertheless, as with any generative model, misuse for fabricating deceptive or unsafe data is possible. To mitigate this, our open-source release emphasizes research and educational use with clear documentation.

## D  Related work

**Diffusion based generative modeling**. Diffusion models [61, 62] have achieved remarkable success in generative modeling, setting state-of-the-art performance across image [52, 63] and molecule generation [64, 65]. They have had a major impact across a broad range of domains, including materials and drug discovery [66, 67], realistic audio synthesis [68, 69], 3D object and texture generation [4, 5, 70], medical imaging [71, 72], aerospace design [73], and DNA sequence modeling [74, 75]. Building on the seminal contribution of Song et al. [3], who introduced a continuous-time framework for score-based diffusion models via stochastic processes with an exact reverse-time model, a large body of subsequent work has expanded this perspective by analyzing its properties [76–78] and generalizing it to subspaces [79], Riemannian manifolds [80, 81], alternative stochastic dynamics such as non-linear drifts [82], general corruptions [12] and reflecting processes [83, 84], as well as by learning the drift of the forward process [85]. A unifying perspective on diffusion and diffusion bridge models has been proposed through mixtures of diffusion bridges [21], optimal control [86], and the generalized Schrödinger bridge problem [59, 87], with applications to sampling from unnormalized densities. Recent methods incorporated non-Gaussian priors, as well as non-Gaussian conditioning into diffusion modeling and considered the *boundary value problem* through diffusion bridges [88–90]. In line with our research, non-standard noise sources for continuous-time diffusion models have been explored, including heavy-tailed Lévy processes [14, 15], and non-Markovian fractional Brownian motion [11, 16, 17].

**Fractional Brownian motion in machine learning**. Memory-aware fractional Brownian motion has been employed in machine learning for generative modeling [11, 16, 17, 91], variational inference [26], and stochastic optimal control [45]. Our work builds directly on the Markovian approximation of fractional Brownian motion (MA-fBM) introduced by Harms and Stefanovits [25] and further refined through the derivation of optimal approximation coefficients by Daems et al. [26]. Daems et al. [26] demonstrate how variational inference can be performed for SDEs driven by MA-fBM, a framework later enhanced by Daems et al. [45] using techniques from stochastic optimal control. Nobis et al. [16] introduced a continuous-time score-based diffusion model driven by MA-fBM, which Liang et al. [17] extended to protein generation.

**The Schrödinger bridge problem**. The Schrödinger bridge problem [39–42] is a stochastic optimal control formulation that serves as an entropy-regularized generalization of the optimal transport problem on path spaces. It offers a principled alternative to Diffusion Models[3, 61, 62] and Flow Matching approaches [30, 92], by directly interpolating between marginal distributions via maximum entropy dynamics [19, 20]. While the algorithm proposed by Bortoli et al. [19] was based on Iterative Proportional Fitting (IPF) [93–97], Shi et al. [34] and Peluchetti [33] concurrently introduced Iterative Markovian Fitting (IMF) for Brownian-driven diffusion processes, which directly learns the time-dependent drift of a stochastic process solving an SDE. Specifically, Shi et al. [34] considered a scalar, positive diffusion function, whereas Peluchetti [33] formulated their approach for matrix-valued diffusion functions that may depend on the state of the process. For unpaired data translation, we build upon the framework of Bortoli et al. [27], where IMF is extended to $\alpha$-IMF, an online variant of IMF, summarized in detail in Section E. See also Peyré and Cuturi [38] for a comprehensive overview of optimal transport methods.

**Stochastic bridges for paired data translation**. Recent studies have extended stochastic bridges to the paired data settings. Liu et al. [30] proposed a structured diffusion framework for constrained domains, alongside a task-specific training loss. Liu et al. [36] propose a generative bridge model for image-to-image translation and Somnath et al. [31] introduced aligned diffusion bridges that interpolate between matched samples and evaluated the method on toy datasets, cell differentiation, and predicting conformational changes in proteins. Bortoli et al. [32] identified limitations in preserving the coupling of the training data in the approach of Somnath et al. [31] and Liu et al. [36], which they resolved by augmenting the drift of the learned process with the starting value. Our framework FDBM in the paired setting is built upon the repository provided by Somnath et al. [31][4], including the training setup, model architectures, data visualization, and all used datasets. Conceptually, we adopt the viewpoint of Bortoli et al. [32], providing the initial value to the neural network, approximating the drift, at all points in time.

## E   The Schrödinger bridge problem for unpaired data translation

In this section, we summarize the Schrödinger Bridge Flow (SBFlow) introduced by Bortoli et al. [27], which our FDBM builds upon for unpaired data translation. Adopting the perspective of Entropic Optimal Transport (EOT) and assuming unpaired data samples from the distributions $\Pi_0$ and $\Pi_1$ on $\mathbb{R}^d$, Bortoli et al. [19] seek to find the coupling distribution

$$\Pi^\star = \arg \min_{\Pi \in \mathcal{P}(\mathbb{R}^d \times \mathbb{R}^d)} \left\{ \int_{\mathbb{R}^d \times \mathbb{R}^d} \frac{1}{2} ||x_0 - x_1||^2 \mathrm{d}\Pi(x_0, x_1) - \varepsilon \mathcal{H}(\Pi) \right\}, \tag{169}$$

where the differential entropy $\mathcal{H}(\Pi)$ can be controlled by a regularization parameter $\varepsilon > 0$, and $\mathcal{P}(\mathbb{R}^d \times \mathbb{R}^d)$ is the set of coupling probability measures on $\mathbb{R}^d \times \mathbb{R}^d$. Adopting EOT rather than optimal transport (OT)—restored when $\varepsilon = 0$—allows a degree of regularizing stochasticity when solving for a transport map. The formulation of EOT in eq. (169) can be understood as a *static* version of the dynamic formulation of the Schrödinger bridge problem described eq. (4). We refer the reader to Léonard [42] for a detailed discussion of the relation between the static and dynamic Schrödinger bridge Problem

$$\mathbb{T}^{SB} = \arg \min_{\mathbb{T} \in \mathcal{P}(\mathcal{C}^d)} \left\{ D_{\mathsf{KL}}(\mathbb{T}|\mathbb{Q}) \; ; \; \mathbb{T}_0 = \Pi_0, \mathbb{T}_1 = \Pi_1 \right\}, \tag{170}$$

where we now seek a path measure $\mathbb{P}^{SB}$ with marginal distributions $\Pi_0$ and $\Pi_1$. The reference path measure $\mathbb{Q}$ in Bortoli et al. [27] is associated with a scaled Brownian motion $\sqrt{\varepsilon} B_t$ with $\varepsilon > 0$. Remarkably, under some assumptions, eq. (169) and eq. (170) share the same unique solution [42] for the coupling distribution in the sense that $\mathbb{T}^{SB} = \Pi_{0,1}^\star$.

The difficulty of solving eq. (170) stems from the need to optimize over the infinite-dimensional space of path measures. Traditional approaches like Iterative Proportional Fitting (IPF) [93–97] become computationally costly in high dimensions as they require simulating complex conditioned processes. The Iterative Markov Fitting (IMF), concurrently introduced by Peluchetti [33], Shi et al. [34], bypasses this bottleneck by operating directly on learning the time-evolving drift of a stochastic process solving an SDE. It operates by iteratively alternating between fitting a forward-time process and a backward-time process. Bortoli et al. [27] introduced an online version of IMF called $\alpha$-IMF that is described in the following.

$\alpha$-IMF, much like IMF, builds on reciprocal projections and Markovian projections [33, 34, 47]. These projections accomplish two key objectives. Projections to the reciprocal class ensure matching terminal distributions $\Pi_0$, $\Pi_1$, while Markovian projections ensure that the drift of the learned process depend only in expectation on $X_1$ and that the learned process satisfies an SDE. A path measure $\mathbb{P}$ is in the reciprocal class of some other path measure $\mathbb{Q}$ if

$$\mathbb{P} = \int_{\mathbb{R}^d \times \mathbb{R}^d} \mathbb{Q}(\cdot | x_0, x_1) \mathrm{d}\mathbb{P}_{0,1}(x_0, x_1) =: \mathbb{P}_{0,1} \mathbb{Q}_{|0,1}. \tag{171}$$

Now, when we assume that $\mathbb{Q}$ is induced by the scaled Brownian Motion $(\sqrt{\varepsilon} B_t)_{t \in [0,1]}$, then following Bortoli et al. [27, Definition 2.2] and Shi et al. [34, Definition 1] the Markovian projection of the path measure $\Pi$ is the Markovian path measure $\mathbb{M}$ associated with $X'$ solving

$$\mathrm{d}X'_t = v_t(X'_t)\mathrm{d}t + (\sqrt{\varepsilon}\mathrm{d}B_t), \quad X'_0 = X_0 \tag{172}$$

---

[4]https://github.com/vsomnath/aligned_diffusion_bridges

and the intractable drift function

$$v_t(x_t) = \frac{\mathbb{E}_{\Pi_{1|t}}[X_1|X_t = x_t] - x_t}{1 - t} \tag{173}$$

being learned by a neural network. In the following we will refer to the projection for the reciprocal class (see eq. (171)) of $\mathbb{Q}$ as $\mathrm{proj}_{\mathbb{Q}}(\cdot)$ and to the Markovian projection associated to the SDE in eq. (172) as $\mathrm{proj}_{\mathbb{M}}(\cdot)$. Bortoli et al. [27] consider IMF from the perspective of a *flow of path measures* $(\mathbb{P}^s, \hat{\mathbb{P}}^s)_{s\geq 0}$, describing Markovian and reciprocal class states respectively

$$\hat{\mathbb{P}}^0 = (\Pi_0 \otimes \Pi_1)\mathbb{Q}_{|0,1}, \tag{174}$$

$$\mathbb{P}^s = \mathrm{proj}_{\mathbb{M}}(\hat{\mathbb{P}}^s), \tag{175}$$

$$\partial\hat{\mathbb{P}}^s = \mathrm{proj}_{\mathbb{Q}}(\mathrm{proj}_{\mathbb{M}}(\hat{\mathbb{P}}^s)) - \hat{\mathbb{P}}^s, \tag{176}$$

where the only fixed point w.r.t. the vector field of the *flow of path measures* in (176) is the Schrödinger bridge. Finaly Bortoli et al. [27] propose a novel discretization approach

$$\hat{\mathbb{P}}^{s+1} = (1 - \alpha)\hat{\mathbb{P}}^s + \alpha \, \mathrm{proj}_{\mathbb{Q}}(\mathbb{P}^s), \tag{177}$$

which converges to the Schrödinger bridge [27, Theorem 3.1] and recovers IMF for $\alpha = 1$.

To counteract error accumulation issues, a bidirectional online procedure can be implemented to achieve $\alpha$-IMF. This involves concurrently training two models or a single direction-conditioned model: one approximating the forward drift for the $\Pi_0 \to \Pi_1$ process, and another approximating the backward drift for the $\Pi_1 \to \Pi_0$ process. Bortoli et al. [27] first pretrain a bridge matching model $v^\theta$ for both directions following DSBM [34] w.r.t. eq. (174), where samples are drawn from $\Pi_0 \otimes \Pi_1$, such that $v_t^\theta(x) \approx (\mathbb{E}_{\hat{\mathbb{P}}_{1|t}^0}[X_1|X_t = x] - x)/(1 - t)$. Furthermore, they propose a bidirectional loss formulation of the online procedure of $\alpha$-DSBM, where samples are drawn from the opposing directional processes

$$\mathcal{L}_t(v_t^{\rightarrow}, v_t^{\leftarrow}, \mathbb{P}^{\rightarrow}, \mathbb{P}^{\leftarrow}) = \int_{(\mathbb{R}^d)^3} \left|\left| v_t^{\rightarrow}(x_t) - \frac{x_1 - x_t}{1 - t} \right|\right|^2 \mathrm{d}\mathbb{P}_{0,1}^{\leftarrow}(x_0, x_1)\mathrm{d}\mathbb{Q}_{t|0,1}(x_t|x_0, x_1)$$
$$+ \int_{(\mathbb{R}^d)^3} \left|\left| v_{1-t}^{\leftarrow}(x_t) - \frac{x_1 - x_t}{t} \right|\right|^2 \mathrm{d}\mathbb{P}_{0,1}^{\rightarrow}(x_0, x_1)\mathrm{d}\mathbb{Q}_{t|0,1}(x_t|x_0, x_1)$$
$$\tag{178}$$

with associated forward and backward SDEs following the Markovian projection, as described by eq. (172), in respective directions.

## F  Implementation details for paired data translation

In the followign we will provide implementation details for all experiments with paired data translations. We emphasize here again that the implementation of FDBM in the paired setting is built upon the repository provided by Somnath et al. [31][5], including the training setup, model architectures, data visualization, and all used datasets.

### F.1  Network architectures

**Toy experiments and cell differentiation**. For SBALIGN, we use two multilayer perceptrons (MLPs) to approximate the drift $b^\theta$ and Doobs $h$-score $m^\phi$. For ABM and FDBM, we use only the MLP employed in SBALIGN to approximate the drift $b^\theta$, but the initial state $x_0$ is additionally provided to the network by concatenating it with the input, following Bortoli et al. [32]. This setup is used in the experiments shown in Figures 2 and 7, on the *Moons* and *T-shape* datasets, as well as in the cell differentiation task, with the respective number of parameters reported in Table 4.

**Conformational changes in proteins**. We use the GNN architecture from Somnath et al. [31]. However, following Bortoli et al. [32], the initial state $x_0$ is additionally provided to the network by concatenating it with the input. See Table 4 for a comparison of the number of parameters.

---

[5] https://github.com/vsomnath/aligned_diffusion_bridges

### F.2   Training & Sampling

**Toy experiment**. We follow precisely the training of Somnath et al. [31]. For sampling, we use 100 steps of the Euler–Maruyama method and generate a single trajectory for each test starting point. This procedure is used both for calculating the WSD and for the visualization in Figure 3, whereas Somnath et al. [31, Figure 2] report trajectories averaged over multiple trials.

**Cell differentiation**. We follow precisely the training and sampling setup of Somnath et al. [31].

**Conformational changes in proteins**. The results reported in Table 1 were obtained by averaging over 5 training trials, each run for 300 epochs, and performing one sampling trial per trained model, generating a single path over 100 time steps. The remaining training set-up closely follows Somnath et al. [31]. We use the AdamW [98] optimizer with an initial learning rate of 0.001 and a training batch size of 2. During validation, inference is performed using the exponential moving average of the model parameters, which is updated at every optimization step with a decay rate of 0.9. After each epoch, we simulate trajectories on the validation set and compute the mean RMSD. The model achieving the lowest mean RMSD on the validation set is selected for final evaluation on the test set. We observe that the best model was saved for ABM and FDBM towards the end of training, indicating that a longer training could further improve the overall results.

### F.3   Compute

The toy experiments were run locally on a CPU and completed within minutes. Each trial of 300 training epochs for the protein conformational change task was completed within 24 hours on a single NVIDIA A100 GPU (40 GB VRAM).

### F.4   Datasets

**Toy datasets**. The *Moons* dataset is obtained by generating two moons to produce samples from $\Pi_0$ and then rotating them clockwise 90 degrees around the center to produce samples from $\Pi_1$. The *T-Shape* dataset is produced by a bi-modal distribution, where $\Pi_0$ is supported on two of the four extremes of an imaginary T-shaped area. The target distribution $\Pi_0 1$ is created by shifting $\Pi_0$ to the opposite side. The rotations and shifts imply paired data, since there is a one-to-one correspondence between samples in $\Pi_0$ and $\Pi_1$. For a detailed description of the datasets, we refer the reader to Somnath et al. [31], who designed both datasets. See Figure 7 for a visualization of the dataset marginals.

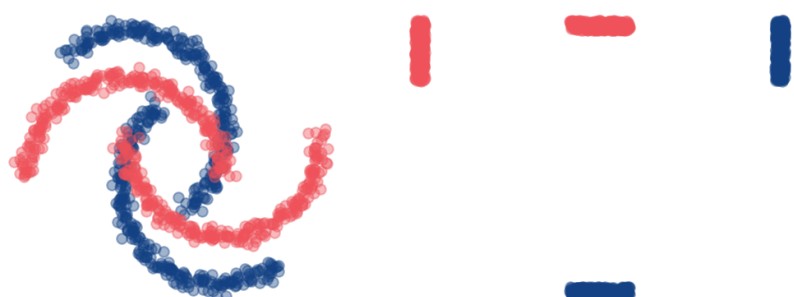

Figure 7: Marginals of the *Moons* dataset and the *T-shape* dataset introduced by Somnath et al. [31].

**Cell differentiation**. We use a dataset of genetically traced cells during the process of blood formation, created by Weinreb et al. [99] and curated by Somnath et al. [31]. The dataset consists of two snapshots: one recorded on day 2, when most cells remain undifferentiated, and another on day 4, which includes a diverse set of mature cell types. For a detailed descirption of the dataset we refer the reader to Somnath et al. [31].

**Conformational changes in proteins**. We use the curated subset from Somnath et al. [31] of the D3PM dataset [49], which focuses on structure pairs with $C_\alpha$ RMSD $> 3$Å . This subset initially comprises of $2,370$ ligand-free (apo) - ligand-bound (holo) pairs. To ensure high-quality alignment,

Somnath et al. [31] compute the $C_\alpha$ RMSD between pairs of proteins common residues superimposed using the Kabsch algorithm [100] and retain only those examples where the computed RMSD closely matches the original D3PM value. This results in a cleaned dataset of $1,591$ pairs, which is split into training, validation, and test sets of $1,291/150/150$ examples, respectively. All structures are Kabsch-superimposed to remove global translational and rotational artifacts, ensuring that the model focuses solely on internal conformational changes. For more details see Somnath et al. [31].

## G  Implementation details for unpaired data translation

In the followign we will provide implementation details for all experiments with upaired data translations on AFHQ [51].

### G.1  Experiments on unpaired data translation

**Network architecture**. The Diffusion Transformer (DiT) [55] is a scalable architecture that adapts the Vision Transformer (ViT) [101] for generative modeling with diffusion processes. Unlike convolution-based U-Nets commonly used in image diffusion models, a DiT model treats denoising as a sequence modeling task by operating directly on (latent) patches of an image, capturing long-term dependencies via Attention [102]. DiT architectures are grouped into small (DiT-S), base (DiT-B), large (DiT-L), and extra large (DiT-XL) variants, where Peebles and Xie [55] observed diminishing returns after scaling from DiT-L to DiT-XL. Notably, Peebles and Xie [55] show that the model scales with FLOPs, rather than parameter size. Therefore, a smaller model with more tokens (i.e., smaller patches) can achieve identical performance to a larger model with fewer patches. Following this finding, we used the models with the most tokens for respective parameter sizes. Hence, we selected the variants DiT-B/2 and DiT-L/2—where the "/2" indicates a patch size of $2 \times 2$ for respective tokens—as suitable backbone architectures for all experiments on imaging data.

**Training & sampling parameterization**. We used the same training and sampling parameterizations for all datasets and experiments. Parameterizations for DiT-B/2 and DiT-L/2 were kept identical. See Table 3 for detailed parameterizations of all experiments.

| Model | Optimizer | Learning Rate | EMA Rate | Linear Warmup | Cosine Decay | Online Finetuning | Euler–Maruyama Steps | Parameters |
|---|---|---|---|---|---|---|---|---|
| DiT-B/2 | lion [103] | 0.0001 | 0.999 | 10K | 90K | 4K | 200 | 130M |
| DiT-L/2 | lion [103] | 0.0001 | 0.999 | 10K | 90K | 4K | 200 | 458M |

Table 3: Hyperparameters for experiments with Diffusion Transformers.

**Compute**. Experiments were conducted in single- and mutli-GPU settings, using full precision (FP32) for all runs. Computation times are denoted in an equivalent of A100 GPU (40GB VRAM) hours, as a common reference for scientific compute time. All pretrainings of 100K steps for the AFHQ-32 and AFHQ-256 datasets were completed in 16 hours (A100) for the DiT-B/2 variant and 54 hours (A100) for the DiT-L/2 variant. The online finetunings of 4K steps were completed in 12 hours (A100) for the DiT-B/2 variant and 43 hours (A100) for the DiT-L/2 variant. Samplings experiments were completed in 0.5 hours (A100) for the DiT-B/2 variant and 1.5 hours (A100) for the DiT-L/2 variant. All pretrainings of 100K steps for the AFHQ-512 datasets were completed in 256 hours (A100) for the DiT-L/2 variant. Samplings experiments were completed in 5 hours (A100) for the DiT-L/2 variant.

## H  Computational efficiency

**Number of learnable parameters**. We use the GNN architecture from Somnath et al. [31], but following Bortoli et al. [32] the initial state $x_0$ is additionally provided to the network by concatenating it with the input. Nevertheless, the GNN we use for ABM and FDBM has fewer parameters, since Somnath et al. [31] approximate two functions ($b_t$ and $\nabla_x \log h_t$) with a single GNN resulting in more parameters in the output layer. We emphasize that ABM and FDBM deploy the same model architecture and summarize the number of learnable parameters in Table 4. Throughout all unpaired

data translation experiments, we use the same model architecture for both SBFlow and FDBM with the same number of learnable parameters.

| # parameters per task | SBALIGN [31] | ABM [32] | FDBM |
|---|---|---|---|
| Coupling-preserving (Figure 2) | $58,692$ | $31,618$ | $31,618$ |
| Moons | $58,692$ | $31,618$ | $31,618$ |
| T-Shape | $19,204$ | $10,754$ | $10,754$ |
| Cell Differentiation | $310,372$ | $177,970$ | $177,970$ |
| Predicting Conformations | $545,220$ | $537,900$ | $537,900$ |

Table 4: Number of learnable parameters in SBALIGN, ABM, and FDBM.

**Runtime comparison**. We provide a runtime comparison of FDBM in the paired setting in Table 5. The runtime per training step is averaged over 1000 training steps, and the runtime to sample one conformation is averaged over the 150 test samples of the D3PM test set. Training times for ABM and FDBM are nearly identical and both outperform SBALIGN, which requires approximating two functions and thus involves a larger model. For sampling, ABM and FDBM again show an advantage over SBALIGN. FDBM requires, on average, only 0.0422 seconds more than ABM to sample a conformation over 100 Euler–Maruyama steps. This slight increase is due to simulating a higher-dimensional stochastic process. However, the effect is minor, as the dominant computational cost during sampling comes from forward passes through the GNN, which are identical for both ABM and FDBM. Throughout all unpaired data translation experiments, we use the same model architecture for both SBFlow and FDBM.

The differing components during training are the sampling from the (partially) pinned process and the loss computation, both showing nearly identical runtime in Table 6. The sampling algorithm of FDBM during inference is identical for the paired and unpaired settings. All computations of this section were performed on an NVIDIA A100 GPU (40 GB VRAM).

| Average Runtime [s] | SBALIGN[31] | ABM[32] | FDBM |
|---|---|---|---|
| Training step | $0.0159 \pm 0.0075$ | $0.01438 \pm 0.0065$ | $0.01412 \pm 0.0063$ |
| Sampling one conformation over 100 sampling steps | $0.7078 \pm 0.3409$ | $0.6424 \pm 0.2992$ | $0.6846 \pm 0.3021$ |

Table 5: Runtime comparison of SBALIGN, ABM, and FDBM.

| Average Runtime [s] | SBFlow[27] | FDBM |
|---|---|---|
| Sampling from (partially) pinned process | $0.0010 \pm 0.0002$ | $0.0011 \pm 0.0003$ |
| Calculation of loss term | $0.0132 \pm 0.0074$ | $0.0132 \pm 0.0018$ |

Table 6: Runtime comparison of SBFlow and FDBM. The runtimes are averaged over 1000 computations. All computations were performed on an NVIDIA A100 GPU (40 GB VRAM).

# I  Evaluation metrics

**Wasserstein distance**.  To measure the distance from the original data distribution from the predicted data distribution we use Wasserstein-1 distance [104]. The Wasserstein-1 distance between ground truth data distribution $p_t$ and sampled data distribution $p_s$ is defined as

$$W_1(p_t, p_s) = \inf_{\gamma \sim \Pi(p_t, p_s)} \mathbb{E}_{(x, \hat{x})}[||x - \hat{x}||]. \tag{179}$$

The lower the Wasserstein distance, the better are the distributions $p_t$ and $p_s$ aligned.

**Root Mean Square Deviation**.  Root mean square deviation of $C_\alpha$ atomic positions is a distance between two superimposed molecules/proteins. If $\mathbf{x}$ is an observed 3D structure/configuration of the

protein and $\hat{\mathbf{x}}$ is a predicted configuration of the protein then

$$\text{RMSD}(\mathbf{x}, \hat{\mathbf{x}}) = \sqrt{\frac{1}{n} \sum_{i=1}^{n} ||x_i - \hat{x}_i||^2}. \tag{180}$$

The lower the RMSD, the lower their L2-distance w.r.t. some unit of measure. In our example, the unit of the measure is Angstrom, Å.

**Fréchet Inception Distance (FID)**. The Fréchet Inception Distance (FID) [53] measures the distance between the feature distributions of real and generated images, typically using embeddings from a pretrained Inception network. Given the empirical mean and covariance of real images $(\mu_r, \Sigma_r)$ and generated images $(\mu_g, \Sigma_g)$ in this feature space, FID is defined as

$$\text{FID}(X, Y) = ||\mu_r - \mu_g||_2^2 + \text{Tr}\left(\Sigma_r + \Sigma_g - 2(\Sigma_r \Sigma_g)^{1/2}\right), \tag{181}$$

where $X$ and $Y$ denote the sets of real and generated images, respectively. The first term captures differences in mean features (style/content shifts), while the second term accounts for differences in variability. FID is widely used in image generation and style transfer as it correlates well with human judgment of realism and diversity.

**Learned Perceptual Image Patch Similarity (LPIPS)**. The LPIPS metric [54] quantifies perceptual similarity between two images by comparing deep features extracted from a pretrained network (e.g., VGG, AlexNet). Let $x$ and $y$ be two images. The LPIPS score is computed by comparing their normalized feature maps $\hat{f}_l(x), \hat{f}_l(y)$ at multiple layers $l$

$$\text{LPIPS}(x, y) = \sum_{l=1}^{L} \frac{1}{H_l W_l} \sum_{h=1}^{H_l} \sum_{w=1}^{W_l} \left\|\mathbf{w}_l \odot \left(\hat{f}_l(x)_{h,w} - \hat{f}_l(y)_{h,w}\right)\right\|_2^2, \tag{182}$$

where $\mathbf{w}_l$ are learned per-channel weights, and $H_l, W_l$ denote the spatial dimensions of layer $l$. LPIPS has been shown to align well with human perceptual similarity judgments, making it valuable for evaluating and training generative models, especially in style transfer tasks where pixel-wise metrics fall short.

## J  Cell Differentiation

| Methods | MMD $\downarrow$ | $W_\varepsilon \downarrow$ | $\ell_2(\text{PS}) \downarrow$ | RMSD $\downarrow$ |
|---|---|---|---|---|
| FBSB$^*$ | 1.55e-2 | 12.50 | 4.08 | 9.64e-1 |
| FBSB WITH SBALIGN$^*$ | **5.31e-3** | 10.54 | 0.99 | 9.85e-1 |
| SBALIGN$^*$ | 1.07e-2 | 11.11 | 1.24 | 9.21e-1 |
| ABM | 4.10e-2 | 9.50 | 0.89 | 8.72e-1 |
| FDBM($\bar{H} = 0.3$) (**ours**) | 5.34e-2 | **9.32** | 0.89 | 8.11e-1 |
| FDBM($H = 0.4$) (**ours**) | 4.52e-2 | 9.35 | **0.85** | 8.21e-1 |

Table 7: Comparison of performance on the cell differentiation task. Results marked with an asterisk ($*$) are obtained from Somnath et al. [31].

We evaluate FDBM on the cell differentiation task introduced by Somnath et al. [31]. We fix the diffusion coefficient to $\sqrt{\varepsilon} = 1$ across all retrained methods SBALIGN, ABM and FDBM. All scores are averaged over 10 training trials and 10 sampling trials for each trained model. We follow the approach of Somnath et al. [31] and average for each prediction over 20 sampled paths. This task allows us to assess FDBM for cell differentiation prediction on both the distributional quality and perturbation accuracy of the generated data using distributional metrics such as Wasserstein-2 distance ($W_\varepsilon$) [105] and kernel maximum mean discrepancy (MMD) [106], as well as the Perturbation signature $\ell_2(\text{PS})$ [107] and RMSD. The dataset consists of two snapshots: one recorded on day 2, when most cells remain undifferentiated, and another on day 4, which includes a diverse set of mature cell types. We assess the performance of FDBM against forward-backward Schrödinger bridge models (FBSB) [108], SBALIGN, and ABM. Consistent with our findings on protein conformational changes, we observe in Table 7 that ABM shows superior performance compared to all other Brownian baselines in all metrics except MMD. FDBM achieves the best performance in the rough regime ($H = 0.3$ and $H = 0.4$), with slightly better average $W_\epsilon$ and RMSD scores, while ABM remains superior in terms of MMD.

## K  Extended Experiments

In the following we provide more details on the results reported in the main paper. Detailed scores of all ablations are listed in Tables 12 to 14. Additional evaluations of AFHQ-256 Dogs ↔ Wild and Dogs ↔ Cats are listed in Table 15. Additional visual examples for AFQH-512 samples are displayed in Figures 10 and 11 and for AFHQ-256 in Figures 8 and 9. Additional results for the D3PM dataset are listed in Table 10, as well as additional experiments with toy data in Tables 8 and 9.

Table 8: Average Wasserstein distance over 10 runs between samples generated by the Brownian-driven baseline and the target distribution, for varying diffusion coefficient $\sqrt{\varepsilon}$.

| BM driven | Wasserstein Distance ↓ | | | | | | | |
|---|---|---|---|---|---|---|---|---|
| | $\sqrt{\varepsilon}=1.0$ | $\sqrt{\varepsilon}=0.8$ | $\sqrt{\varepsilon}=0.6$ | $\sqrt{\varepsilon}=0.4$ | $\sqrt{\varepsilon}=0.2$ | $\sqrt{\varepsilon}=0.1$ | $\sqrt{\varepsilon}=0.05$ | $\sqrt{\varepsilon}=0.01$ |
| **Moons** | 0.020±0.008 | **0.015**±0.005 | 0.019±0.006 | 0.025±0.003 | 0.033±0.011 | 0.206±0.008 | 0.121±0.016 | 0.206±0.019 |
| **T-shaped** | 0.395±0.045 | 0.346±0.029 | 0.251±0.008 | 0.154±0.010 | **0.082**±0.028 | 0.178±0.049 | 0.529±0.007 | 0.570±0.092 |

Table 9: Wasserstein distance (10 runs average) between generated samples and target distribution.

| | Wasserstein Distance ↓ | | | | | | |
|---|---|---|---|---|---|---|---|
| | $H=0.8$ | $H=0.7$ | $H=0.6$ | ABM [32] | $H=0.4$ | $H=0.3$ | $H=0.2$ |
| **Moons** | 0.017±0.002 | **0.012**±0.002 | **0.012**±0.003 | 0.015±0.019 | 0.029±0.006 | 0.033±0.008 | 0.048±0.016 |
| **T-shaped** | 0.082±0.043 | 0.091±0.041 | 0.083±0.031 | 0.082±0.028 | 0.068±0.015 | 0.062±0.013 | **0.048**±0.039 |

Table 10: Ablation of the diffusion coefficient $\sqrt{\varepsilon}$ of our Brownian driven baseline ABM [32]. Additionally compared to the scores reported in Somnath et al. [31].

| | RMSD(Å) | | | % RMSD(Å) $< \tau$ | | |
|---|---|---|---|---|---|---|
| **D3PM Test Set [31]** | Median | Mean | Std | $\tau=2$ | $\tau=5$ | $\tau=10$ |
| EGNN [31, 109] | 19.99 | 21.37 | 8.21 | 1% | 1% | 3% |
| SBALIGN$_{(10,10)}$ [31] | 3.80 | 4.98 | 3.95 | 0% | 69% | 93% |
| SBALIGN$_{(100,100)}$ [31] | 3.81 | 5.02 | 3.96 | 0% | 70% | 93% |
| ABM($\varepsilon=1.0$) [32] (1 trial) | 3.14 | 4.11 | 3.32 | 1% | 79% | 97% |
| ABM($\varepsilon=0.8$) [32] (1 trial) | 2.68 | 3.93 | 3.39 | 23% | 79% | 96% |
| ABM($\varepsilon=0.6$) [32] (1 trial) | 2.47 | 3.65 | 3.59 | 35% | 85% | 97% |
| ABM($\varepsilon=0.4$) [32] (1 trial) | 2.47 | 3.60 | 3.66 | 43% | 86% | 95% |
| ABM($\varepsilon=0.2$) [32] (1 trial) | **2.20** | **3.58** | 3.45 | **45%** | **81%** | **97%** |
| ABM($\varepsilon=0.1$) [32] (1 trial) | 2.70 | 3.67 | 3.54 | 43% | 83% | 96% |
| ABM($\varepsilon=0.05$) [32] (1 trial) | 2.69 | 3.59 | 3.83 | 35% | 82% | 95% |
| ABM($\varepsilon=0.01$) [32] (1 trial) | 2.96 | 3.78 | 4.08 | 30% | 77% | 93% |
| ABM($\varepsilon=0.2$) [32] (5 trials) | 2.40 | 3.49 | 3.54 | 43% | 84% | 96% |
| FDBM($H=0.4, \varepsilon=0.2$) (5 trials) | 2.24 | 3.39 | 3.57 | 45% | 84% | 97% |
| FDBM($H=0.3, \varepsilon=0.2$) (5 trials) | 2.33 | 3.42 | 3.42 | 43% | 85% | 97% |
| FDBM($H=0.2, \varepsilon=0.2$) (5 trials) | **2.12** | **3.34** | 3.59 | **48%** | **86%** | 96% |
| FDBM($H=0.1, \varepsilon=0.2$) (5 trials) | 2.20 | 3.44 | 3.57 | 46% | 83% | **97%** |

Table 11: Comparison of FID and LPIPS for AFHQ-512 across Cats → Wild and Wild → Cats translation tasks.

| AFHQ-512 | cats → wild | wild → cats |
|---|---|---|
| | FID ↓ | FID ↓ |
| SBFlow | 17.79 ± 0.66 | **24.17 ± 0.81** |
| FDBM (H=0.4) | **14.27 ± 0.86** | 30.11 ± 0.75 |

Table 12: Pretraining ablation for entropic regularization $\varepsilon$ of the SBFlow [27] baseline.

(a) AFHQ-32 with DiT-B/2.

| Method | $\varepsilon$ | cats $\rightarrow$ wild | | cats $\leftarrow$ wild | |
|---|---|---|---|---|---|
| | | FID ↓ | LPIPS ↓ | FID ↓ | LPIPS ↓ |
| SBFlow | 0.75 | 161.95 ±2.19 | 0.159 ±0.002 | 138.20 ±2.27 | 0.135 ±0.001 |
| SBFlow | 1 | **59.04** ±1.14 | 0.104 ±0.001 | **74.36** ±1.02 | 0.151 ±0.001 |
| SBFlow | 1.125 | 77.24 ±0.85 | 0.106 ±0.001 | 77.90 ±1.40 | 0.163 ±0.001 |
| SBFlow | 1.25 | 96.66 ±1.37 | 0.110 ±0.000 | 88.77 ±1.16 | 0.172 ±0.001 |

(b) AFHQ-256 with DiT-B/2.

| Method | $\varepsilon$ | cats $\rightarrow$ wild | | cats $\leftarrow$ wild | |
|---|---|---|---|---|---|
| | | FID ↓ | LPIPS ↓ | FID ↓ | LPIPS ↓ |
| SBFlow | 0.75 | 42.67 ±0.73 | 0.659 ±0.001 | 46.42 ±0.89 | 0.588 ±0.001 |
| SBFlow | 1 | **15.67** ±0.65 | 0.578 ±0.002 | **30.75** ±0.88 | 0.594 ±0.001 |
| SBFlow | 1.125 | 33.46 ±1.25 | 0.592 ±0.001 | 37.36 ±0.93 | 0.609 ±0.002 |
| SBFlow | 1.25 | 54.05 ±1.10 | 0.623 ±0.001 | 48.63 ±1.16 | 0.629 ±0.001 |

Table 13: Pretraining ablation for hurst index $H$ related parameterization of our method. $K = 5$ was fixed for all experiments. The best results and results where the mean is within the standard deviation of the best result are highlighted in boldface.

(a) AFHQ-32 with DiT-B/2 and $K = 5$ for FDBM (**ours**).

| Method | $H$ | cats $\rightarrow$ wild | | cats $\leftarrow$ wild | |
|---|---|---|---|---|---|
| | | FID ↓ | LPIPS ↓ | FID ↓ | LPIPS ↓ |
| FDBM | 0.9 | 47.03 ±1.53 | 0.099 ±0.001 | 52.38 ±1.06 | 0.155 ±0.002 |
| FDBM | 0.8 | 45.18 ±1.05 | 0.095 ±0.001 | 50.59 ±0.65 | 0.155 ±0.001 |
| FDBM | 0.7 | 48.36 ±0.92 | 0.095 ±0.001 | 51.65 ±0.74 | 0.156 ±0.002 |
| FDBM | 0.6 | 43.45 ±0.93 | 0.097 ±0.001 | 48.79 ±0.73 | 0.155 ±0.001 |
| FDBM | 0.5 | **40.21** ±1.18 | 0.097 ±0.001 | **45.74** ±0.69 | 0.154 ±0.002 |
| FDBM | 0.4 | 44.84 ±1.32 | 0.096 ±0.001 | 47.65 ±0.97 | **0.152** ±0.001 |
| FDBM | 0.3 | 58.27 ±0.97 | **0.090** ±0.001 | 54.89 ±0.78 | 0.153 ±0.001 |
| FDBM | 0.2 | 83.62 ±1.45 | – | 68.05 ±1.21 | – |
| FDBM | 0.1 | 131.04 ±1.51 | – | 123.20 ±1.92 | – |

(b) AFHQ-256 with DiT-B/2 and $K = 5$ for FDBM (**ours**).

| Method | $H$ | cats $\rightarrow$ wild | | cats $\leftarrow$ wild | |
|---|---|---|---|---|---|
| | | FID ↓ | LPIPS ↓ | FID ↓ | LPIPS ↓ |
| FDBM | 0.9 | 21.15 ±1.26 | **0.522** ±0.002 | 19.50 ±0.36 | **0.539** ±0.002 |
| FDBM | 0.8 | 19.65 ±1.39 | **0.523** ±0.001 | 19.88 ±0.63 | 0.542 ±0.002 |
| FDBM | 0.7 | 18.64 ±1.11 | 0.529 ±0.002 | 19.46 ±0.46 | 0.547 ±0.002 |
| FDBM | 0.6 | **16.77** ±0.71 | 0.530 ±0.002 | **19.14** ±0.38 | 0.551 ±0.001 |
| FDBM | 0.5 | **16.19** ±0.83 | 0.534 ±0.002 | 21.91 ±0.55 | 0.565 ±0.002 |
| FDBM | 0.4 | **17.02** ±0.78 | 0.542 ±0.002 | 24.32 ±0.63 | 0.577 ±0.001 |
| FDBM | 0.3 | 28.50 ±1.68 | 0.549 ±0.002 | 30.53 ±0.79 | 0.591 ±0.001 |
| FDBM | 0.2 | 59.83 ±2.36 | – | 37.17 ±0.62 | – |
| FDBM | 0.1 | 81.36 ±1.38 | – | 43.69 ±1.00 | – |

Table 14: Pretraining ablation for hurst index $H$ related parameterization of our method. $K = 5$ was fixed for all experiments. The best results and results where the mean is within the standard deviation of the best result are highlighted in boldface.

(a) AFHQ-32 with DiT-B/2, $\varepsilon = 1$ and $H = 0.5$.

| Method | $K$ | cats $\rightarrow$ wild | | cats $\leftarrow$ wild | |
|---|---|---|---|---|---|
| | | FID ↓ | LPIPS ↓ | FID ↓ | LPIPS ↓ |
| FDBM | 6 | 63.99 ±2.13 | **0.091** ±0.001 | 53.46 ±0.72 | **0.150** ±0.001 |
| FDBM | 5 | **40.21** ±1.18 | 0.097 ±0.001 | **45.74** ±0.69 | 0.154 ±0.002 |
| FDBM | 4 | **41.13** ±1.19 | 0.097 ±0.001 | 46.92 ±0.89 | 0.155 ±0.002 |
| FDBM | 3 | **41.32** ±1.38 | 0.096 ±0.001 | 47.82 ±0.82 | 0.154 ±0.001 |
| FDBM | 2 | 42.14 ±1.45 | 0.095 ±0.001 | **44.61** ±0.93 | 0.154 ±0.002 |
| FDBM | 1 | 41.15 ±1.11 | 0.097 ±0.001 | 46.64 ±0.97 | 0.153 ±0.002 |

(b) AFHQ-256 with DiT-B/2, $\varepsilon = 1$ and $H = 0.6$.

| Method | $K$ | cats $\rightarrow$ wild | | cats $\leftarrow$ wild | |
|---|---|---|---|---|---|
| | | FID ↓ | LPIPS ↓ | FID ↓ | LPIPS ↓ |
| FDBM | 6 | 59.08 ±1.95 | 0.547 ±0.001 | 38.54 ±1.11 | 0.592 ±0.002 |
| FDBM | 5 | **16.77** ±0.71 | 0.530 ±0.002 | **19.14** ±0.38 | **0.551** ±0.001 |
| FDBM | 4 | 18.67 ±0.75 | **0.528** ±0.002 | 19.77 ±0.52 | 0.555 ±0.002 |
| FDBM | 3 | 17.89 ±0.88 | **0.528** ±0.002 | 20.27 ±0.45 | 0.555 ±0.001 |
| FDBM | 2 | 19.20 ±1.04 | **0.527** ±0.002 | 21.74 ±0.48 | 0.555 ±0.002 |
| FDBM | 1 | 19.98 ±0.86 | 0.550 ±0.002 | 30.61 ±1.20 | 0.594 ±0.001 |

Table 15: Additional evaluations of AFHQ-256 Dogs $\leftrightarrow$ Wild and Dogs $\leftrightarrow$ Cats. The best results and results where the mean is within the standard deviation of the best result are highlighted in boldface.

(a) AFHQ-256 with DiT-B/2 and $K = 5$ for FDBM (**ours**).

| Method | $H$ | dogs $\rightarrow$ wild | | dogs $\leftarrow$ wild | |
|---|---|---|---|---|---|
| | | FID ↓ | LPIPS ↓ | FID ↓ | LPIPS ↓ |
| SBFlow | | 20.74 ± 0.64 | 0.53 ± 0.002 | 47.07 ± 0.80 | 0.56 ± 0.002 |
| FDBM | 0.9 | 20.37 ± 0.98 | **0.52 ± 0.002** | 43.11 ± 0.68 | **0.54 ± 0.002** |
| FDBM | 0.8 | 19.22 ± 0.83 | 0.53 ± 0.003 | 41.76 ± 0.84 | 0.55 ± 0.002 |
| FDBM | 0.7 | 18.11 ± 0.75 | 0.53 ± 0.002 | 40.08 ± 0.73 | 0.56 ± 0.002 |
| FDBM | 0.6 | 18.43 ± 0.72 | 0.53 ± 0.002 | 39.84 ± 0.89 | 0.57 ± 0.002 |
| FDBM | 0.5 | **14.74 ± 0.53** | 0.55 ± 0.002 | **37.68 ± 0.55** | 0.58 ± 0.002 |
| FDBM | 0.4 | 15.78 ± 0.85 | 0.56 ± 0.002 | 38.51 ± 0.64 | 0.59 ± 0.001 |

(b) AFHQ-256 with DiT-B/2 and $K = 5$ for FDBM (**ours**).

| Method | $H$ | dogs $\rightarrow$ cats | | dogs $\leftarrow$ cats | |
|---|---|---|---|---|---|
| | | FID ↓ | LPIPS ↓ | FID ↓ | LPIPS ↓ |
| SBFlow | | **18.38 ± 0.36** | 0.56 ± 0.002 | 50.08 ± 1.38 | 0.56 ± 0.002 |
| FDBM | 0.9 | 19.86 ± 0.67 | **0.55 ± 0.002** | 45.19 ± 0.74 | **0.55 ± 0.002** |
| FDBM | 0.8 | 20.14 ± 0.64 | 0.56 ± 0.002 | 45.08 ± 0.98 | **0.55 ± 0.001** |
| FDBM | 0.7 | 21.12 ± 0.62 | 0.56 ± 0.002 | 43.44 ± 0.82 | 0.56 ± 0.002 |
| FDBM | 0.6 | 22.06 ± 0.57 | 0.57 ± 0.001 | **41.35 ± 0.90** | 0.57 ± 0.002 |
| FDBM | 0.5 | 22.15 ± 0.75 | 0.58 ± 0.002 | 42.36 ± 0.77 | 0.58 ± 0.002 |
| FDBM | 0.4 | 24.79 ± 0.68 | 0.59 ± 0.002 | **41.21 ± 1.12** | 0.59 ± 0.002 |

**Input** Output **Input** Output

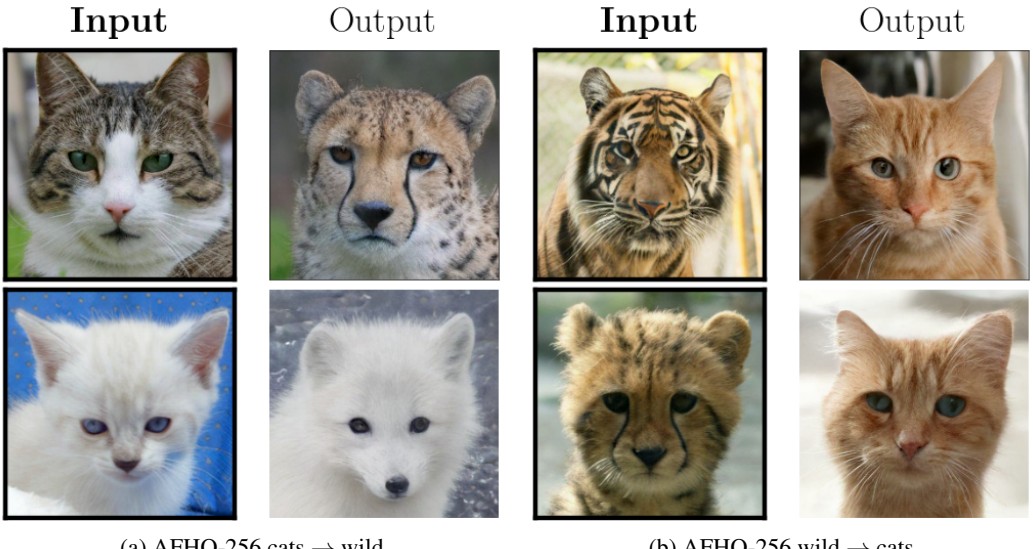

(a) AFHQ-256 cats → wild        (b) AFHQ-256 wild → cats

Figure 8: A detailed look at exemplary samplings with our method with H=0.4, K=5 for AFHQ-256.

**Input** Output **Input** Output **Input** Output **Input** Output

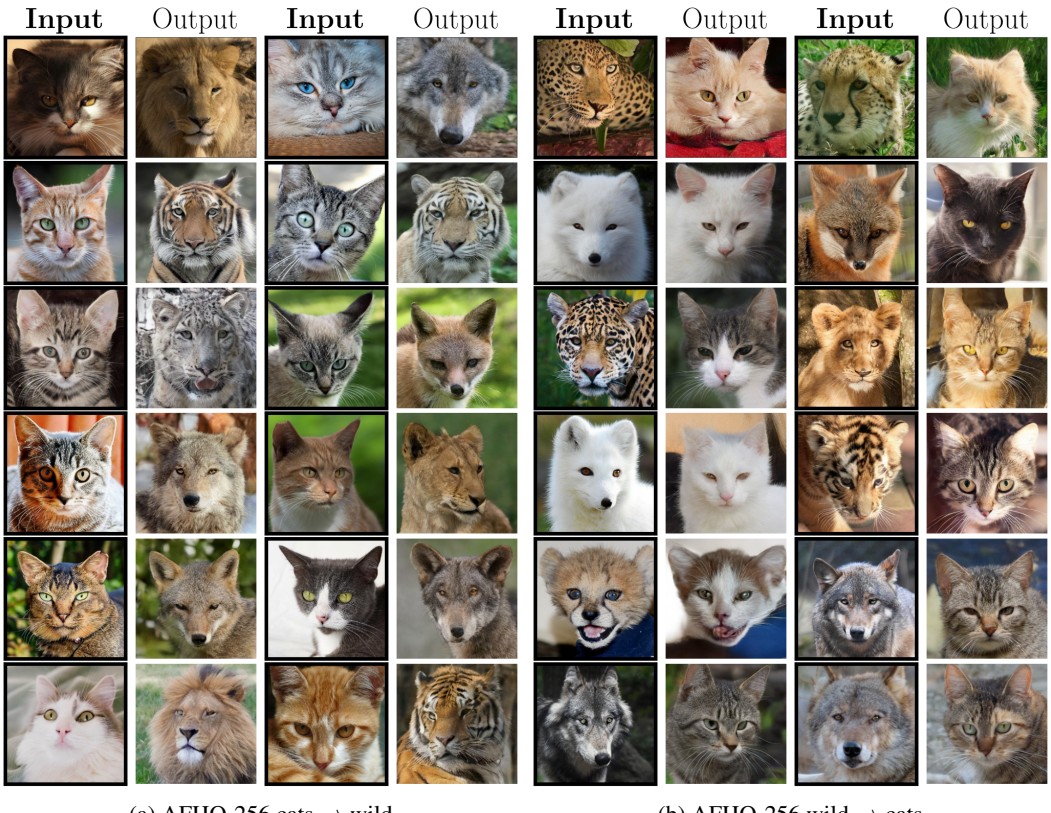

(a) AFHQ-256 cats → wild        (b) AFHQ-256 wild → cats

Figure 9: Overview of exemplary samplings with our method with H=0.4, K=5 for AFHQ-256.

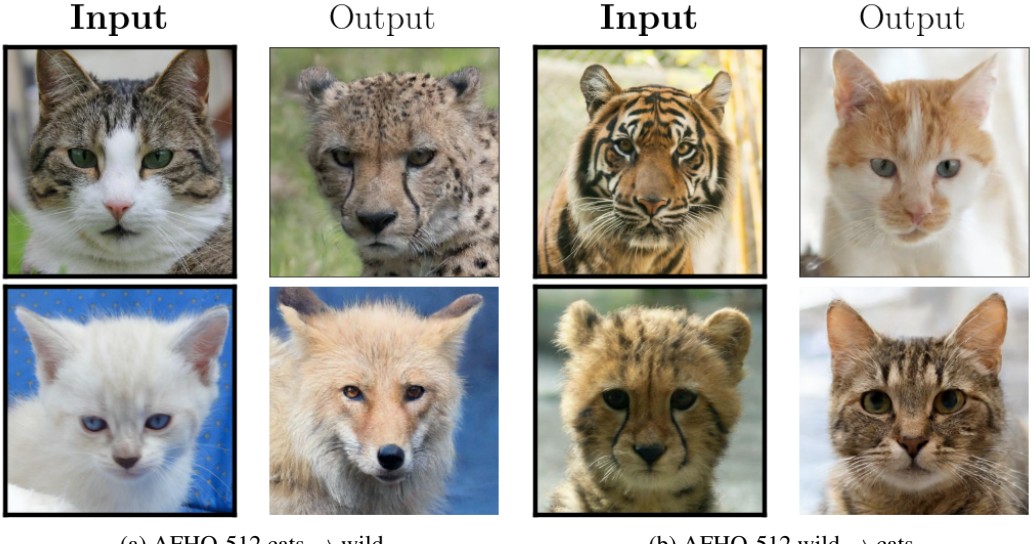

(a) AFHQ-512 cats → wild       (b) AFHQ-512 wild → cats

Figure 10: A detailed look at exemplary samplings with our method with H=0.4, K=5 for AFHQ-512.

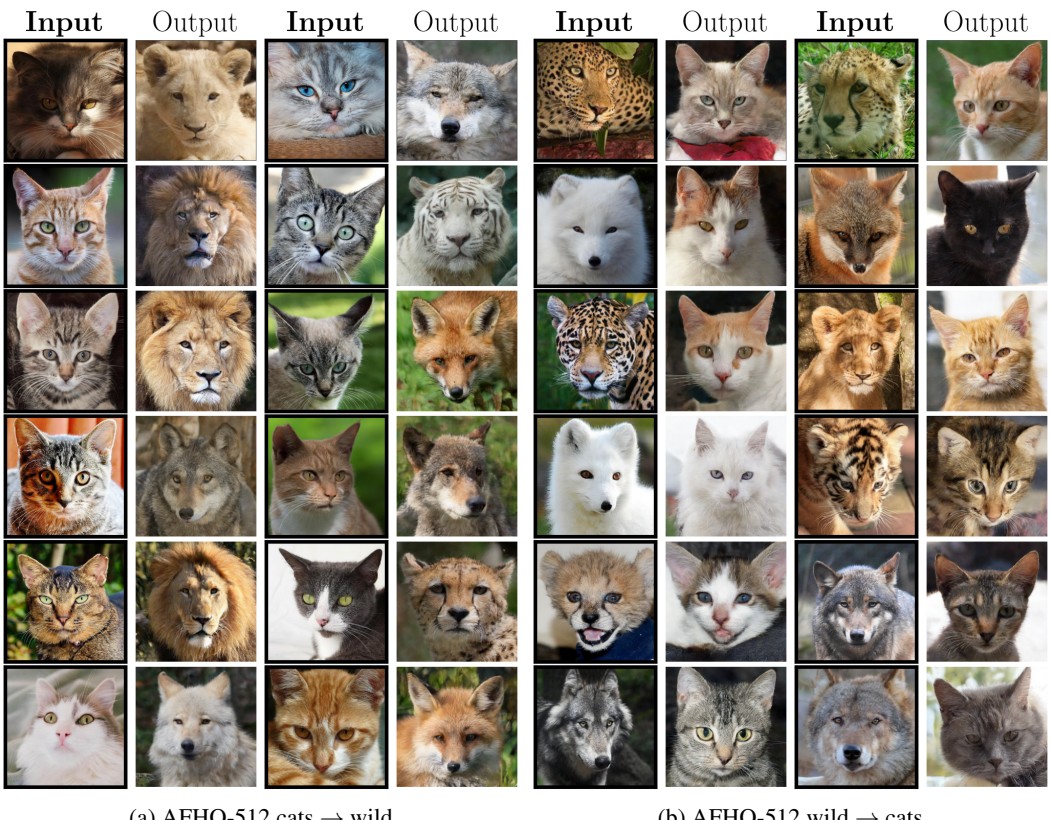

(a) AFHQ-512 cats → wild       (b) AFHQ-512 wild → cats

Figure 11: Overview of exemplary samplings with our method with H=0.4, K=5 for AFHQ-512.

