# OpenReview forum: "Fractional Diffusion Bridge Models"
_NeurIPS.cc/2025/Conference — NeurIPS 2025 poster_

### Official Review · Reviewer_fk33 · 2025-06-15

**Clarity:** 3
**Significance:** 3
**Originality:** 3
**Rating:** 3
**Confidence:** 3

**Summary:**

This paper introduces Fractional Diffusion Bridge Models (FDBM), a new class of generative stochastic bridge models that incorporate fractional Brownian motion (fBM) as a noise process. Traditional diffusion bridge models rely on Brownian motion, which is memoryless and fails to capture long-range dependencies and temporal correlations observed in real-world systems (e.g., protein conformational dynamics, visual transformations). FDBM addresses this by leveraging a Markovian approximation to fBM (MA-fBM) using a finite mixture of Ornstein–Uhlenbeck (OU) processes, which allows efficient simulation and learning while retaining non-Markovian statistical properties. The authors evaluate their method in both paired (protein conformational prediction) and unpaired (image translation) settings, achieving state-of-the-art performance compared to Brownian-based baselines across multiple metrics (RMSD, FID, LPIPS).

**Questions:**

- **Can you provide more intuition or visual illustrations** (e.g., in Figure 1) of how varying the Hurst index $H$ changes the nature of the generated paths? This would help readers unfamiliar with fBM better grasp its effect.


- **Does your method generalize to discrete-time settings or other noise models** (e.g., Lévy noise, jump diffusions)? If not, what are the main challenges?

- **Could you comment more on computational cost** (memory and runtime) of FDBM vs. Brownian models in high-dimensional settings, e.g., image synthesis?

- Can you give more experimental details of the protein task?

**Ethical Concerns:**

["NO or VERY MINOR ethics concerns only"]

**Final Justification:**

I appreciate the authors’ detailed rebuttal and the additional experiment on the cell differentiation task. That said, my main concerns remain. The paper is technically solid and clearly builds on a deep understanding of stochastic processes, but I found the core method difficult to follow, even after reading the responses. Key parts of the framework—like the role of the MA-fBM and how it integrates into the learning process—still lack accessible explanations, which makes the work hard to engage with for a broader ML audience.

The new temporal experiment helps but doesn’t add much in terms of empirical strength. The gains over existing baselines are small, and it's not enough to clearly justify the added complexity of the approach. I do think the idea is promising and worth pursuing, but in its current form, the paper doesn’t feel ready for publication at NeurIPS. For these reasons, I’m leaning toward a borderline reject.

**Limitations:**

Yes. The authors have an explicit section addressing the limitations and clearly state the potential bottlenecks (e.g., finetuning, choice of $H$, scalability). They also reflect on the non-universality of bridge reversibility for fBM, which is commendable.

**Paper Formatting Concerns:**

None. The paper appears to follow the NeurIPS formatting guidelines. Equations, figures, and citations are all properly handled.

**Quality:**

3

**Strengths And Weaknesses:**

#### **Strengths**


* **Theoretical Soundness**: The paper rigorously derives the FDBM framework from foundational principles, including its compatibility with both bridge matching and Schrödinger bridge objectives. The augmentation trick via MA-fBM is well-motivated and mathematically justified.

* **Empirical Results**: The method significantly improves over Brownian baselines in both protein and image domains. The experiments are well-chosen: synthetic datasets highlight the effect of Hurst index $H$, and real-world tasks (protein conformational change, AFHQ translation) show strong performance in practice.


* **Clarity of Formulation**: While technically deep, the paper is careful in defining all notation and path measures. The derivations (e.g., the conditional drift for the MA-fBM bridge) are nontrivial but complete.

#### **Weaknesses**

* **Accessibility and Background**: The paper presumes strong familiarity with stochastic processes (e.g., fBM, OU processes, Schrödinger bridges), which may hinder accessibility for machine learning audiences. Although appendices cover more details, some higher-level intuitions could be integrated into the main text.

* **Empirical Scope**: The evaluation, while thorough in its selected domains, is limited to a few types of datasets. It would be helpful to demonstrate FDBM on temporal tasks, in, say, finance or speech, to generalize further the claim about modeling long-range dependencies. Also, it makes the theory more powerful if you can make one task excel rather than working on multiple tasks but get marginal improvements.

* **Hyperparameter Sensitivity**: The choice of the number of OU components $K$, reversion speeds $\gamma_k$, and the tuning of $H$ are empirically explored, but more principled insights on selection would strengthen practical utility.

* **Computational Overhead**: Although the use of MA-fBM is tractable, the augmented state dimension can become large (e.g., $K=6$ OU processes per dimension). Some discussion on scalability (e.g., for long sequences or high-dimensional states) would be useful.

---

> ### Author Rebuttal · Authors · 2025-07-30
>
> We thank the reviewer for their thoughtful and thorough evaluation of our work. We appreciate that the reviewer acknowledges the *theoretical soundness*, *empirical performance* and the *clarity in our formulation*.
>
> >some higher-level intuitions could be integrated into the main text.
>
> We kindly refer the reviewer to our response to Reviewer fb9f, where we present a revised version of our introduction. Our paper is indeed rigorously rooted in stochastic processes, which we see as a strength rather than a weakness.
>
> > It would be helpful to demonstrate FDBM on temporal tasks
>
> As suggested, we have conducted an additional experiment on the cell differentiation task introduced by Somnath et al. [2]. This task allows us to evaluate FDBM for cell differentiation prediction on both the distributional quality and perturbation accuracy of the generated data using distributional metrics such as Wasserstein-2 distance ($W_\varepsilon$) [9] and kernel maximum mean discrepancy (MMD) [10], as well as the Perturbation signature $\ell_2(\mathrm{PS})$ [11] and RMSD. The dataset provided in Somnath et al. [2] consists of two snapshots: one recorded on day 2, when most cells remain undifferentiated, and another on day 4, which includes a diverse set of mature cell types. We evaluate the performance of FDBM in the table below and compare its performance against forward-backward Schrödinger bridge models (FBSB) [12], SBALIGN [2], and ABM [1]. Consistent with our findings on protein conformational changes, FDBM shows an advantage in the rough regime ($H=0.3$ and $H=0.4$) showing better performance compared to FBSB, SBALIGN and ABM in $3$ out of $4$ metrics. All scores are averaged over $10$ training trials and $10$ sampling trials for each trained model.
>
> |**Methods**|**MMD↓**|**Wₑ↓**|**ℓ₂(PS)↓**|**RMSD↓**|
> |-------------------|----------:|-------:|-----------:|---------:|
> |FBSB [2]|1.55e-2|12.50|4.08|9.64e-1|
> |FBSB with SBALIGN [2]|**5.31e-3**|10.54|0.99|9.85e-1|
> |SBALIGN [2]|1.07e-2|11.11|1.24|9.21e-1|
> |ABM [1] (retrained)|4.10e-2|9.50|0.89|8.72e-1|
> |FDBM(H=0.3)|5.34e-2|**9.32**|0.89|**8.11e-1**|
> |FDBM(H=0.4)|4.52e-2|9.35|**0.85**|8.21e-1|
>
> >The choice of the number of OU components $K$, reversion speeds $\gamma_k$ and the tuning of $H$
>
> First of all we stress that we fix $K=5$ and $\gamma_1,...,\gamma_K$ throughout all experiments reported in the paper.
> - The choice of $K$ trades quality of approximation for computation cost. We fix $K=5$ to only minimally increase the computational cost while having a good approximation. We included in our Appendix a plot of the error of approximating in the different regimes of MA-fBM. For $H>0.5$ we observe a maximal $L_2$ error of 1.5e-4 and for $H\in[0.2,0.5]$ a maximal $L_2$ error of 5e-2.
> - We follow Daems et al. [14] in the choice of $\gamma_1,...,\gamma_K$ as a geometrically spaced grid to minimize the $L_2$-error of approximation.
>
> We observe that FDBM performs best in the rough regime ($H < 0.5$) for both protein conformational change prediction and the cell differentiation task, with $H = 0.2$ and $H = 0.3$ yielding the strongest results. However, we believe that the optimal choice of $H$ depends on the specific application, as demonstrated in the task of unpaired image translation.
>
> >Can you provide more intuition or visual illustrations (e.g., in Figure 1)
>
> To better illustrate the nature of the generated paths as well as Proposition 5, we replicated the toy experiment from Bortoli et al. [Figure 1, 1], where initial samples from $(-2,-2)$ are paired with $(2,2)$, and $(-2,2)$ with $(2,-2)$. We observe that SBALIGN fails to preserve this coupling while ABM and FDBM preserves the coupling while offering in the rough regime $(H=0.2)$, trajectories that explore more of the space, whereas in the smooth regime $(H=0.9)$, nearly straight-line paths emerge. Although we cannot share the figure during the rebuttal, we included it as a figure in the revision.
>
> > Does your method generalize to discrete-time settings or other noise models (e.g., Lévy noise, jump diffusions)?
>
> Instead of sampling $t \sim \mathcal{U}[0, T]$ during training, one could sample from a discrete set of time points, e.g., $t \sim \mathcal{U}\{0 = t_0, t_1, \ldots, t_N = T \}$, and use only these time steps for inference via Euler–Maruyama discretization (or any other suitable scheme). The resulting forward process $X_{t_0}, \ldots, X_{t_N}$ would then follow the marginals of the MA-fBM-driven SDE at those discrete times. Extending Schrödinger bridge models to Lévy processes is an interesting but technically challenging endeavor, as the discontinuous nature of Lévy paths requires a generalization of the theory for continuous paths presented in Léonard [21, 22], including an extension of the Girsanov theorem.
>
> > Could you comment more on computational cost
>
> We now provide a complete runtime comparison in the tables below. Training times for ABM and FDBM are nearly identical and both outperform SBALIGN, which requires approximating two functions and thus involves a larger model. For sampling, ABM and FDBM again show an advantage over SBALIGN. FDBM requires, on average, only 4.22e-2 seconds more than ABM to sample a conformation over 100 Euler–Maruyama steps. This slight increase is due to simulating a higher-dimensional stochastic process. However, the effect is minor, the computation during sampling is dominated by the forward passes through the GNN, which are identical for both ABM and FDBM.
>
> |**Runtime paired setting [s]**|**SBALIGN**|**ABM**|**FDBM (ours)**|
> |---------------------------------------------------------------------------------------|--------------:|---------------:|---------------:|
> |Average training step time|0.0159±0.0075|0.01438±0.0065|0.01412±0.0063|
> |Average sampling time of one conformation over $100$ steps |0.7078±0.3409 |0.6424±0.2992 |0.6846±0.3021|
>
> Throughout all unpaired data translation experiments, we use the same model architecture for both SBFlow and FDBM, resulting in nearly identical training runtimes. The sampling algorithm during inference is the same for the paired and unpaired setting.
>
> |Runtime unpaired setting [s]| Sampling from (partially) pinned process  |Calculation of loss |
> |-|-|-|
> |SBFLow|0.0010 $\pm$ 0.0002|0.0132 $\pm$ 0.0074|
> |FDBM|0.0011 $\pm$ 0.0003|0.0132 $\pm$ 0.0018|
>
> >Can you give more experimental details of the protein task?
>
> For a description of the dataset, we refer the reviewer to our response to Reviewer v2k6. In addition, we have now included the following section on implementation details in the appendix:
>
> *The results reported in Table 2 were obtained by averaging over 5 training trials, each run for $300$ epochs, and performing one sampling trial per trained model, generating a single path over $100$ time steps. We use the AdamW optimizer with an initial learning rate of 0.001 and a training batch size of 2. During validation, inference is performed using the exponential moving average of the model parameters, which is updated at every optimization step with a decay rate of 0.9. After each epoch, we simulate trajectories on the validation set and compute the mean RMSD. The model achieving the lowest mean RMSD on the validation set is selected for final evaluation on the test set.*
>
> *We use the GNN architecture from Somnath et al. [2], but—following Bortoli et al. [1]—we additionally provide the initial state $x_0$ by concatination as input to the network. Nevertheless, the GNN we use for ABM and FDBM has fewer parameters, since Somnath et al. [2] approximate two functions ($b_t$ and $\nabla_x \log h_t$ via that GNN resulting in more parameters in the output layer. We would like to stress that ABM and FDBM deploy the same model architecture and summarize the number of used parameters in the following table*:
>
> |**Model parameters per task (paired)**|**SBALIGN**|**ABM**|**FDBM (ours)**|
> |------------------------|-----------|-------|---------------|
> |Rotated Moons|58692|31618|31618|
> |T-Shape|19204|10754|10754|
> |Cell Differentiation|310372|177970|177970|
> |Predicting Conformations|545220|537900|537900|
>
> **Additional empirical evaluations.**
> We kindly refer the reviewer to our response to Reviewer v2k6, where we included the $\Delta \text{RMSD}$ metric to assess whether FDBM predicts structures closer to the holo conformation than the initial apo conformation in the task of protein conformational change prediction. Moreover, we included an additional baseline—Sesame [3], a recent method based on Flow matching presented in Miñán et al. [6]. We find that FDBM shows consistently better performance compared to ABM and Sesame in the rough regime ($H=0.3,0.2,0.1$), as measured by both median and mean $\Delta RMSD$. For $H=0.2$ and $H=0.3$, FDBM matches or exceeds ABM and Sesame across all evaluated metrics.
>
> We sincerely thank the reviewer for the valuable feedback and their detailed engagement with our work. We hope that our response adequately addresses all the questions and concerns raised by the reviewer. We kindly ask the reviewer to consider increasing their score in light of our clarifications and additional results.
>
> [1] Bortoli et al. Augmented bridge matching, 2023.
>
> [2] Somnath et al. Aligned diffusion schrödinger bridges. UAI 2023.
>
> [5] Miñán et al. Sesame: Opening the door to protein pockets. GEM Workshop, ICLR 2025.
>
> [6] Zhang et al. Bending and Binding: Predicting Protein Flexibility upon Ligand Interaction using Diffusion Models. GenBio Workshop, NeurIPS 2023.
>
> [9] Cuturi et al. Sinkhorn Distances: Lightspeed Computation of Optimal Transport. NeurIPS 2013.
>
> [10] Gretton et al. A Kernel TwoSample Test. JMLR 2012.
>
> [11] Bunne et al. Supervised Training of Conditional Monge Maps. NeurIPS 2022.
>
> [21] Christian Léonard. A survey of the schrödinger problem and some of its connections with optimal transport, 2013.
>
> [22] Christian Léonard. Girsanov theory under a finite entropy condition, 2011.

---

> > ### Comment · Reviewer_fk33 · 2025-08-02
> > **response**
> >
> > Thanks to the authors for the detailed rebuttal and the additional experiments. I appreciate the effort, but I still find the paper quite difficult to follow—particularly the methodological sections involving MA-fBM and its integration into the bridge framework. Even with the revisions mentioned, the paper feels tailored to a very narrow audience, which limits its broader impact. The added temporal task (cell differentiation) is a good gesture, but the performance gains over baselines are modest and don’t clearly showcase the benefits of modeling long-range dependencies. I think the core idea is interesting and potentially valuable, but the presentation needs to be much clearer for the paper to be more impactful. I’m downgrading my score.

---

> ### Author Response · Authors · 2025-08-03
>
> We respectfully disagree with several aspects of the reviewer's assessment, which we believe are subjective and overlook key contributions of our work. We urge the other reviewers and the AC to consider this assessment in context, as it appears to fail an objective evaluation of the work.
>
> First, we acknowledge that our approach builds on rigorous concepts from stochastic processes, such as fractional Brownian motion and Schrödinger bridges. This is a deliberate and necessary choice: modeling complex dynamics in generative settings demands mathematically sound tools. As the reviewer themselves notes, our theoretical development is solid. Far from being a limitation, our foundation ensures the robustness, extensibility, and long-term relevance of our framework. We already noted that, to further improve accessibility, we have expanded the main text to include more high-level intuitions and conceptual framing.
>
> Second, we are concerned by the reviewer's assertion that the work targets too narrow an audience. This seems speculative and unsubstantiated. MA-fBM is already being explored in generative modeling [17] and protein structure prediction [23], and its use in our bridge formulation offers potential for broader applications across scientific machine learning. Many impactful papers at NeurIPS are rooted in specialized subfields—such as Lévy processes [16] or Riemannian geometry [24] — and the specificity of a method should not be viewed as a weakness if the problem is meaningful and the technical contribution sound.
>
> Finally, the inclusion of the cell differentiation task is not a superficial gesture, but a significant empirical validation of our method’s ability to capture temporal dynamics in real-world, high-impact data. This experiment complements our theoretical contributions and expands the applicability of our framework.
>
> We appreciate the reviewer’s engagement and suggestions, but believe our paper should be evaluated on the strength of its contributions, clarity of its revised exposition, and its scientific merit in both theoretical and applied domains.
>
> [16] Yoon et al. Score-based Generative Models with Lévy Processes. NeurIPS 2023.
>
> [17] Nobis et al. Generative Fractional Diffusion Models. NeurIPS 2024.
>
> [23] Liang et al. ProT-GFDM: A Generative Fractional Diffusion Model for Protein Generation, 2025.
>
> [24] Bortoli et al. Riemannian Score-Based Generative Modelling, NeurIPS 2022.

---

### Official Review · Reviewer_i6UN · 2025-06-18

**Clarity:** 2
**Significance:** 2
**Originality:** 3
**Rating:** 4
**Confidence:** 4

**Summary:**

This work proposes a novel generative diffusion bridge model, FDBM, with generalized stochastic dynamics, driven by non-Markovian fractional Brownian motion (fBM). The framework preserves the coupling information given paired training data, and is further exteneded to the Schrodinger bridge problem to handle unpaired training data. FDBM is evaluated on both paired and unpaired data tasks, including protein docking and image translation, achieving comparable performances compared to all baselines.

**Questions:**

1. I have strong reservations regarding the added complexity of parameter tuning due to the introduction of two additional hyperparameters, $H$ and $K$. Could you please give a prior intuition on how hyperparameter choices (especially $H$ and $K$) may affect model performance, and further explain for the variation in optimal hyperparameters across different tasks.
2. As noted in Weaknesses, the protein docking task is less representative. It is strongly recommended to evaluate FDBM on a molecular dynamics dataset, even a toy case like Chignolin.
3. I wonder about the backbone model used for the protein docking task. Please make sure that it is the same as used in SBALIGN and ABM, otherwise the comparison is unfair. Meanwhile, SBALIGN, ABM and FDBM are all generative frameworks while EGNN is a GNN architecture, which should not be compared together.
4. Could you please give some more details on pretraining and fine-tuning on AFHQ?
5. I would like to know whether such a complex generative framework like FDBM will harm the generative efficiency of the model. Please give a comparison of the efficiency of each model on the protein docking task and image translation task.
6. The standard deviation of RMSD should not be used as a sole metric for comparison, as both high and low values may have context-dependent justifications. Please do not bold std in Table 2.
7. Typo: "the the" in line 47, 130, 179.

**Ethical Concerns:**

["NO or VERY MINOR ethics concerns only"]

**Final Justification:**

My main concern was that, compared to ABM, FDBM introduces additional hyperparameters $H$ and $K$, which I believed could increase the difficulty of hyperparameter tuning. However, the authors clarified in the rebuttal that all experiments were conducted with a fixed $K=5$, and the model still achieved strong performance. I find this demonstrates that FDBM is reasonably robust to the appropriate choice of $K$. On the other hand, I agree with the authors' explanation that $H$ can be viewed as an extension of the diffusion dynamics framework. Additionally, the supplementary experiments regarding inference time further show that FDBM does not significantly impair inference efficiency, which addresses my concern. Therefore, I will raise my score to 4.

**Limitations:**

The current experiments do not illustrate the necessity of introducing more hyperparameters in FDBM, which requires a more detailed explanation.

**Quality:**

3

**Strengths And Weaknesses:**

**Strengths**
- The proposed framework demonstrates clear novelty in its formulation, supported by a rigorous theoretical foundation and well-structured and comprehensive proofs.
- The framework is applicable to both paired data tasks and unpaired data tasks, broadening its usefulness.

**Weaknesses**
- Compared with ABM, FDBM introduces two more hyperparameters ($H$ and $K$), which greatly increases the difficulty of hyperparameter tuning.
- The paper does not provide a prior analysis of how hyperparameter choices may affect model performance, nor does it offer a convincing explanation for the variation in optimal hyperparameters across different tasks.
- The experiments on paired data tasks are insufficiently representative. In the protein docking task, once the protein and its binding partner are specified, the bound conformation of the complex is largely deterministic and independent of the initial configuration. As a result, the primary focus lies on accurately modeling the target marginal distribution $\Pi_1$, while the coupling information $\Pi_{0,1}$ is of limited relevance. Consequently, the advantages of FDBM in preserving coupling structures cannot be effectively demonstrated in this setting.

---

> ### Author Rebuttal · Authors · 2025-07-30
>
> We appreciate the reviewer’s thoughtful engagement with our work and thank them for recognizing the novelty of our framework and its rigorous theoretical foundation.
>
> > the advantages of FDBM in preserving coupling structures cannot be effectively demonstrated
>
> We would like to point out that coupling preservation is not an advantage of FDBM over ABM, as ABM preserves couplings [1]; rather, this property distinguishes both from SBALIGN. In Proposition 5, we show that ABM’s coupling preservation extends to the generalized dynamics of FDBM. To illustrate this, we replicated the toy experiment from Bortoli et al. [Figure 1, 1], where initial samples from $(-2,-2)$ are paired with $(2,2)$, and $(-2,2)$ with $(2,-2)$. We observe that SBALIGN fails to preserve this coupling. Consistent with Proposition 5, FDBM preserves the coupling while offering a wider range of trajectories: in the rough regime $(H=0.2)$, trajectories explore more of the space, whereas in the smooth regime $(H=0.9)$, nearly straight-line paths emerge. Although we cannot share the figure during the rebuttal, we have included it in the revised manuscript to illustrate Proposition 5.
>
> > It is strongly recommended to evaluate FDBM on a molecular dynamics dataset, even a toy case like Chignolin.
>
> We thank the reviewer for the valuable suggestion to evaluate FDBM on a molecular dynamics dataset. While we agree that molecular dynamics is an interesting application domain for FDBM, conducting such experiments requires considerable setup time and computational resources. Given the limited duration of the rebuttal period, we instead opted to perform an additional experiment on the cell differentiation task introduced by Somnath et al. [2]. This task allows us to assess FDBM for cell differentiation prediction on both the distributional quality and perturbation accuracy of the generated data using distributional metrics such as Wasserstein-2 distance ($W_\varepsilon$) [9] and kernel maximum mean discrepancy (MMD) [10], as well as the Perturbation signature $\ell_2(\mathrm{PS})$ [11] and RMSD. The dataset consists of two snapshots: one recorded on day 2, when most cells remain undifferentiated, and another on day 4, which includes a diverse set of mature cell types. We evaluate FDBM on this task of inferring cell differentiation in the table below and assess its performance against forward-backward Schrödinger bridge models (FBSB) [12], SBALIGN, and ABM. Consistent with our findings on protein conformational changes, FDBM shows an advantage in the rough regime ($H=0.3$ and $H=0.4$) showing superior performance compared to FBSB, SBALIGN and ABM in $3$ out of $4$ metrics. All scores are averaged over $10$ training trials and $10$ sampling trials for each trained model.
>
> |**Methods**|**MMD↓**|**Wₑ↓**|**ℓ₂(PS)↓**|**RMSD↓**|
> |-------------------|----------:|-------:|-----------:|---------:|
> |FBSB [2]|1.55e-2|12.50|4.08|9.64e-1|
> |FBSB with SBALIGN [2]|**5.31e-3**|10.54|0.99|9.85e-1|
> |SBALIGN [2]|1.07e-2|11.11|1.24|9.21e-1|
> |ABM [1] (retrained)|4.10e-2|9.50|0.89|8.72e-1|
> |FDBM(H=0.3)|5.34e-2|**9.32**|0.89|**8.11e-1**|
> |FDBM(H=0.4)|4.52e-2|9.35|**0.85**|8.21e-1|
>
> > I have strong reservations regarding the added complexity of parameter tuning due to the introduction of two additional hyperparameters,
>
> First, we would like to clarify that we fixed $K = 5$ throughout all experiments reported in the main paper. The choice of $K$ involves a trade-off between the quality of the fBM approximation and computational cost. Similar to Daems et al., we fix $K=5$ to only minimally increase the computational cost while having a good approximation of fBM. We now included in our Appendix a plot of the error of approximation in the different regimes of MA-fBM. For $H>0.5$ we observe a maximal $L_2$ error of 1.5e-4 and for $H\in[0.2,0.5]$ a maximal $L_2$ error of 5e-2. Moreover, the setting $H = 0.5$ and $K = 1$ closely recovers the original model driven by Brownian motion. Rather than merely introducing additional hyperparameters, our framework extends the original model by enabling a broader class of stochastic dynamics for diffusion bridge modeling. From this perspective, we do not consider $H$ to be a hyperparameter, but rather a modeling choice—one that allows access to diffusion dynamics that were previously out of reach. We observe that FDBM performs best in the rough regime ($H < 0.5$) for both protein conformational change prediction and the cell differentiation task (see below), with $H = 0.2$ and $H = 0.3$ yielding the strongest results. We do believe that the optimal choice of $H$ depends on the specific application, as demonstrated in the task of unpaired image translation.
>
> > I wonder about the backbone model used for the protein docking task.
>
> We use the GNN architecture from Somnath et al. [2], but—following Bortoli et al. [1]—the initial state $x_0​$ is additionally provided to the network by concatenating it with the input. Nevertheless, the GNN we use for ABM and FDBM has fewer parameters, since Somnath et al. [2] approximate two functions ($b_t$ and $\nabla_x \log h_t$) with a single GNN resulting in more parameters in the output layer. We would like to stress that ABM and FDBM deploy the same model architecture and summarize the number of used parameters in the following table:
>
> |**Model parameters per task (paired)**|**SBALIGN**|**ABM**|**FDBM (ours)**|
> |------------------------|-----------|-------|---------------|
> |Rotated Moons|58692|31618|31618|
> |T-Shape|19204|10754|10754|
> |Cell Differentiation|310372|177970|177970|
> |Predicting Conformations|545220|537900|537900|
>
> > while EGNN is a GNN architecture, which should not be compared together.
>
> EGNN is a widely used baseline in the literature [2,5,6], and both SBALIGN and our method incorporate a GNN backbone within the diffusion process. For this reason, we believe that a comparison with EGNN is both appropriate and informative. Additionally, we refer the reviewer to our response to Reviewer v2k6, where we included an additional baseline—Sesame [3]—which is based on flow matching.
>
> > Could you please give some more details on pretraining and fine-tuning on AFHQ?
>
> We used the same training and sampling parameterizations for all datasets and experiments. Parameterizations for DiT-B/2 and DiT-L/2 were kept identical. See the table below for the parameterizations of all experiments. Note that for DiT architectures compute also grows with the number of tokens (i.e., patches), which grow with the image resolution.
>
> |Model|Optimizer|Learning Rate|EMA Rate|Linear Warmup|Cosine Decay|Online Finetuning|Euler-Maruyama Steps|Parameters|
> |-------|-------------------------|-------------|--------|-------------| ------------|-----------------|--------------------|-|
> |DiT-B/2|lion|0.0001|0.999|10K|90K|4K|200|130M|
> |DiT-L/2|lion|0.0001|0.999|10K|90K|4K|200|458M|
>
> > The standard deviation of RMSD should not be used as a sole metric for comparison
>
> We thank the reviewer for pointing out this mistake and have removed the bold formatting from the standard deviation values of the RMSD.
>
> > Please give a comparison of the efficiency of each model on the protein docking task and image translation task.
>
> We provide a complete runtime comparison in the tables below. Training times for ABM and FDBM are nearly identical and both outperform SBALIGN, which requires approximating two functions and thus involves a larger model. For sampling, ABM and FDBM again show an advantage over SBALIGN. FDBM requires, on average, only 4.22e-2 seconds more than ABM to sample a conformation over 100 Euler–Maruyama steps. This slight increase is due to simulating a higher-dimensional stochastic process. However, the effect is minor, as the dominant computational cost during sampling comes from forward passes through the GNN, which are identical for both ABM and FDBM.
>
> |**Runtime paired setting**|**SBALIGN**|**ABM**|**FDBM (ours)**|
> |---------------------------------------------------------------------------------------|--------------:|---------------:|---------------:|
> |Average training step time|0.0159±0.0075|0.01438±0.0065|0.01412±0.0063|
> |Average sampling time of one conformation over $100$ steps |0.7078±0.3409 |0.6424±0.2992 |0.6846±0.3021|
>
> Throughout all unpaired data translation experiments, we use the same model architecture for both SBFlow and FDBM, resulting in nearly identical training runtimes. The sampling algorithm during inference is the same for the paired and unpaired setting:
>
> |Runtime unpaired setting [s]| Sampling from (partially) pinned process  |Calculation of loss term|
> |-|-|-|
> |SBFLow|0.0010 $\pm$ 0.0002|0.0132 $\pm$ 0.0074|
> |FDBM|0.0011 $\pm$ 0.0003|0.0132 $\pm$ 0.0018|
>
> We truly appreciate the reviewers detailed engagement with our work and thank the reviewer again for the valuable feedback. We hope that we have sufficiently addressed the weaknesses and questions raised by the reviewer and kindly ask the reviewer to reconsider the evaluation of our work.
>
> [1] Bortoli et al. Augmented bridge matching, 2023.
>
> [2] Somnath et al. Aligned diffusion schrödinger bridges. UAI 2023.
>
> [5] Miñán et al. Sesame: Opening the door to protein pockets. GEM Workshop, ICLR 2025.
>
> [6] Zhang et al. Bending and Binding: Predicting Protein Flexibility upon Ligand Interaction using Diffusion Models. GenBio Workshop, NeurIPS 2023.
>
> [8] Jing et al. Generative Modeling of Molecular Dynamics Trajectories, NeurIPS 2024.
>
> [9] Cuturi et al. Sinkhorn Distances: Lightspeed Computation of Optimal Transport. NeurIPS 2013.
>
> [10] Gretton et al. A Kernel TwoSample Test. JMLR 2012.
>
> [11] Bunne et al. Supervised Training of Conditional Monge Maps. NeurIPS 2022.
>
> [12] Chen et al. Optimal Transport in Systems and Control. Annual Review of Control, Robotics, and Autonomous Systems 2021

---

> > ### Comment · Reviewer_i6UN · 2025-08-01
> >
> > Thanks for the authors' detailed response, which addressed most of my concerns. I agree that the additional variable $H$ introduced in FDBM can indeed be regarded as an extension of the diffusion dynamics framework. I will raise my score accordingly.

---

### Official Review · Reviewer_v2k6 · 2025-06-24

**Clarity:** 3
**Significance:** 3
**Originality:** 3
**Rating:** 5
**Confidence:** 2

**Summary:**

This paper proposes fractional diffusion bridge models, a novel generative model that captures long range dependencies of the underlying stochastic process by leveraging MA-fBM. In addition to theoretical insights, the model is benc marked on protein conformation sampling and shown to perform competitively against the state-of-the-art.

**Questions:**

I am curious to see how the model performs in the different types of motions, namely apo & apo, apo & holo, holo & apo (different ligands), and holo & apo (same ligand), surveyed in D3PM. Providing this context will elucidate certain biases present in the model with respect to the capability to predict physically realistic conformations. Additionally, more description on the dataset splits is necessary for faithful reproduction of the model trained for this experiment.

**Ethical Concerns:**

["NO or VERY MINOR ethics concerns only"]

**Final Justification:**

I have read the authors’ rebuttal and comments to other reviewers and thank them for the detailed explanations and additional work. As I stated before, I am not very familiar with diffusion bridge models and hence cannot comment on the technical  concerns of the other reviewers. However, I do find the work to be impactful as a tool to effectively generate 3D conformations and for the rebuttal arguments to my own questions satisfactory. For these  reasons, I will raise my rating to accordingly.

**Limitations:**

yes

**Quality:**

2

**Strengths And Weaknesses:**

The paper is clearly written and the results are presented fairly intuitively. I am not very familiar with this field so I am can comment on neither the significance nor originality.

The D3PM benchmarking results are show some clear performance gains numerically, however it is not clear what these performance improvements imply in terms of the model's capability to generate meaningful and realistic conformations.

---

> ### Author Rebuttal · Authors · 2025-07-29
>
> We thank the reviewer for their constructing feedback. We appreciate that the reviewer highlights both the clarity of our presentation and the clear numerical performance gains of FDBM on the D3PM benchmark.
>
> >more description on the dataset splits is necessary for faithful reproduction of the model trained for this experiment.
>
> We agree with the reviewer and now add a description of the D3PM dataset splits used in our experiments in addition to referencing the dataset description by Somnath et al. [2]. For the best comparability, we closely follow the training set-up of SBALIGN [2] with the same dataset splits. We have added *Appendix E.3 Datasets*, where we give the following detailed description of dataset preparation and split strategy:
>
> *Following the training and evaluation setup of Somnath et al. [2], we use their curated subset of the D3PM dataset, which focuses on structure pairs with Cα RMSD > 3 Å. This subset initially comprises of 2,370 ligand-free (apo) - ligand-bound (holo) pairs. To ensure high-quality alignment, Somnath et al. [2] compute the Cα RMSD between pairs of proteins' common residues superimposed using the Kabsch algorithm and retain only those examples where the computed RMSD closely matches the original D3PM value. This results in a cleaned dataset of 1,591 pairs, which is split into training, validation, and test sets of 1,291 / 150 / 150 examples, respectively. All structures are Kabsch-superimposed to remove global translational and rotational artifacts, ensuring that the model focuses solely on internal conformational changes.*
>
> >I am curious to see how the model performs in the different types of motions, namely apo & apo, apo & holo, holo & apo (different ligands), and holo & apo (same ligand), surveyed in D3PM.
>
> As described in the SBALIGN implementation (repository provided by Somnath et al. [2]), the dataset split used in Somnath et al. [2] and our work consists exclusively of ligand-free (apo) and ligand-bound (holo) structures. Consequently, the scope of our current work is limited to apo → holo transitions. We agree that evaluating additional motion types would provide valuable insight. We explore this direction now in our future work.
>
> >however it is not clear what these performance improvements imply in terms of the model's capability to generate meaningful and realistic conformations.
>
> The numerical improvements in RMSD and threshold accuracy (i.e., the percentage of predictions with $RMSD < \tau$) reflect meaningful gains in biological relevance when interpreted in the context of structural biology. RMSD is a standard and widely accepted metric to assess the accuracy of predicted protein structures [2,4,5,6]. In particular, an RMSD below $2Å$ is commonly used as a threshold for correct bound structure prediction [7] and structural discernibility [2,5]. Accordingly, the proportion of predictions falling below this threshold is a direct indicator of the model's ability to generate physically realistic conformations. FDBM increases the proportion of correct and discernible predictions ( $RMSD < 2 Å$ ) from $43$% with ABM to $48$ %, while also improving the median RMSD from $2.40 Å$ to $2.12 Å$. This indicates that FDBM produces a higher fraction of near-native structures, compared to ABM and SBALIGN. To further address the reviewers concern, we added an additional metric to evaluate if FDBM is able to predict structures closer to the holo conformation than to the initial apo conformation. We computed the $\Delta RMSD$ [6] defined via
>
> $\Delta RMSD := RMSD(x_0,x_1)-RMSD(\hat{x}_1,x_1)$,
>
> where $(x_0,x_1)$ is a pair of the D3PM test set and $\hat{x}_1$ is predicted starting from $x_0$. Additionally, we compare our results to Sesame, a recent method based on Flow matching presented in Miñán et al. [6] that uses the exact same dataset split as Somnath et al. [2] and our work:
>
> |**D3PM Test Set**|**RMSD(Å) Median**↓|**RMSD(Å) Mean**↓|**RMSD(Å) Std**| **%RMSD(Å) < 2**↑|**%RMSD(Å) < 5**↑| **%RMSD(Å) < 10**↑|**ΔRMSD(Å) Median**↑|**ΔRMSD(Å) Mean**↑| **ΔRMSD(Å) Std**|
> |------------------------------|--------------------:|------------------:| -----------------:|-----------------:|-----------------:|------------------: |--------------------:|------------------:|-----------------:|
> |EGNN [2]|19.99|21.37|8.21|1%|1%|3%|–|–|–|
> |SBALIGN [2]|3.80|4.98|3.95|0%|69%|93%|–|–|-|
> |Sesame [5]|2.87|3.65|2.95|38%|82%|96%|2.15|3.11|4.26|
> |ABM [1] (retrained)|2.40|3.49|3.54|43%|84%|96 %|2.43|3.35|4.29|
> |FDBM(H=0.3)|2.33|3.42|3.42|43%|85%|97%|**2.52**|**3.49**|4.39|
> |FDBM(H=0.2)|**2.12**|**3.34**|3.59|**48%**|**86%**|96%|2.44|3.39|4.28|
> |FDBM(H=0.1)|2.20|3.44|3.57|46%|83%|**97%**|2.47|3.45|4.29|
>
> *Values for ABM and FDBM are averaged over 5 runs.*
>
> We find that FDBM shows consistently better performance compared to ABM and Sesame in the rough regime ($H=0.3,0.2,0.1$), as measured by both median and mean $\Delta RMSD$. This indicates that FDBM generated structures are closer to the holo conformations—relative to their apo starting points—than those produced by ABM or Sesame. For $H=0.2$ and $H=0.3$, FDBM matches or exceeds ABM and Sesame across all evaluated metrics.
>
> ### Additional Evaluations of our FDBM
> We would like to draw the reviewer’s attention to our rebuttals to Reviewer i6UN and Reviewer fk33 for an additional evaluation of FDBM on the the cell differentiation task introduced by Somnath et al. [2]. The dataset provided in Somnath et al. [2] consists of two snapshots: one recorded on day 2, when most cells remain undifferentiated, and another on day 4, which includes a diverse set of mature cell types.
>
> We evaluate the performance of FDBM in the table below and compare its performance against forward-backward Schrödinger bridge models (FBSB) [12], SBALIGN [2], and ABM [1]. Consistent with our findings on protein conformational changes, FDBM shows an advantage in the rough regime ($H=0.3$ and $H=0.4$) showing better performance compared to FBSB, SBALIGN and ABM in $3$ out of $4$ metrics. The values of ABM and FDBM is averaged over $10$ training trials and $10$ sampling trials for each trained model.
>
> |**Methods**|**MMD↓**|**Wₑ↓**|**ℓ₂(PS)↓**|**RMSD↓**|
> |-|-:|-:|-:|-:|
> |FBSB [2]|1.55e-2|12.50|4.08|9.64e-1|
> |FBSB with SBALIGN [2]|**5.31e-3**|10.54|0.99|9.85e-1|
> |SBALIGN [2]|1.07e-2|11.11|1.24|9.21e-1|
> |ABM [1] (retrained)|4.10e-2|9.50|0.89|8.72e-1|
> |FDBM(H=0.3)|5.34e-2|**9.32**|0.89|**8.11e-1**|
> |FDBM(H=0.4)|4.52e-2|9.35|**0.85**|8.21e-1|
>
>
> We hope that our response adequately addresses the questions and concerns raised by the reviewer. In light of these clarifications and additional results, we kindly invite the reviewer to raise their final rating.
>
> [1] Bortoli et al. Augmented bridge matching, 2023.
>
> [2] Somnath et al. Aligned diffusion schrödinger bridges. UAI, 2023.
>
> [4] Corso et al. DiffDock: Diffusion Steps, Twists, and Turns for Molecular Docking, ICLR 2023.
>
> [5] Miñán et al. Sesame: Opening the door to protein pockets. GEM Workshop, ICLR 2025.
>
> [6] Zhang & Geffner et al. Bending and Binding: Predicting Protein Flexibility upon Ligand Interaction using Diffusion Models. GenBio Workshop, NeurIPS 2023.
>
> [7] Chang et al. Empirical entropic contributions in computational docking: evaluation in APS reductase complexes. J Comput Chem. 2008.
>
> [12] Chen et al. Optimal Transport in Systems and Control. Annual Review of Control, Robotics, and Autonomous Systems 2021

---

> ### Author Response · Authors · 2025-08-07
>
> We thank the reviewer for their time and thoughtful feedback. In response, we evaluated FDBM using the $\Delta \text{RMSD}$ metric, verifying that the generated conformations are closer to the holo structures than the initial apo conformations. Additionally, we included a new evaluation on the cell differentiation task, where FDBM outperforms FBSB, SBALIGN, and ABM in 3 out of 4 metrics.
>
> If the rebuttal has addressed the reviewer’s concerns, we would greatly appreciate it if the reviewer would consider raising their final rating. Otherwise, we are happy to provide further clarification on any remaining questions.

---

### Official Review · Reviewer_fb9f · 2025-07-03

**Clarity:** 2
**Significance:** 2
**Originality:** 2
**Rating:** 4
**Confidence:** 3

**Summary:**

The authors introduce Fractional Diffusion Bridge Models (FBDMs), a novel generative diffusion bridge framework based on non-Markovian fractional Brownian motion (fBM), which captures memory effects and long-range dependencies that standard Brownian motion cannot. The authors use a Markovian approximation for the fBM (MA-fBM) to enable tractable inference of the fBM while preserving its non-Markovian properties. The authors provide a principled theoretical framework for FBDMs based on the MA-fBM and demonstrate its use case through a set of empirical experiments on synthetic and real data. They evaluate FBDMs for the tasks of paired and unpaired data translation. Specifically, the authors demonstrated that their novel framework outperforms Brownian-based methods in both protein conformation prediction and unpaired image translation.

**Questions:**

What is still missing for me after reading the paper is intuition for why defining the stochastic bridge between two distributions using "non-Markovian fractional Brownian motion" would lead to "more flexible modelling of real world variability and biological dynamics" (lines 30-32). Could the authors provide some detail/backing for this hypothesis/claim?


Minor comments:

- Line 116: hyper-ref seems to be missing a reference to another section or appendix.
- Lines 180-184 seem to be a repetition of lines 91-97. You could save space by just referencing the previous paragraph.
- Line 189: "dynamik" -> "dynamic".
- Line 192: "the path probability for P'$\mathbb{P}$ ..." This is unclear or seems like a typo.

**Ethical Concerns:**

["NO or VERY MINOR ethics concerns only"]

**Final Justification:**

The authors did a good job of addressing most of my questions and comments, which include an improved motivation for the overall work and the addition of new experimental settings. They did not explicitly address my suggestions to test their method on the dhigh-imensional AFHQ-512 setting, and there is no way of confirming that the added toy example for Figure 1 makes sense. Nonetheless, their additions and responses to my concerns suffice for a score increase. I remain slightly reserved that this paper could benefit from another round of reviews, hence I remain in my position that this is a borderline paper.

**Limitations:**

Yes.

**Paper Formatting Concerns:**

No formatting concerns.

**Quality:**

3

**Strengths And Weaknesses:**

**Strengths:**

- The authors introduce a novel and principled framework for Fractional Diffusion Bridge Models (FDBMs) and provide comprehensive theoretical support for their approach.
- The authors provide empirical support for their proposed framework through a set of experiments on synthetic (toy) data, images (unpaired data translation), and protein conformation (predicting conformational changes of proteins).
- Moreover, the authors conduct several ablations to observe the effects of how changes in hyperparameters affect the performance of the proposed method across several scenarios.

**Weaknesses:**

- For the toy example in Figure 1, it would be useful to see how a (simple) baseline performs on these toy datasets. I think this would benefit in improving the overall understanding of FDBM and more clearly demonstrate the advantages of your proposed method. Moreover, this would provide a clear visual comparison with existing approaches.
- The authors claim that "FDBM scales robustly across high-dimensional domains ... ", but do not provide a thorough quantitative evaluation for AFHQ-512. i.e. referencing Table 3, evaluation is only provided up to AFHQ-256. It would be beneficial to include a Table 3.c, which shows results on AFHQ-512 to provide further support for the respective claim.
- Furthermore, for the unpaired data translation task, the authors consider the task of image translation only using the "cat" and "wildlife" subsets of the AFHQ dataset. To my understanding, the AFHQ dataset also has a "dog" subset. It would be useful to also observe two additional scenarios for this experiment (Table 3): (1) "dog" -> "wild", "wild" -> "dog", and (2) "dog" -> "cat", "cat" -> "dog". I think the addition of these settings would strengthen the claims of this work for the unpaired data translation task and improve the robustness of these empirical results.
- Motivation/intuition for why FBDM (compared to simpler approaches) to the application of predicting conformational changes of proteins is sparse/missing (see question below). More generally, I feel that the introduction could benefit from more contextualization and motivation to better place this paper in the field and to identify the gap this work is addressing.

---

> ### Author Rebuttal · Authors · 2025-07-30
>
> We thank the reviewer for their insightful feedback. We are particularly grateful for the recognition of the novelty and theoretical depth of our proposed framework.
>
> > For the toy example in Figure 1, it would be useful to see how a (simple) baseline performs
>
> We thank the reviewer for pointing out the missing illustration of the baseline. We have now included plots of the paths and alignments for the ABM baseline in both toy experiments. While FDBM achieves a lower Wasserstein distance than ABM on the Moons dataset, the paths and alignments appear visually indistinguishable. In contrast, on the T-shape dataset, we observe a visible advantage of FDBM in terms of alignment quality. Although we cannot share the figure during the rebuttal, we encourage the reviewer to visually compare Figure 1 of our work with Figure 2 in Somnath et al. [1]. Notably, Somnath et al. [1] visualize an average over $7$ paths for each test pair, whereas we display a single sample path for each pair.
>
> > The authors claim that "FDBM scales robustly across high-dimensional domains … ", but do not provide a thorough quantitative evaluation for AFHQ-512
>
> While Table 3 provides detailed results up to AFHQ-256, we emphasize that this already constitutes a high-dimensional and complex benchmark. Specifically, the Cat $\leftrightarrow$ Wild translation task in AFHQ-256 contains significant semantic variation and rich textural detail, even when compared to AFHQ-512 (see **Figure 2**, as well as **Figures 5 and 6** in the appendix). Our results in **Table 3 (b)** and  **Figure 3 (e,f)** show that performance stabilizes at $H = 0.4$, making it a critical configuration for analysis.
>
> We emphasize that our evaluation for AFHQ-512 at $H = 0.4$ included complete training and **quantitative evaluation for 10 sampling runs with respective standard deviations**, within our available computational budget. Extending such detailed ablations to AFHQ-512 was not feasible under current constraints, but we believe the results provided, combined with stable trends from AFHQ-32 to AFHQ-256, offer strong evidence for the scalability of FDBM.
> We were not able to replicate ablations for AFHQ-512, but we added an additional experiment for the base-line under best possible entropic regularization. To support the claim of scalability beyond AFHQ-256, we include below the AFHQ-512 results for $H = 0.4$, which was the worst performing stable H in AFHQ-256 (Fig 3 (c)) in comparison to the best possible baseline:
>
> |AFHQ-512|Cats $\rightarrow$ Wild|Wild $\rightarrow$ Cats|
> |-|-|-|
> ||FID $\downarrow$|FID $\downarrow$|
> |SBFlow|17.79  $\pm$ 0.66|**24.17 $\pm$ 0.81**|
> |FDBM (H=0.4)|**14.27  $\pm$ 0.86**|30.11 $\pm$ 0.75|
>
> These results are consistent with our findings on AFHQ-256, and show that FDBM maintains strong performance even as resolution increases. Visual consistency with AFHQ-256 (e.g., $H = 0.6$) is also observed, as discussed in relation to **Figures 2, 5, 6, 7, and 8**, where semantic correspondences remain stable across values of $H$.
>
> We note that **AFHQ-512 imposes substantial computational demands**, particularly in terms of GPU memory and training runtime. This is a key reason why we restricted our analysis at this scale to representative settings. Despite these limitations, we opted to show visual results at even higher resolution than related work (e.g., [20]), which do not report either scores or images at 512 $\times$ 512 resolution.
>
> We hope this clarifies our design decisions and supports the robustness of the method across increasing resolution scales.
>
> > It would be useful to also observe two additional scenarios for this experiment (Table 3): (1) "dog" -> "wild", "wild" -> "dog", and (2) "dog" -> "cat", "cat" -> "dog".
>
> We appreciate the reviewer’s suggestion to evaluate additional sub-domains in the AFHQ dataset, specifically *Dog $\leftrightarrow$ Wild* and *Dog $\leftrightarrow$ Cat*, to strengthen the empirical support for unpaired translation capabilities.
>
> While these experiments come at considerable computational cost, we allocated all remaining available compute to run ablations at **AFHQ-256 resolution** under the same settings as those used in *Cat $\leftrightarrow$ Wild*. Due to time and resource constraints, we were unable to run evaluations at AFHQ-512 for these tasks, but ensured training and evaluation at AFHQ-256.
>
> The results reported below showcase the robustness of FDBM across a wide range of Hurst indices $H$, and provide a comprehensive comparison to the SBFlow baseline. In all sub-domains, we observe clear improvements in FID and LPIPS, which further validates the effectiveness of FDBM in diverse unpaired translation scenarios.
>
> Below, we list the complete quantitative results. The best scores are highlighted in **bold**.
>
> **Table 1 AFHQ-256 Dogs $\leftrightarrow$ Wild**
>
> |Method|Dogs $\rightarrow$ Wild||Wild $\rightarrow$ Dogs||
> |-|-|-|-|-|
> ||FID  $\downarrow$|LPIPS  $\downarrow$|FID  $\downarrow$|LPIPS  $\downarrow$|
> |SBFlow [20]|20.74 $\pm$ 0.64|0.53 $\pm$ 0.002|47.07 $\pm$ 0.80|0.56 $\pm$ 0.002|
> |FDBM (H=0.9)|20.37 $\pm$ 0.98|**0.52 $\pm$ 0.002**|43.11 $\pm$ 0.68|**0.54 $\pm$ 0.002**|
> |FDBM (H=0.8)|19.22 $\pm$ 0.83|0.53 $\pm$ 0.003|41.76 $\pm$ 0.84|0.55 $\pm$ 0.002|
> |FDBM (H=0.7)|18.11 $\pm$ 0.75|0.53 $\pm$ 0.002|40.08 $\pm$ 0.73|0.56 $\pm$ 0.002|
> |FDBM (H=0.6)|18.43 $\pm$ 0.72|0.53 $\pm$ 0.002|39.84 $\pm$ 0.89|0.57 $\pm$ 0.002|
> |FDBM (H=0.5)|**14.74 $\pm$ 0.53**|0.55 $\pm$ 0.002|**37.68 $\pm$ 0.55**|0.58 $\pm$ 0.002|
> |FDBM (H=0.4)|15.78 $\pm$ 0.85|0.56 $\pm$ 0.002|38.51 $\pm$ 0.64|0.59 $\pm$ 0.001|
>
> **Table 2 AFHQ-256  Dogs $\leftrightarrow$ Cats**
>
> |Method|Dogs $\rightarrow$ Cats||Cats $\rightarrow$ Dogs||
> |-|-|-|-|-|
> |Method|FID  $\downarrow$|LPIPS $\downarrow$|FID  $\downarrow$|LPIPS $\downarrow$|
> |SBFlow [20]|**18.38 $\pm$ 0.36**|0.56 $\pm$ 0.002|50.08 $\pm$ 1.38|0.56 $\pm$ 0.002|
> |FDBM (H=0.9)|19.86 $\pm$ 0.67|**0.55 $\pm$ 0.002**|45.19 $\pm$ 0.74|**0.55 $\pm$ 0.002**|
> |FDBM (H=0.8)|20.14 $\pm$ 0.64|0.56 $\pm$ 0.002|45.08 $\pm$ 0.98|**0.55 $\pm$ 0.001**|
> |FDBM (H=0.7)|21.12 $\pm$ 0.62|0.56 $\pm$ 0.002|43.44 $\pm$ 0.82|0.56 $\pm$ 0.002|
> |FDBM (H=0.6)|22.06 $\pm$ 0.57|0.57 $\pm$ 0.001|41.35 $\pm$ 0.90|0.57 $\pm$ 0.002|
> |FDBM (H=0.5)|22.15 $\pm$ 0.75|0.58 $\pm$ 0.002|42.36 $\pm$ 0.77|0.58 $\pm$ 0.002|
> |FDBM (H=0.4)|24.79 $\pm$ 0.68|0.59 $\pm$ 0.002|**41.21 $\pm$ 1.12**|0.59 $\pm$ 0.002|
>
> > Motivation/intuition for why FBDM (compared to simpler approaches) to the application of predicting conformational changes of proteins is sparse/missing (see question below).
>
> We thank the reviewer for raising this point, and have now incorporated the following motivation and contextualization in our introduction:
>
> *Stochastic differential equations offer a natural framework for modeling the inherent randomness and continuous-time dynamics of real-world systems. This is precisely why they serve as the backbone of state-of-the-art generative models, such as diffusion models (both latent and non-latent). Traditionally, these models assume noise driven by standard Brownian motion (BM), which is Markovian with independent increments. However, this choice is motivated by mathematical tractability and simplicity than by faithfullness and fidelity to real-world data.*
>
> *Empirical data, particularly in complex systems like proteins often exhibit long-range temporal dependencies, heavy-tailed behaviors, and intricate dynamics that are poorly captured by memoryless processes. Consequently, diffusion models driven by BM may require many iterations to approximate the target distribution, leading to slow sampling or limited expressivity. These limitations have motivated recent efforts to explore generative models with non-standard noise sources [16,17,18].*
>
> *Our work extends such line of research to bridge models, where the goal is to transform a structured, non-Gaussian prior into a complex target distribution. We specifically investigate stochastic bridges driven by fractional Brownian motion (fBM), a generalization of BM with stationary but **dependent** increments, characterized by the Hurst index $H$, which governs both roughness and long-range dependence. By using fBM as the driving process, we introduce a more expressive and flexible framework for building bridges: When $H=0.5$, our method recovers classical BM-driven bridges, while other values of $H$ flexibly allow us to model a broader range of temporal behaviors, as demonstrated in our experiments.*
>
> *In the context of protein modeling, fBM-driven diffusion processes have already proven effective due to their superdiffusive properties, which better capture long-range correlations in protein structures [19]. This has led to improvements in both sample fidelity and diversity. We build upon this foundation, proposing fBM-driven bridges as a principled extension to model protein conformational changes, as studied in our paper.*
>
> We believe to have sufficiently addressed all the weaknesses and questions raised by the reviewer and kindly ask the reviewer to reconsider the evaluation of our work.
>
> [16] Yoon et al. Score-based Generative Models with Lévy Processes. NeurIPS 2023.
>
> [17] Nobis et al. Generative Fractional Diffusion Models. NeurIPS 2024.
>
> [18] Shariatian et al. Denoising Levy Probabilistic Models. ICLR 2024
>
> [19] Paquet et al. Annealed fractional Lévy–Itō diffusion models for protein generation. Computational and Structural Biotechnology Journal, 2024.
>
> [20] De Bortoli et al. Schrödinger bridge flow for unpaired data translation. NeurIPS 2024.

---

> > ### Comment · Reviewer_fb9f · 2025-08-05
> >
> > I thank the authors for their detailed rebuttal. Their response has addressed the majority of my questions and concerns. With that, I am happy to raise my score.
> >
> > One last comment. The authors mention, "Extending such detailed ablations to AFHQ-512 was not feasible under current constraints". I completely understand the challenges of computational and time constraints during the short turnaround period for rebuttals, and I did not consider this in my score change. However, I encourage the authors to conduct these experiments and include the results in the final version of the manuscript, should the paper be accepted.

---

### Comment · Area_Chair_6czT · 2025-08-05
**Could you kindly confirm that you have read the authors’ rebuttal?**

Dear Reviewer,
We sincerely appreciate your time and contribution to the review process. Could you kindly confirm that you have read the authors’ rebuttal if you haven't done it yet?
Best regards,
AC

---

### Author Response · Authors · 2025-08-09

Dear Reviewers and Respected Area Chair,

As the Reviewer-Author Discussion comes to an end, we would like to take this opportunity to thank the reviewers again for the time they dedicated to reviewing our paper and for their valuable suggestions. We appreciate the reviewers’ recognition of the novelty of our framework and the strength of its rigorous theoretical foundation. Based on the reviewers suggetions, we extended our empirical evaluation of FDBM in both setting - paired and unpaired data.

**Paired Data** We included an additional evaluation of FDBM on a cell differentiation task, where we find that FDBM performs best in the rough regime, achieving better performance compared to all baselines in $3$ out of $4$ metrics, including the distributional Wasserstein-2 distance and the RMSD to the true target. Furthermore, to address the reviewer’s concern regarding whether FDBM generates meaningful and realistic conformations when predicting conformational changes in proteins, we added the $\Delta \text{RMSD}$ metric, which verifies that the generated conformations are closer to the holo structures than to the initial apo conformations.

**Unpaired Data** We improved the robustness of our empirical results on the task of unpaired image translation by adding the translation tasks Dogs → Wild, Wild → Dogs, Dogs → Cats, and Cats → Dogs. In $3$ out of $4$ sub-domains, we observe clear improvements in both FID and LPIPS, further validating the effectiveness of FDBM in diverse unpaired translation scenarios.

Lastly, we respectfully disagree with the assessment by one reviewer that the potential audience of this paper is too narrow. We believe our paper should be evaluated on the strength of its contributions, the clarity of its revised exposition, and its scientific merit in both theoretical and applied domains.

We are confident that FDBM is a widely applicable framework that enables the use of fractional noise for generative bridge modeling in machine learning across a broad range of applications. We thank the reviewers again for their valuable feedback and their time reviewing our paper.

With best regards,
The authors

---

### Decision · Program_Chairs · 2025-09-17

**Decision:**

Accept (poster)

**Comment:**

The author proposes FDBM, a framework designed to better capture long-range dependencies and temporal correlations often observed in real-world systems—areas where existing diffusion models may fall short. FDBM achieves this by employing a Markovian approximation to fractional Brownian motion (MA-fBM), implemented as a finite mixture of Ornstein–Uhlenbeck (OU) processes. This formulation enables efficient simulation and learning while preserving essential non-Markovian statistical properties.

* Good point
I find this idea both interesting and novel, and I vote in favor of acceptance.

* Weakness/suggestion
That said, as one reviewer noted, the paper is difficult to follow for readers outside of experts in diffusion models. I strongly encourage the authors to address this in their revision to make the presentation more accessible to a broader audience.